# LLMOPT: Learning to Define and Solve General Optimization Problems from Scratch

**Caigao Jiang**[♥,*], **Xiang Shu**[♣,*], **Hong Qian**[♣,†], **Xingyu Lu**[♥,†],
**Jun Zhou**[♥], **Aimin Zhou**[♣], **Yang Yu**[♦]

[♣] East China Normal University, China   [♥] Ant Group, China   [♦] Nanjing University, China

`shux@stu.ecnu.edu.cn`, `{hqian,amzhou}@cs.ecnu.edu.cn`,
`{caigao.jcg,sing.lxy,jun.zhoujun}@antgroup.com`, `yuy@nju.edu.cn`

## Abstract

Optimization problems are prevalent across various scenarios. Formulating and then solving optimization problems described by natural language often requires highly specialized human expertise, which could block the widespread application of optimization-based decision making. To automate problem formulation and solving, leveraging large language models (LLMs) has emerged as a potential way. However, this kind of approach suffers from the issue of optimization generalization. Namely, the accuracy of most current LLM-based methods and the generality of optimization problem types that they can model are still limited. In this paper, we propose a unified learning-based framework called LLMOPT to boost optimization generalization. Starting from the natural language descriptions of optimization problems and a pre-trained LLM, LLMOPT constructs the introduced five-element formulation as a universal model for learning to define diverse optimization problem types. Then, LLMOPT employs the multi-instruction tuning to enhance both problem formalization and solver code generation accuracy and generality. After that, to prevent hallucinations in LLMs, such as sacrificing solving accuracy to avoid execution errors, the model alignment and self-correction mechanism are adopted in LLMOPT. We evaluate the optimization generalization ability of LLMOPT and compared methods across six real-world datasets covering roughly 20 fields such as health, environment, energy and manufacturing, etc. Extensive experiment results show that LLMOPT is able to model various optimization problem types such as linear/nonlinear programming, mixed integer programming, and combinatorial optimization, and achieves a notable 11.08% average solving accuracy improvement compared with the state-of-the-art methods. The code is available at `https://github.com/caigaojiang/LLMOPT`.

## 1 Introduction

Optimization problems are widespread in a variety of scenarios, such as job scheduling (Brandimarte, 1993), path planning (Hong et al., 2016; Li et al., 2023), matching problem (Wang et al., 2020), reinforcement leaning (Qian & Yu, 2021), revenue management (Trimborn et al., 2018) and security games (Brown et al., 2012; Qian et al., 2025), etc. Although powerful solvers are available, formally defining domain-specific optimization problems from natural language descriptions and then developing the solver code are highly complex tasks. It requires specialized domain knowledge, expert participation, and significant time investment. For example, in finance, the portfolio problems (Chen et al., 2021) often manage thousands of variables and complex constraints, such as full investment constraints, cardinality constraints and pre-assignment constraints, where expert intervention is crucial for achieving accurate (i.e., low simple regret) solutions due to the complexity and specialization of optimization problems.

With the rapid development of large language models (LLMs) like ChatGPT (Brown et al., 2020) and GPT-4 (OpenAI, 2023), using LLMs to automatically formulate and solve optimization prob-

---

[*]Both authors contribute equally to this research.
[†]Corresponding authors.

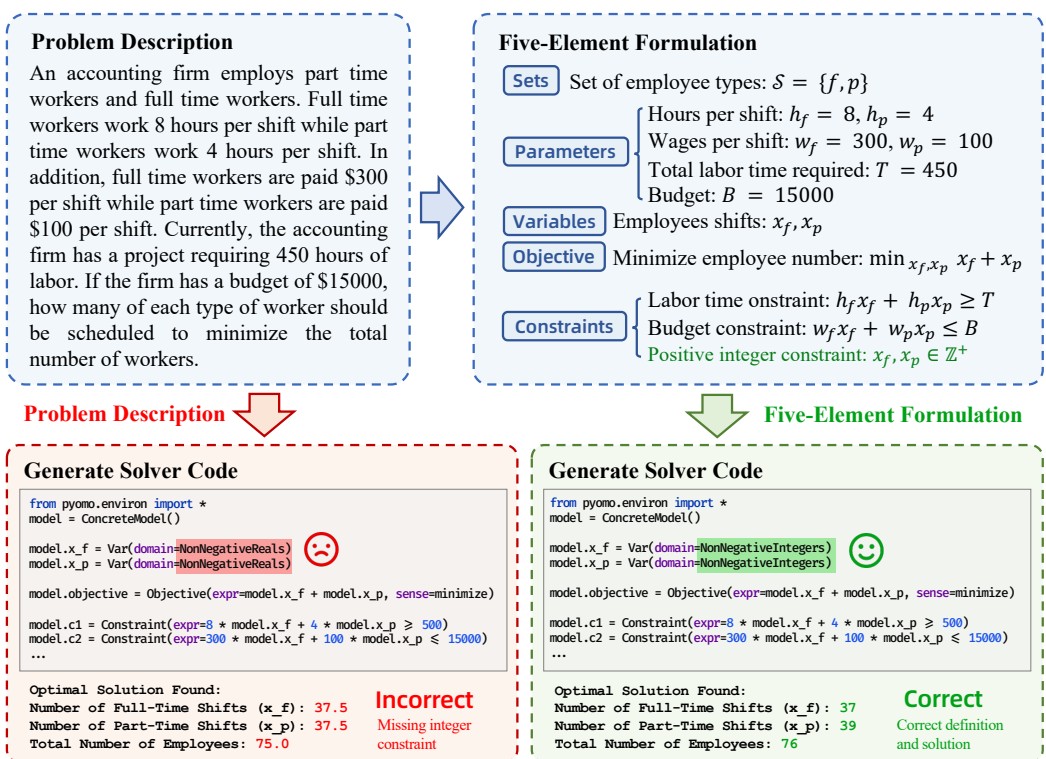

Figure 1: An example of the proposed five-element formulation, which provides a structured definition for general optimization problems. Using the five-element formulation as an intermediate step can lead to more accurate solver code and solution.

lems described by natural language is becoming increasingly attractive. Existing work on it can be roughly divided into prompt-based and learning-based methods. Pioneering studies like Chain-of-Expert (Xiao et al., 2024) and OptiMUS (AhmadiTeshnizi et al., 2024) explore prompt-based methods, using LLM interfaces to extract key information and solve optimization problems. Although these methods show promise, recent advances in learning-based methods, such as LLaMoCo (Ma et al., 2024) and ORLM (Tang et al., 2024), could show even greater potential. ORLM (Tang et al., 2024) reveals that a fine-tuned 7B open-source model can outperform larger pre-trained models like GPT-4, highlighting the value of learning-based methods for optimization tasks.

Although previous research has established the use of LLMs for formulating and solving optimization problems described by natural language, their optimization generalization remains limited, restricting broader applicability and deployment across diverse problems. *Optimization generalization*, which involves both *accuracy* and *generality* in assessing whether LLMs can effectively solve general optimization problems, remains a significant challenge. Specifically, accuracy refers to solving problems correctly, and generality refers to the ability to model and solve diverse optimization problem types across different task scenarios.

In this paper, we propose a unified learning-based framework called LLMOPT to significantly boost the optimization generalization. Through focusing on improving both accuracy and generality to effectively define and solve a wide range of optimization problems, this work tries to narrow the gap between methods and practical applications. Specifically, we propose the five-element formulation to define optimization problems to improve the accuracy of solving, as illustrated in Figure 1. To ensure the effectiveness of learning, we design a data augmentation process and perform data labeling, resulting in high-quality datasets. Leveraging these datasets, we implement multi-instruction supervised fine-tuning to improve the LLM's accuracy in both defining and solving optimization problems. Additionally, we introduce model alignment to further enhance accuracy and reduce the risk of hallucinations. We also develop an auto-testing process with a self-correction mechanism, enabling the accurate and automated resolution of optimization problems. Finally, the optimization generalization ability of LLMOPT is extensively evaluated on six real-world optimization and operation problem datasets involving roughly 20 domains, including health, environment, energy and manufacturing, etc. The results show that LLMOPT effectively handles various optimization

problems, such as linear and nonlinear programming, mixed integer programming and combinatorial optimization, etc. Ablation studies validate the contributions of problem definition and model alignment to improving accuracy. Notably, *LLMOPT achieves an average accuracy improvement of 11.08% over state-of-the-art methods*.

The consequent sections respectively introduce the related work, present the proposed LLMOPT, show and analysis the empirical results, discuss important issues, and finally conclude the paper.

## 2 RELATED WORK

**LLMs for Formulating and Solving Optimization Problems**. The NL4Opt competition (Ramamonjison et al., 2021) encourages researchers to explore how LLMs can be leveraged to solve optimization problems efficiently (Huang et al., 2024; Tang et al., 2024; Xiao et al., 2024; AhmadiTeshnizi et al., 2024; Ma et al., 2024). Related work can be divided into prompt-based and learning-based methods. The prompt-based methods utilize existing LLM interfaces to automate the solving optimization problems. Chain-of-Expert (CoE) (Xiao et al., 2024) introduces an agent-based workflow, where specific tasks are assigned to LLM agents at each stage of reasoning, while OptiMUS (AhmadiTeshnizi et al., 2024) employs tailored prompts designed specifically for solving linear programming and mixed-integer linear programming problems. The learning-based method remains relatively underexplored. LLaMoCo (Ma et al., 2024) introduces an instruction tuning framework in a code-to-code manner and demonstrates its efficacy through model fine-tuning. ORLM (Tang et al., 2024) proposed a semi-automated approach to generating synthetic training data during the instruction tuning phase, addressing the data-hungry nature of optimization modeling. Besides, Mamo (Huang et al., 2024) offers a benchmark for mathematical modeling with two optimization datasets, and introduces a standardized process to generate solver code using LLMs.

**LLMs as Components of Optimizers**. Yang et al. (2024b) propose to use LLMs as optimizers, inspired by Bayesian optimization (Garnett, 2023). One LLM iteratively optimizes prompts, which are then used to guide another LLM in performing specific optimization tasks. LLMs can also serve as components within an optimizer (Guo et al., 2024; Yang & Li, 2023; Liu et al., 2024a; 2023; 2024b). Guo et al. (2024) uses LLMs to as the crossover and mutation operators in genetic algorithms, thereby finding high-quality prompts. When solving the traveling salesman problem, Liu et al. (2024a) uses LLMs to generate new trajectories based on existing ones, iteratively optimizing them to produce higher-quality solutions. Liu et al. (2023) and Yang & Li (2023) apply LLMs to specific multi-objective optimization tasks. Liu et al. (2024b) employs LLMs to simulate both the acquisition function and the surrogate model in the Bayesian optimization process.

## 3 METHODOLOGY

### 3.1 AN OVERVIEW OF THE PROPOSED LLMOPT FRAMEWORK

This work aims to achieve a better optimization generalization of defining and solving the optimization problems through a learning-based approach. To this end, this paper proposes the LLMOPT framework, as illustrated in Figure 2, which comprises three key components: data, learning, and auto-testing. The following sections provide a detailed introduction: First, in Section 3.2, we introduce the five-element formulation to define general optimization problems well, and discuss the augmentation and labeling of training data. Next, in Section 3.3, we detail the multi-instruction supervised fine-tuning (SFT) process and model alignment to enhance solving accuracy. Finally, in Section 3.4, we present the auto-testing process with self-correction.

### 3.2 DATA: DEFINING AND LABELING GENERAL OPTIMIZATION PROBLEMS

#### 3.2.1 UNIVERSAL FORMULATION TO DEFINE GENERAL OPTIMIZATION PROBLEMS

To enable LLMs to formulate more general optimization problems described in natural language, we propose the five-element, a universal formulation to define optimization problems. Firstly, in mathematics, an optimization problem can be formally represented as the following expression:

$$\min_{\boldsymbol{x} \in \mathcal{X} \subseteq \mathbb{R}^D} f(\boldsymbol{x}), \quad \text{s.t. } G(\boldsymbol{x}) \leq \boldsymbol{c}, \tag{1}$$

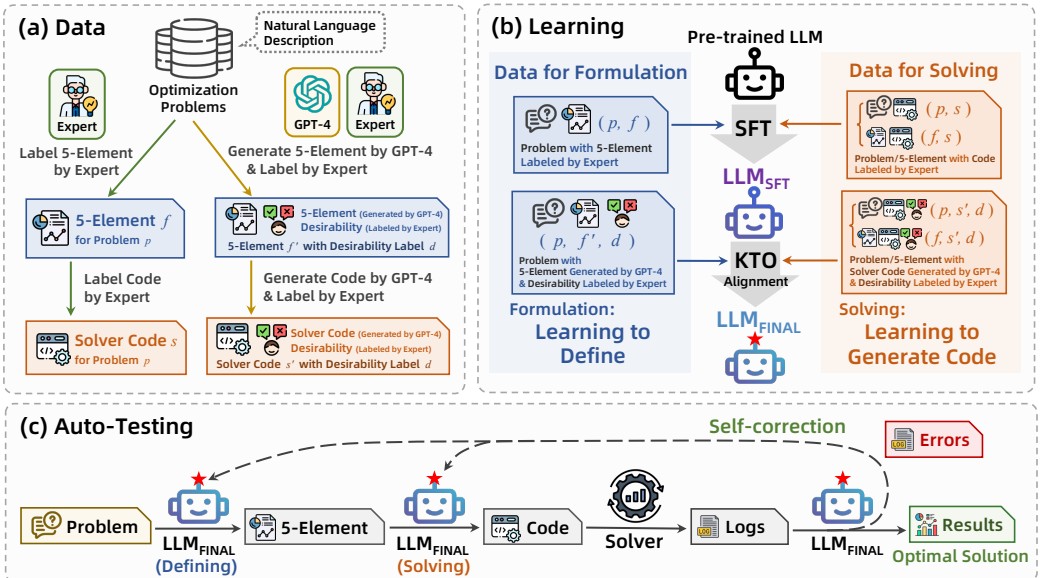

Figure 2: The framework of LLMOPT. Sub-figure (a) shows the data labeling process, where experts and GPT-4 work together to label both the five-element formulation and solver code. Sub-figure (b) shows the learning process, in which multi-instruction supervised fine-tuning and model alignment are employed to learn-to-define and generate code. Sub-figure (c) shows the auto-testing process with self-correction mechanism, which can define and solve optimization problems automatically.

where $\boldsymbol{x} = (x_1, x_2, \ldots, x_D)^\top$ is the $D$-dimensional decision variable, $\mathcal{X} \subseteq \mathbb{R}^D$ is the feasible domain of $\boldsymbol{x}$, and the goal is to minimize the objective function $f : \mathcal{X} \to \mathbb{R}$. The constraints are represented by the vector-valued function $G(\boldsymbol{x}) : \mathbb{R}^D \to \mathbb{R}^m$, where $G(\boldsymbol{x}) \leq \boldsymbol{c}$ denotes a series of inequality constraints, and $\boldsymbol{c} = (c_1, c_2, \ldots, c_m)^\top$ is the vector of upper bounds for these constraints.

In practice, a vast majority of optimization problems can be formulated as shown in Formula 1. Based on it, the five-element formulation provides a structured and intuitive way to define optimization problems, retaining essential descriptions and making them more accessible and understandable, even for LLMs. As the name suggests, the five-element consist of the five key components of an optimization problem: *Sets, Parameters, Variables, Objective* and *Constraints*. As shown in Figure 1, the element *Variables, Objective*, and *Constraints* describe the decision variables $\boldsymbol{x}$, the objective function $f(\boldsymbol{x})$, and the constraints $G(\boldsymbol{x})$ in Formula 1, along with concise descriptions for each. Meanwhile, *Sets* and *Parameters* provide detailed information of indices and descriptions, as well as the specific numerical values of the parameters involved in the objective function and constraints. Importantly, the five-element provides a more accurate formulation of the problem by mining and preserving key descriptions of the problem. For example, in Figure 1, the positive integer constraint is implicit in the problem but is effectively captured by the five-element formulation.

As a universal formulation, five-element can define various types of optimization problems, thereby enhancing the optimization generalization ability. For instance, integer programming can be modeled by adjusting the feasible region $\mathcal{X}$, while linear and non-linear programming constraints are captured by different forms of $G(\boldsymbol{x})$. More complex problems, such as multi-objective optimization, extend the objective function to a vector-valued form $F(\boldsymbol{x})$. Additional examples illustrating the formulation of various optimization problems are provided in Appendix J.

### 3.2.2 DATA AUGMENTATION AND LABELING

The five-element formulation enables the mapping of optimization problems described in natural language to a universal structure, which can then be used to generate solver code. However, the effectiveness of learning is significantly influenced by the quality, quantity, and distribution of training data. Current available datasets are limited in these aspects and lack proper labels of formulations and solver code. Therefore, data augmentation and labeling are employed to address these gaps.

Firstly, we collect almost all existing optimization problem datasets, including NL4Opt (Ramamonjison et al., 2021), Mamo (EasyLP and ComplexLP) (Huang et al., 2024), IndustryOR (Tang et al.,

2024), NLP4LP (AhmadiTeshnizi et al., 2024) and ComplexOR (Xiao et al., 2024) whose detailed information is introduced in the Appendix A. Subsequently, 100 samples are randomly selected from each dataset as the reserved test dataset. For datasets with fewer than 100 samples, all data are used for testing. The remaining samples are used for data augmentation, ensuring a clear separation between training and testing data. In the data augmentation process, LLMs effectively generate data through prompt engineering (Tang et al., 2024; Luo et al., 2023). To build a high-quality dataset, seven distinct instructions are applied to 1,763 seed problems. These instructions comprehensively extend the original problems from various perspectives, such as modifying constraints, changing scenarios, altering optimization types, and branching out from the original problem, among other approaches. Finally, experts review the generated problems, removing those with unclear descriptions or infeasible solutions to ensure data diversity and quality. Prompts are detailed in Appendix I.

Then, the labeling of optimization problems using five-element and solver code is performed by experts. GPT-4 (OpenAI, 2023) is also used to generate labels with uncertain correctness, which can be used in the model alignment. As shown in Figure 2(a), for each optimization problem $p$, experts label the five-element formulation $f$ and solver code $s$, and GPT-4 generates the formulation $f'$ and code $s'$, respectively. Since GPT-4 may produce errors, experts validate it and assign a *desirability* label $d'$ to indicate whether GPT-4's label is accurate (True) or not (False). This process produces two types of labeled data: fully accurate formulations and solver codes, and those with potential errors but validated by experts. The two kinds of data are well-suited for the processes of multi-instruction supervised fine-tuning and model alignment, as discussed in the next section. The detailed process of data labeling and augmentation is introduced in Appendix K.

## 3.3 LEARNING: MULTI-INSTRUCTION SFT AND MODEL ALIGNMENT

### 3.3.1 MULTI-INSTRUCTION SFT

Direct utilization of LLM to address optimization problems described in natural language often results in inaccuracies, primarily due to their inability to comprehensively capture implicit information. To address this issue, we enhance the capability to both define and solve the problems through multi-instruction supervised fine-tuning (SFT).

The multi-instruction dataset $\mathcal{D}_{\text{SFT}} = \mathcal{D}_{\text{SFT}}^f \cup \mathcal{D}_{\text{SFT}}^s$, labeled by experts, is designed to improve both the problem formulation and solving code generation abilities of LLM. The formulation dataset $\mathcal{D}_{\text{SFT}}^f = \{(u_i, v_i)\}_i^{N_f}$ focuses on enhancing the LLM's ability to accurately define general optimization problems by providing data pairs $(u_i, v_i)$, where $u_i$ represents a problem $p$ after being applied to a template, and $v_i$ is its corresponding five-element formulation $f$. Meanwhile, the solving dataset $\mathcal{D}_{\text{SFT}}^s = \{(u_i, v_i)\}_i^{N_s}$ focuses on improving the LLM's ability to generate solver code. It contains two types of data: in one, $u_i$ is the problem $p$ and $v_i$ is the solver code $s$; in the other, $u_i$ is the formulation $f$ and $v_i$ is the solver code $s$. Both are aligned with the appropriate templates and all the labels in $\mathcal{D}_{\text{SFT}}$ are correctly labeled by experts. The instruction templates used in the learning process are detailed in Appendix H, which provides a comprehensive explanation, including a more in-depth description of the five-element to facilitate better understanding by the LLM.

Then, given the instruction $u$ and its label $v$, the objective of SFT is to maximize the conditional probability $p(v \mid u)$. Assuming that $p(v \mid u) = \prod_{i=1} p(v_i \mid v_{0:i-1}, u)$, this leads to minimizing the negative log-likelihood:

$$\mathcal{L}_{\text{SFT}}(\theta) = -\mathbb{E}_{(u,v) \sim \mathcal{D}_{\text{SFT}}} \sum_{i=1} \log \pi(v_i \mid v_{0:i-1}, u; \theta), \qquad (2)$$

where $\mathcal{D}_{\text{SFT}}$ denotes the multi-instruction SFT dataset, which combines instructions from $\mathcal{D}_{\text{SFT}}^f$ and $\mathcal{D}_{\text{SFT}}^s$ to train the LLM in defining and solving optimization problems. $\pi(\cdot)$ represents the predicted probability distribution of the LLM, and $\theta$ denotes its parameters.

### 3.3.2 MODEL ALIGNMENT

Despite SFT training the model to learn how to write solving code, LLMs may still exhibit hallucination when faced with novel problems. This hallucination manifests as outputs that appear plausible but are, in fact, fabricated or inaccurate. To mitigate hallucinations, we incorporate model

alignment, which is not utilized by other learning-based methods. We use Kahneman-Tversky Optimization (KTO)(Ethayarajh et al., 2024) as our alignment method, as it avoids reward model bias and mitigates the data demands of ranking-based approaches like PPO(Ouyang et al., 2022) and DPO (Rafailov et al., 2023).

The KTO algorithm aligns the model using *(instruction, completion, desirability)* data, where desirability reflects the correctness of the completion using a binary label: True or False. Specifically, the dataset $\mathcal{D}_{\text{KTO}} = \{(u_i, v_i, d_i)\}_i^{N_{\text{KTO}}}$, used by KTO, contains GPT-4 completions with expert-assigned desirability labels. The instructions $u_i$ in $\mathcal{D}_{\text{KTO}}$ match those in $\mathcal{D}_{\text{SFT}}$, covering formulation and code generation tasks. However, the completions $v_i$, generated by GPT-4, do not always fulfill the instructions $u_i$ correctly, and experts assign desirability labels $d_i$ to achieve the model alignment.

Building upon the previous work by Rafailov et al. (2023), it is evident that the LLM can be interpreted as a reward function. Consequently, we can express the optimal reward for KTO as follows:

$$r_{\text{KTO}}^*(u, v) = \beta \log \frac{\pi^*(v \mid u)}{\pi_{\text{ref}}(v \mid u)} \,. \tag{3}$$

In this equation, $r^*(\cdot)$ denotes the optimal reward function of the completion $v$ in response to the instruction $u$, and $\pi^*(\cdot)$ represents the optimal model (i.e., the KTO model), $\pi_{\text{ref}}(\cdot)$ indicates the reference model (i.e., the SFT model), and $\beta$ serves as the scaling factor. This ratio measures the relative confidence of the optimal model in generating $v$ compared to the reference model.

Inspired by classic prospect theory (Tversky & Kahneman, 1992), the value function employs a logistic transformation on the adjusted ratio of log probabilities, thereby assessing the value of each generation in terms of its desirability and the divergence of the policy from the reference. By substituting the exponential function with a sigmoid function, we derive the value function as follows:

$$\phi_{\text{KTO}}(u, v; \beta) = \begin{cases} \sigma(r_{\text{KTO}}(u, v) - z_{\text{ref}}) & \text{if } d = \text{True} \mid u, v \,, \\ \sigma(z_{\text{ref}} - r_{\text{KTO}}(u, v)) & \text{if } d = \text{False} \mid u, v \,, \end{cases} \tag{4}$$

where $z_{\text{ref}}$ is defined as the reference point, formulated as: $z_{\text{ref}} = \beta \text{KL}(\pi^*(v' \mid u) \mid\mid \pi_{\text{ref}}(v' \mid u))$.

Another essential component of KTO loss is the weight function, which can be expressed as follows:

$$w(v) = \begin{cases} \lambda_D & \text{if } d = \text{True} \mid u, v \,, \\ \lambda_U & \text{if } d = \text{False} \mid u, v \,. \end{cases} \tag{5}$$

The weight function $w(v)$ assigns weights to the loss depending on the desirability of the outcome $v$, with $\lambda_D$ and $\lambda_U$ representing the weights for desirable and undesirable outcomes, respectively.

Finally, the KTO loss function is formulated as:

$$\mathcal{L}_{\text{KTO}}(\pi^*, \pi_{\text{ref}}) = \mathbb{E}_{(u,v,d) \sim \mathcal{D}_{\text{KTO}}}[w(v)(1 - \phi_{\text{KTO}}(u, v; \beta))] \,. \tag{6}$$

The KTO loss function encourages the optimal model $\pi^*$ to produce completions that align more closely with expert-labeled data, improving the overall optimization generalization of both problem formulation and solver generation.

## 3.4 Auto-Testing: Formulation, Solving and Self-Correction

In the auto-testing process, first, the optimization problem is formulated using the five-element framework based on its natural language description; second, the solver code is generated for the five-element formulation and executed; finally, the solver's running logs, including output results and errors, are analyzed to determine whether self-correction is necessary. The auto-testing process automates the entire workflow of problem definition and solver code generation, while also integrating the self-correction mechanism for continuous improvement.

Inspired by Chen et al. (2024), to enhance optimization generalization, we implement self-correction to automatically analyze the output results and identify errors arising during the execution of the solver code. Specifically, the instruction is organized around the problem, the five-element formulation, the solver code, and the execution output results, following a template outlined in Appendix H. The self-correction LLM then determines whether further resolution is necessary or whether the optimal solution has been achieved. If resolution is needed, the model generates an analysis with suggestions and decides to return to whether the problem formulation step or the code generation step. The self-correction mechanism ensures a more robust and adaptive optimization process, improving optimization generalization especially in the accuracy of formulations and final solutions.

## 4 EXPERIMENT

To analyze the performance of LLMOPT, we conduct SFT and model alignment based on the open-source LLM Qwen1.5-14B (Bai et al., 2023) and compare it with various learning-based and prompt-based methods on extensive datasets. The experiments aim to answer four key questions below.

**(Q1)** Learning-based vs. Prompting-based Methods: What advantages do learning-based methods, such as LLMOPT, have over LLMs that rely solely on prompt engineering?

**(Q2)** Optimization Generalization (Accuracy and Generality): To what extent can LLMOPT improve optimization generalization ability compared with existing methods?

**(Q3)** Importance of Problem Definition in LLMOPT: How does the proposed five-element formulation as an intermediate step contribute to boost accuracy in solving optimization tasks?

**(Q4)** Effectiveness of Model Alignment in LLMOPT: How effective is model alignment in enhancing the solving accuracy of LLMs for optimization tasks?

The four questions are answered in order in the next sections which include a detailed introduction to the experiments, followed by the analysis of results.

### 4.1 EXPERIMENTAL SETUP

The experiments are conducted on six real-world optimization and operation task datasets, namely NL4Opt (Ramamonjison et al., 2021), Mamo (EasyLP and ComplexLP) (Huang et al., 2024), IndustryOR (Tang et al., 2024), NLP4LP (AhmadiTeshnizi et al., 2024), ComplexOR (Xiao et al., 2024). These datasets encompass about 20 scenario and 7 types of optimization problems. Detailed descriptions of these datasets are provided in Appendix A. Training and testing data are strictly separated. For the NL4Opt, Mamo (EasyLP and ComplexLP) datasets, we shuffle the original datasets and randomly extract 100 data from each dataset as the test datasets, and the remaining data are used as seed data for data augmentation. The IndustryOR dataset retains its original partitioning. Due to the limited data in NLP4LP and ComplexOR, all data from these datasets are used for testing and excluded from the training process.

In the experiment, we use three performance metrics to comprehensively evaluate the optimization generalization of the algorithm, namely *Execution Rate (ER)*, *Solving Accuracy (SA)*, and *Average Solving Times (AST)*. Specifically, ER refers to the proportion of solutions whose code can run without any errors and has running results output. SA refers to the proportion of solutions that correctly solve the optimization problem, i.e., find the optimal solution. AST refers to the average number of times the self-correction process is performed during the test. In our experiment, the maximum number of self-correction re-solves is set to 12 times.

### 4.2 ANALYSIS OF OPTIMIZATION GENERALIZATION

In this section, we compare LLMOPT with prompt-based methods (Reflexion (Shinn et al., 2023), Chain-of-Experts (Xiao et al., 2024), OptiMUS (AhmadiTeshnizi et al., 2024)), learning-based methods (ORLM (Tang et al., 2024) built on Mistral-7B, Deepseek-Math-7B-Base, LLaMa3-8B), and GPT-4 (OpenAI, 2023), demonstrating the optimization generalization capability of LLMOPT.

Table 1: Comparison of the SA metric between LLMOPT and learning-based methods. The results for ORLM are cited from Tang et al. (2024). Underlined results indicate the previous SOTA, while **bold** results indicate the current SOTA.

| Datasets | | NL4Opt | Mamo Easy | Mamo Complex | IndustryOR |
|---|---|---|---|---|---|
| GPT-4 Directly | | 47.3% | 66.5% | 14.6% | 28.0% |
| ORLM | Mistral-7B | 84.4% | 81.4% | 32.0% | 27.0% |
| | Deepseek-Math-7B-Base | 86.5% | 82.2% | 37.9% | 33.0% |
| | LLaMa3-8B | 85.7% | 82.3% | 37.4% | 38.0% |
| **LLMOPT (Ours)** | Qwen1.5-14B | **93.0%** | **97.0%** | **68.0%** | **46.0%** |
| **Improvement Rate ↑** | | **+6.5%** | **+14.7%** | **+30.1%** | **+8.0%** |

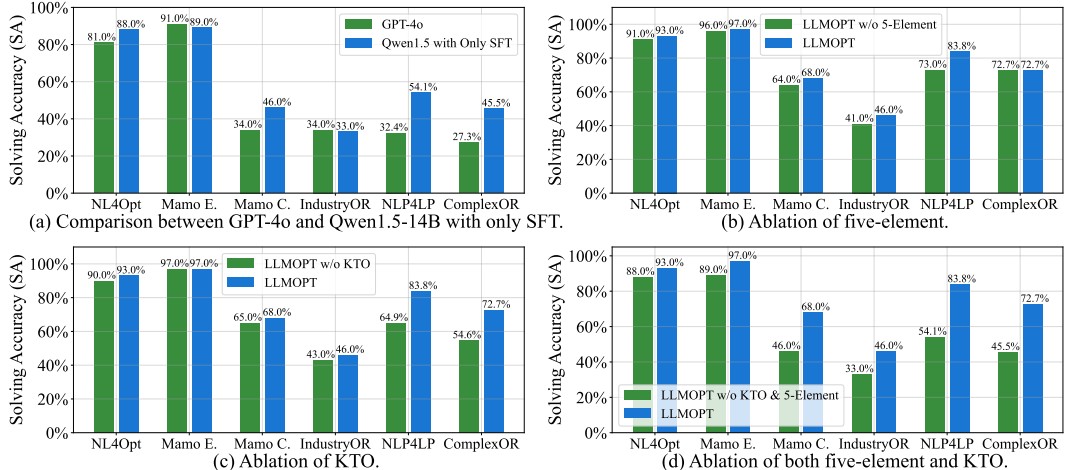

Figure 3: The results of the SA metric. Sub-figure (a) compares the SA performance of GPT-4o and the Qwen1.5-14B model with only SFT, showing the potential of learning-based methods. Sub-figures (b), (c) and (d) perform ablation on five-element and KTO. The Mamo Easy and Mamo Complex datasets are abbreviated as Mamo E. and Mamo C., due to space constraints.

Table 2: Comparison of the SA metric between LLMOPT and prompt-based methods. The results for Reflexion, Chain-of-Experts, and OptiMUS are cited from Xiao et al. (2024). Underlined results indicate the previous SOTA, while **bold** results indicate the current SOTA.

| Datasets | NL4Opt | NLP4LP | ComplexOR |
|---|---|---|---|
| GPT-4 Directly | 47.3% | 35.8% | 9.5% |
| Reflexion | 53.0% | 46.3% | 19.1% |
| Chain-of-Experts | 64.2% | 53.1% | 38.1% |
| OptiMUS | 78.8% | 72.0% | 66.7% |
| **LLMOPT (Ours)** | **93.0%** | **83.8%** | **72.7%** |
| **Improvement Rate ↑** | **+14.2%** | **+11.8%** | **+6.0%** |

**Learning-based vs. Prompting-based Methods (Answer to Q1)**. To demonstrate the potential of the learning-based method, we perform SFT on Qwen1.5-14B Bai et al. (2023) and compare it with GPT-4o using the same prompt (Appendix H). As shown in Figure 3(a), the results indicate that LLM with only SFT can achieve comparable performance to GPT-4o. For the complete LLMOPT (including model alignment and self-correction), as shown in Figure 4, LLMOPT outperforms both GPT-4o and GPT-4-Turbo, achieving higher solving accuracy with fewer solving times across all six datasets. All these results demonstrate the potential of learning-based methods.

**Accuracy (Answer to Q2-1)**. We compare the solving accuracy (SA) of LLMOPT with prompt-based and learning-based methods as shown in Table 1 and 2, respectively. To ensure consistency in reproduction, we only cite the results from the original paper. LLMOPT achieves state-of-the-art (SOTA) performance in SA across six datasets, outperforming both prompt-based and learning-based methods. Compared to learning-based methods, defining with the five-element formulation and self-correction process improves the accuracy, with an average increase of 14.83% on four datasets. For prompt-based methods, SFT and model alignment enhance the LLM's ability to solve optimization problems, yielding a 10.67% average improvement across three datasets. Overall, LLMOPT improves SOTA performance by an average of 11.08% across the six datasets.

**Generality (Answer to Q2-2)**. To analyze the generality of LLMOPT on general optimization problems, we conduct testing on six datasets, each covering diverse optimization types, such as linear programming, nonlinear programming, integer programming, mixed-integer programming, multi-objective optimization, and combinatorial optimization, as detailed in Appendix G. These datasets span over 20 different fields, including agriculture, energy, and healthcare, as detailed in Appendix F. LLMOPT achieves state-of-the-art results across all datasets including a variety of optimization problems, demonstrating its generality. Appendix L shows the specific experimental results by question types. Moreover, despite the absence of NLP4LP and ComplexOR data in the

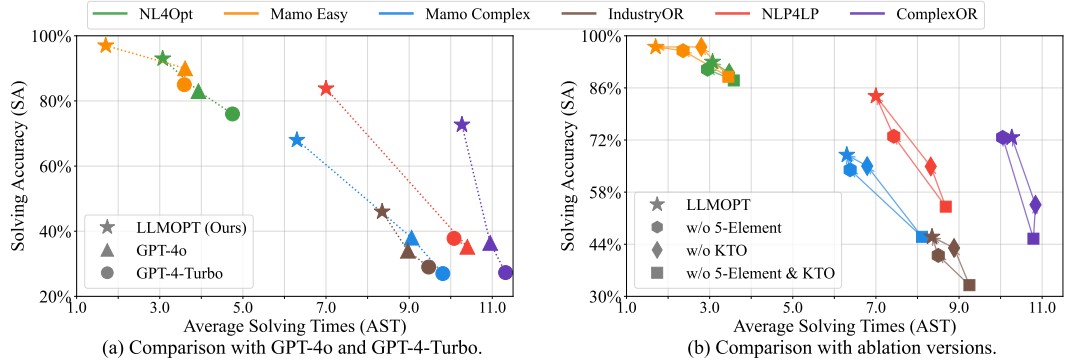

Figure 4: Comparison of SA and AST between LLMOPT, GPT-4, and ablated versions.

Table 3: The ablation results of LLMOPT without (w/o) five-element formulation, KTO alignment and self-correction, respectively. In the experiments without self-correction, the model is called only once, resulting in an AST of 1.00. **Bold** indicates the best performance for each metric.

| Metrics | ER | SA | AST | ER | SA | AST | ER | SA | AST |
|---|---|---|---|---|---|---|---|---|---|
| **Dataset** | **NL4Opt** | | | **Mamo Easy** | | | **Mamo Complex** | | |
| **LLMOPT** | **99.0%** | **93.0%** | 3.07 | **100.0%** | **97.0%** | **1.70** | 97.0% | **68.0%** | **6.30** |
| w/o five-element | **99.0%** | 91.0% | **2.95** | **100.0%** | 96.0% | 2.36 | **98.0%** | 64.0% | 6.38 |
| w/o KTO | **99.0%** | 90.0% | 3.47 | **100.0%** | 97.0% | 2.80 | 95.0% | 65.0% | 6.79 |
| w/o self-correction | 79.0% | 57.0% | (1.00) | 85.0% | 71.0% | (1.00) | 52.0% | 32.0% | (1.00) |
| **Dataset** | **IndustryOR** | | | **NLP4LP** | | | **ComplexOR** | | |
| **LLMOPT** | **92.0%** | **46.0%** | 8.35 | **100.0%** | **83.8%** | 7.00 | 94.7% | **72.7%** | 10.27 |
| w/o five-element | **92.0%** | 41.0% | 8.50 | 96.9% | 73.0% | 7.43 | 94.7% | 72.7% | **10.05** |
| w/o KTO | **92.0%** | 43.0% | 8.88 | 89.2% | 64.9% | 8.32 | **100.0%** | 54.6% | 10.84 |
| w/o self-correction | 55.0% | 31.0% | (1.00) | 47.7% | 35.2% | (1.00) | 47.4% | 18.2% | (1.00) |

training dataset, LLMOPT still achieves SOTA performance on them, as shown in Table 1, further proving its generality. Furthermore, seven examples of the five-element formulation applied to various optimization problems are provided in Appendix J to illustrate the generality of this definition. The high accuracy and generality demonstrate the optimization generalization capability of LLMOPT.

## 4.3 ABLATION STUDY

In this section, we conduct comprehensive ablation experiments, including LLMOPT without five-element formulation, without KTO alignment, and without self-correction. The results of these experiments are shown in Figure 3, Figure 4, and Table 3. We also design detailed ablation experiments on the self-correction mechanism, which are introduced in Appendix M.

**Importance of Problem Definition (Answer to Q3)**. In order to explore the importance of problem definition, we evaluate three metrics on the ablation experiments: Execution Rate (ER), Solving Accuracy (SA), and Average Solving Time (AST). As detailed in Table 3 and Figure 3(b), these results show that using the five-element formulation as the problem definition improves SA across all six datasets. However, this definition can sometimes lower the ER, as the LLM may oversimplify the problem without it, producing error-free but inaccurate code. In contrast, using the five-element approach ensures correct code generation, improving SA at the cost of slightly reduced ER.

**Effectiveness of Model Alignment (Answer to Q4)**. Alignment typically enhances the efficiency and effectiveness of LLMs on specific tasks. As shown in Figure 4(b) and 3(c), the ablation results shows that KTO alignment not only significantly boosts SA but also reduces AST across all six datasets. Furthermore, as shown in Figure 4(b), 3(d), and Table 3, the combination of five-element and KTO alignment improves performance in terms of SA and AST on complex datasets.

## 5 DISCUSSION

**Attempt on Larger Models**. To explore the potential for further performance improvement of LL-MOPT on larger models, we deploy it on Qwen2-72B (Yang et al., 2024a). The detailed results are provided in Appendix D. We conduct experiments on two complex task datasets, Mamo Complex and IndustryOR, and the results show that LLMOPT based on Qwen2-72B shows significant improvements in both SA and ER compared with its performance on Qwen1.5-14B. Given the trade-off between performance and the costs of training and deployment, i.e., green computing issue, we opt not to use larger models, since Qwen1.5-14B already achieves state-of-the-art performance.

**Compared with OpenAI o1 Model**. OpenAI o1 model (OpenAI, 2024) has recently garnered attention for its strong reasoning ability. However, due to the limited availability of its API and restrictions on the number of weekly trials, our evaluation of the o1 model's performance is limited. We conduct experiments on 10 easy and 10 complex problems from the Mamo Complex dataset. With a single call, the o1 model successfully generated code to solve 7 easy problems and 5 complex problems. These results suggest that the o1 model is more accurate in solving optimization problems compared with GPT-4 series models. However, due to the lack of open access, the absence of detailed technical specifications and training data description of o1, as well as the limitations on usage and high costs, fine-tuning open-source large models with LLMOPT presents a more cost-effective solution for achieving better optimization generalization in real industrial scenarios.

**The Seesaw Issue of LLMs**. The seesaw issue in LLMs refers to the trade-off between improving the performance on specialized tasks and its generalization across diverse tasks, where gains in one area often result in losses in another. To assess whether enhancing the LLM's ability to define and solve optimization problems affects its performance on other tasks, we compared the model's performance before and after fine-tuning across 10 general tasks, including math, code, classification, information extraction, open QA, closed QA, text generation, brainstorming, rewriting and summarization. The results indicate that the fine-tuned model showed performance improvements in 6 tasks and declines in 4 tasks, with an average performance increase of 0.3% across all tasks. Importantly, no significant trade-off or seesaw effect is observed. Detailed results are provided in Appendix E.

**About High-Quality Training Data**. High-quality data are crucial for fine-tuning LLMs (Villalobos et al., 2024). Accurate and diverse training data allow the model to improve its task-specific performance. However, optimization problem data described in natural language are relatively scarce. Although we have collected as many optimization problem datasets as possible, which are detailed in Appendix A, these datasets often lack high-quality labels. For instance, in the IndustryOR dataset (Tang et al., 2024), some data have incorrectly labeled optimal solutions, and the NL4Opt dataset (Ramamonjison et al., 2021) does not provide optimal solution annotations but only entity labels. This highlights the scarcity of optimization problems training data. Although prompt engineering for data augmentation is employed in this work, expert labeling remains a time-consuming and labor-intensive process. Efficiently gathering, synthesizing and generating more diverse and well-labeled high-quality data remains an issue that cannot be ignored in this research direction in future. Moreover, for large-scale problems, data are typically stored in specific databases or files rather than extracted from natural language descriptions. The understanding of these data structures remains a new topic that requires further exploration.

## 6 CONCLUSION

This paper focuses on the challenge of optimization generalization in LLMs, including the accuracy in solving optimization problems and the generality of the problem types that LLMs can handle. We propose a learning-based framework called LLMOPT, which significantly improves LLMs' ability to define and solve general optimization problems through multi-instruction supervised fine-tuning and model alignment. LLMOPT introduces the five-element formulation as a universal definition for various types of optimization problems to enhance solving accuracy. LLMOPT is extensively evaluated on a wide range of optimization tasks. It achieves state-of-the-art optimization generalization ability across all of them, i.e., a notable 11.08% average solving accuracy improvement.

## ACKNOWLEDGMENTS

We would like to thank the anonymous reviewers for their constructive and valuable suggestions. This work is supported by the National Natural Science Foundation of China (No. 62476091), Ant Group and the Ant Group Research Intern Program.

## ETHICS AND REPRODUCIBILITY STATEMENT

**Ethics**. This work does not involve any human subjects, personal data, or sensitive information. All the test datasets used are publicly available, and no proprietary or confidential information is used. We follow recommendations to use the Azure OpenAI service when using GPT models.

**Reproducibility**. Experimental settings are described in Section 4 and Appendix B, and datasets included in Appendix A. The code is available at `https://github.com/caigaojiang/LLMOPT`.

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

APPENDIX

## A  DATASETS

### A.1  THE INTRODUCTION OF DATASETS

We conduct experiments on the following real-world optimization task datasets.

Table 4: The statistics of the optimization problem datasets.

| Dateset Name | # of Data | Notes |
|---|---|---|
| **NL4Opt** (Ramamonjison et al., 2021) | 1101 | All data are unlabeled optimal solution. |
| **Mamo Complex** (Huang et al., 2024) | 211 | - |
| **Mamo Easy** (Huang et al., 2024) | 652 | - |
| **IndustryOR** (Tang et al., 2024) | 100 | Another 3000 data without optimal solution. |
| **NLP4LP** (AhmadiTeshnizi et al., 2024) | 65 | All data are unlabeled optimal solution. |
| **ComplexOR** (Xiao et al., 2024) | 19 | All data are unlabeled optimal solution. |

**NL4Opt** (Ramamonjison et al., 2021). The NL4Opt Competition curates a dataset comprising 1101 annotated LPWPs across 6 diverse domains. Each set contained LPWPs from source domains like sales, advertising, and investment, ensuring representation across all splits. However, LPWPs from target domains such as production, transportation, and sciences were exclusively reserved for the development and test sets.

**Mamo** (Huang et al., 2024). The optimization dataset of Mamo benchmark consists of two parts, Easy LP and Complex LP. Easy LP contains 652 high school-level MILP problems for basic learning of linear and mixed integer linear programming. Complex LP provides 211 undergraduate-level challenges, integrating LP and MILP, suitable for advanced learners and researchers. These problems are more complex, covering different applications and theoretical challenges, suitable for advanced courses and research projects, and provide comprehensive development of optimization skills from basic to complex.

**IndustryOR** (Tang et al., 2024). IndustryOR is the first industrial dataset designed specifically for optimization modeling. It incorporates data from 13 different industries and covers a variety of real-world scenarios. IndustryOR consists of real operations research (OR) problems from eight different industries. It covers five types of optimization problems: linear programming, integer programming, mixed integer programming, nonlinear programming, and other special problem types. These problems are also divided into three difficulty levels. The training dataset of IndustryOR contains 3000 instances without labeling optimal solution and the test dataset contains 100 instances with optimal solution.

**NLP4LP** (AhmadiTeshnizi et al., 2024). NLP4LP (Natural Language Processing for Linear Programming) is a dataset that includes 65 samples we identified from its repository. These problems are sourced from optimization textbooks and lecture notes covering areas such as facility location, network flow, scheduling, and portfolio management. Each instance in NLP4LP includes a description, sample parameter data file, and optimal value derived from textbook solutions or manual solving, offering a range of complex optimization challenges of varying lengths.

**ComplexOR** (Xiao et al., 2024). ComplexOR dataset is developed with the collaboration of three specialists in operations research. We identify 19 samples from its repository, sourced from diverse references such as academic papers, textbooks, and real-world industrial scenarios. These problems encompass a broad spectrum of topics, including supply chain optimization, scheduling problems, and warehousing logistics.

### A.2  TRAINING DATASETS FOR MULTI-INSTRUCTION SFT AND MODEL ALIGNMENT

We introduce the training datasets used for multi-instruction SFT and model alignment, as illustrated in Table 5. After data augmentation, the SFT dataset comprises 9,828 instances, while the KTO alignment dataset contains 19,563 instances. To keep the generalization capability of the language model, we incorporated additional data into the datasets, which is open source and available at

Table 5: The number of samples in the dataset for SFT and model alignment.

| | | # of Optimization Data | | # of Other Data | | Total | |
|---|---|---|---|---|---|---|---|
| **SFT** | | 9828 | | 20000 | | 29828 | |
| **KTO** | **True Label** | 9827 | 19563 | 26384 | 30000 | 36212 | 49563 |
| | **False Label** | 9735 | | 3616 | | 13351 | |

`https://instructions.apps.allenai.org/` and `https://huggingface.co/datasets/argilla/ultrafeedback-binarized-preferences-cleaned-kto`. It is important to note that we do not utilize the entire datasets, but instead randomly selected 20,000 and 30,000 instances for the learning process.

## B    DETAIL SETTING OF MULTI-INSTRUCTION SFT AND MODEL ALIGNMENT.

We implement all model training using the PyTorch framework and utilize Qwen 1.5 with 14 billion parameters (Bai et al., 2023) as the base model. We utilize NVIDIA 8*A100 Tensor Core GPUs with 80 GB each for model training and employ 1*A100 GPU for model inference. The hyperparameters of the training are shown in Table 6. (Un-)DesirableWeight in the table represents the hyperparameters $\lambda_U$ and $\lambda_D$ of KTO in the paper.

Table 6: The detail training settings for SFT and KTO alignments.

| | LoRA_Dropout | LoRA_R | LoRA_Alpha | LearningRate | WarmUp Ratio | BatchSize | MaxLength | Epochs | (Un-)DesirableWeight | Beta |
|---|---|---|---|---|---|---|---|---|---|---|
| SFT | 0.05 | 64 | 16 | 3.00E-04 | 0.01 | 24 | 2048 | 20 | / | / |
| KTO | 0.05 | 16 | 16 | 5.00E-07 | 0.1 | 4 | 2048 | 20 | 1.0 | 0.1 |

For SFT, we adopt the code from the public Github repository at `https://github.com/QwenLM/Qwen`, and for KTO training, we adopt the code from the public Github repository at `https://github.com/huggingface/trl`. And our project code is available at `https://github.com/caigaojiang/LLMOPT`.

## C    DETAILED RESULTS

In this section, we present detailed results of comparing LLMOPT with the GPT-4 series models, including the Solving Accuracy (SA) shown in Table 7, the Execution Rate (ER) shown in Table 8, and the Average Solving Times (AST) shown in Table 9. All the results LLMOPT are based on Qwen1.5-14B. In all the tables, the values in **bold** represent the best performance achieved by LLMOPT, while the underlined values indicate the best performance among the GPT-4 series models. Across all six datasets, LLMOPT outperforms GPT-4 in both Solving Accuracy (SA) and Execution Rate (ER) metrics. Additionally, LLMOPT often achieves better performance than GPT-4 in terms of Average Solving Times (AST).

## D    COMPARISON BETWEEN QWEN1.5 AND QWEN2

In this section, we compare the performance of LLMOPT based on Qwen1.5-14B and Qwen2-72B (Yang et al., 2024a), focusing on two challenging tasks: Mamo Complex and IndustryOR. As illustrated in Figure 5, LLMOPT based on Qwen2-72B demonstrates superior performance in both tasks. Specifically, LLMOPT on Qwen2-72B outperforms Qwen1.5-14B across both the Solving Accuracy (SA) and Execution Rate (ER) metrics, highlighting the enhanced capabilities of the larger model in handling complex optimization problems.

## E    GENERALIZATION ON OTHER TASKS

In this section, we compare the model's performance before and after fine-tuning across 10 tasks: Math, Code, Classification, Extraction, Open QA, Closed QA, Text Generation, Brainstorming,

Table 7: Detailed results of solving accuracy (SA). **Bold** indicates LLMOPT's best performance, while underlined indicates the best results from GPT-4 models.

| | | | NL4Opt | Mamo Easy | Mamo Complex | IndustryOR | NLP4LP | ComplexOR |
|---|---|---|---|---|---|---|---|---|
| w/o debug | GPT-4-turbo | Directly | 47.0% | 65.0% | 13.0% | 26.0% | 13.5% | 9.1% |
| | | + 5-elem | 48.0% | 66.0% | 19.0% | 28.0% | 10.8% | 0.0% |
| | GPT-4o | Directly | 52.0% | 67.0% | 19.0% | 31.0% | 18.9% | 9.1% |
| | | + 5-elem | 62.0% | 67.0% | 23.0% | 27.0% | 16.2% | 18.2% |
| | LLMOPT | w/o five-element & KTO | 56.0% | 65.0% | 29.0% | 22.0% | 16.2% | 9.1% |
| | | w/o KTO | 52.0% | 63.0% | **33.0%** | 28.0% | 18.9% | 9.1% |
| | | w/o five-element | **60.0%** | **72.0%** | 31.0% | 27.0% | 16.2% | 9.1% |
| | | LLMOPT (w/o debug) | 57.0% | 71.0% | 32.0% | **31.0%** | **35.2%** | **18.2%** |
| with debug | GPT-4-turbo | + debug | 75.0% | 81.0% | 25.0% | 30.0% | 32.4% | 18.2% |
| | | + 5-elem + debug | 76.0% | 85.0% | 27.0% | 29.0% | 37.8% | 27.3% |
| | GPT-4o | + debug | 81.0% | 91.0% | 34.0% | 34.0% | 32.4% | 27.3% |
| | | + 5-elem + debug | 83.0% | 90.0% | 38.0% | 34.0% | 35.2% | 36.4% |
| | LLMOPT | w/o five-element & KTO | 88.0% | 89.0% | 46.0% | 33.0% | 54.1% | 45.5% |
| | | w/o KTO | 90.0% | 97.0% | 65.0% | 43.0% | 64.9% | 54.6% |
| | | w/o five-element | 91.0% | 96.0% | 64.0% | 41.0% | 73.0% | **72.7%** |
| | | **LLMOPT (Full)** | **93.0%** | **97.0%** | **68.0%** | **46.0%** | **83.8%** | **72.7%** |

Table 8: Detailed results of execution rate (ER). **Bold** indicates LLMOPT's best performance, while underlined indicates the best results from GPT-4 models.

| | | | NL4Opt | Mamo Easy | Mamo Complex | IndustryOR | NLP4LP | ComplexOR |
|---|---|---|---|---|---|---|---|---|
| w/o debug | GPT-4-turbo | Directly | 64.0% | 71.0% | 24.0% | 34.0% | 21.5% | 15.8% |
| | | + 5-elem | 57.0% | 68.0% | 23.0% | 33.0% | 20.0% | 15.8% |
| | GPT-4o | Directly | 71.0% | 71.0% | 42.0% | 40.0% | 35.4% | 42.1% |
| | | + 5-elem | 69.0% | 72.0% | 38.0% | 36.0% | 33.9% | 36.8% |
| | LLMOPT | w/o five-element & KTO | 74.0% | 74.0% | 41.0% | 39.0% | 33.9% | 36.8% |
| | | w/o KTO | 73.0% | 72.0% | 39.0% | 38.0% | 33.9% | 26.3% |
| | | w/o five-element | **82.0%** | **86.0%** | **53.0%** | **63.0%** | 38.5% | 42.1% |
| | | LLMOPT (w/o debug) | 79.0% | 85.0% | 52.0% | 55.0% | **47.7%** | **47.4%** |
| with debug | GPT-4-turbo | + debug | 95.0% | 99.0% | 95.0% | 87.0% | 81.5% | 79.0% |
| | | + 5-elem + debug | 93.0% | 98.0% | 91.0% | 83.0% | 78.5% | 79.0% |
| | GPT-4o | + debug | 99.0% | 99.0% | 96.0% | 83.0% | 86.2% | 89.5% |
| | | + 5-elem + debug | 96.0% | 100.0% | 96.0% | 82.0% | 84.6% | 84.2% |
| | LLMOPT | w/o five-element & KTO | **100.0%** | **100.0%** | 96.0% | 90.0% | 73.9% | 73.7% |
| | | w/o KTO | 99.0% | **100.0%** | 95.0% | **92.0%** | 89.2% | **100.0%** |
| | | w/o five-element | 99.0% | **100.0%** | **98.0%** | **92.0%** | 96.9% | 94.7% |
| | | **LLMOPT (Full)** | 99.0% | **100.0%** | 97.0% | **92.0%** | **100.0%** | 94.7% |

Table 9: Detailed results of average solving times (AST). **Bold** indicates LLMOPT's best performance, while underlined indicates the best results from GPT-4 models.

| | | NL4Opt | Mamo Easy | Mamo Complex | IndustryOR | NLP4LP | ComplexOR |
|---|---|---|---|---|---|---|---|
| GPT-4-turbo | + debug | 4.66 | 3.87 | 9.76 | 9.13 | 9.97 | 11.32 |
| | + 5-elem + debug | 4.75 | 3.59 | 9.81 | 9.47 | 10.08 | 11.32 |
| GPT-4o | + debug | 3.89 | 3.26 | 9.15 | 8.96 | 9.89 | 11.13 |
| | + 5-elem + debug | 3.93 | 3.61 | 9.06 | 8.97 | 10.40 | 10.95 |
| LLMOPT | w/o five-element & KTO | 3.58 | 3.45 | 8.11 | 9.25 | 8.68 | 10.79 |
| | w/o KTO | 3.47 | 2.80 | 6.79 | 8.88 | 8.32 | 10.84 |
| | w/o five-element | **2.95** | 2.36 | 6.38 | 8.50 | 7.43 | **10.05** |
| | **LLMOPT (Full)** | 3.07 | **1.70** | **6.30** | **8.35** | **7.00** | 10.27 |

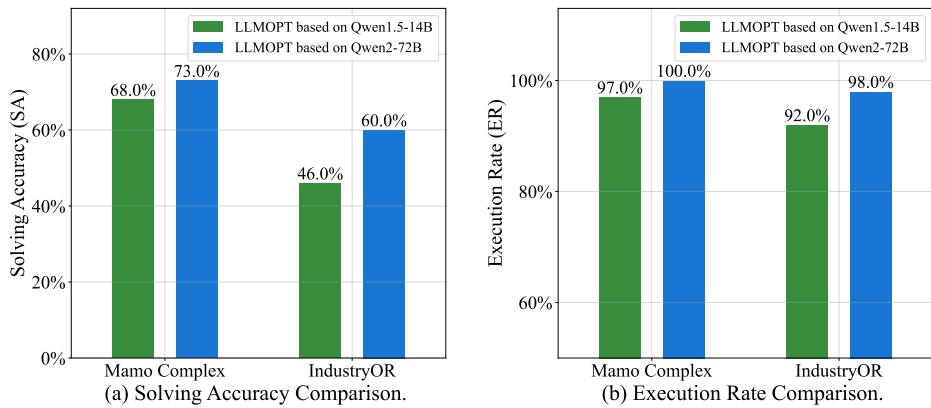

Figure 5: Comparison in (a) solving accuracy and (b) execution rate between Qwen1.5 and Qwen2.

Rewriting, and Summarization, as shown in Figure 6. The results indicate that LLMOPT does not significantly degrade performance on a broad range of tasks.

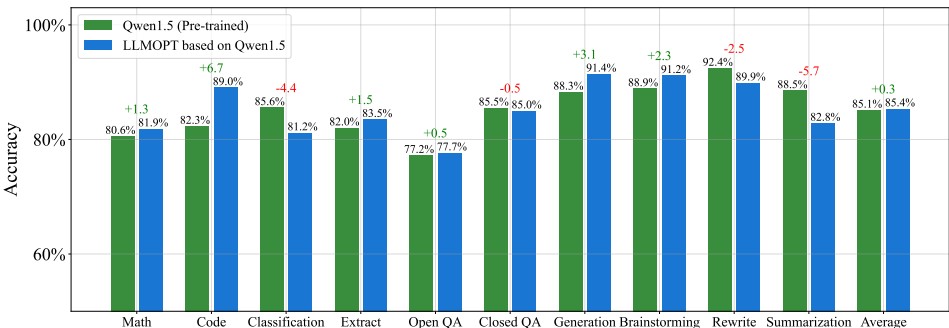

Figure 6: Results of performance evaluation on other task types.

## F  SCENARIOS OF DATASETS

The scenarios of the datasets are presented in Figure 7, covering 20 scenarios approximately.

| | NL4Opt | Mamo Easy | Mamo Complex | IndustryOR | NLP4LP | ComplexOR | Sum. |
|---|---|---|---|---|---|---|---|
| Agriculture | 23 | 30 | 5 | 6 | 3 | 1 | 68 |
| Energy | 5 | 33 | 7 | 1 | 4 | 0 | 50 |
| Health | 52 | 49 | 53 | 3 | 1 | 1 | 159 |
| Retail | 34 | 47 | 37 | 11 | 2 | 1 | 132 |
| Environment | 6 | 40 | 0 | 0 | 0 | 0 | 46 |
| Education | 11 | 32 | 0 | 3 | 1 | 0 | 47 |
| Financial Services | 5 | 46 | 2 | 6 | 4 | 0 | 63 |
| Transportation | 36 | 73 | 76 | 18 | 7 | 7 | 217 |
| Public Utilities | 6 | 29 | 11 | 0 | 2 | 0 | 48 |
| Manufacturing | 44 | 71 | 8 | 45 | 31 | 8 | 207 |
| Software | 2 | 0 | 10 | 1 | 7 | 1 | 21 |
| Construction | 4 | 56 | 1 | 1 | 0 | 0 | 62 |
| Legal | 0 | 0 | 0 | 0 | 2 | 0 | 2 |
| Customer Service | 2 | 2 | 0 | 0 | 0 | 0 | 4 |
| Entertainment | 10 | 44 | 0 | 0 | 0 | 0 | 54 |
| Others | 5 | 100 | 1 | 5 | 1 | 0 | 112 |
| # Data | 245 | 652 | 211 | 100 | 65 | 19 | 1292 |

Figure 7: Scenarios of the datasets.

## G  OPTIMIZATION TYPES OF DATASETS

The types of optimization problems are presented in Figure 8, categorized into 7 different classes.

| | NL4Opt | Mamo Easy | Mamo Complex | IndustryOR | NLP4LP | ComplexOR | Sum. |
|---|---|---|---|---|---|---|---|
| Linear Programming | 104 | 2 | 59 | 20 | 15 | 7 | 207 |
| Integer Programming | 105 | 238 | 12 | 11 | 3 | 1 | 370 |
| Mixed Integer Programming | 35 | 412 | 48 | 44 | 35 | 8 | 582 |
| Nonlinear Programming | 0 | 0 | 2 | 0 | 7 | 0 | 9 |
| Combinatorial Optimization | 0 | 0 | 65 | 9 | 5 | 3 | 82 |
| Multi-objective Programming | 0 | 0 | 0 | 8 | 0 | 0 | 8 |
| Others | 1 | | 25 | 8 | 0 | 0 | 34 |
| # Data | 245 | 652 | 211 | 100 | 65 | 19 | 1292 |

Figure 8: Optimization types of the datasets.

## H  TEMPLATES FOR INSTRUCTIONS ON LEARNING AND AUTO-TESTING

In this section, all instruction templates are introduced. Each template contains instruction constructed by fully filling in the curly braces "{·}" to specify the required content. The following templates are used both for constructing learning instructions and for prompts during auto-testing.

```
1  In mathematics, optimization problem can be modeled as the following
       expression $\min_{\boldsymbol{x} \in \mathcal{X}} f(\boldsymbol{x}),
       {\rm s.t.} G(\boldsymbol{x}) \leq \boldsymbol{c}$, where
       $\boldsymbol{x} = (x_1, x_2, \ldots, x_D)^\top$ is the
       $D$-dimensional decision variable, $\mathcal{X} \subset
       \mathbb{R}^D$ is the feasible domain, $f: \mathcal{X} \rightarrow
       \mathbb{R}$ is the objective function and the goal is to find the
       minima of $f$, $G(\boldsymbol{x}) \leq \boldsymbol{c}$ are the
       constraints of $\boldsymbol{x}$.
2
3  The above definition can be mapped to a five-element consisting of
       ``Variables, Objective, Constraints, Sets, Parameters''. Variables
       indicates what $\boldsymbol{x}$ is, Objective describes the form of
       the objective function $f(\boldsymbol{x})$, and Constraints
       indicates the constraints $G(\boldsymbol{x})$ and $\mathcal{X}$.
       These three can abstract the optimization problem. Sets and
       Parameters are their specific explanations: Sets describes and
       explains the subscripts of the vectors or matrices in them, and
       Parameters supplement their specific values.
4
5  You need to write the corresponding five-element model based on the
       problem description and information provided.
6
7  The problem description is as follows:
8  ```
9  {PROBLEM DESCRIPTION}
10 ```
11
12 Please write the corresponding five-element model. Please use LaTeX and
       ``` plain text environment to complete the following template to
       model the above optimization problem into five-element:
13
14 ```
15 ## Sets:
16 [You need to fill in]
17
18 ## Parameters:
19 [You need to fill in]
20
21 ## Variables:
22 [You need to fill in]
23
```

```
24  ## Objective:
25  [You need to fill in]
26
27  ## Constraints:
28  [You need to fill in]
29  ```
```

Listing 1: Instruction template of define the five-element formulation from the problem description.

```
1   The five-element model is the abstraction of an optimization problem,
        which transforms specific problem scenarios into formal mathematical
        problems. You need to write the corresponding Pyomo code based on
        the five-element model provided.
2
3   The following is the five-element model of an optimization problem:
4   ```
5   {FIVE-Element}
6   ```
7
8   Please write the corresponding Pyomo code. Please add 'from
        pyomo.environ import *' at the beginning of your code (You can add
        other 'import' as well). Please print the optimal solution and the
        value of the objective function. Please do not output the running
        log. You need to write it in the form of a class and add a main
        function:
9
10  ```python
11  [Write your code here]
12  ```
```

Listing 2: Instruction template of generate the solver code from the five-element formulation.

```
1   The five-element model is the abstraction of an optimization problem,
        which transforms specific problem scenarios into formal mathematical
        problems. You need to write the corresponding Pyomo code based on
        the five-element model provided.
2
3   The problem description is as follows:
4   ```
5   {PROBLEM DESCRIPTION}
6   ```
7
8   Please write the corresponding Pyomo code. Please add 'from
        pyomo.environ import *' at the beginning of your code (You can add
        other 'import' as well). Please print the optimal solution and the
        value of the objective function. Please do not output the running
        log. You need to write it in the form of a class and add a main
        function:
9
10  ```python
11  [Write your code here]
12  ```
```

Listing 3: Instruction template of generate the solver code from the problem description.

```
1   For the following optimization problem, modeling is performed, and pyomo
        code is generated and executed based on the modeling. Please judge
        whether the modeling and code are correct.
2   The problem is as follows.
3   ```
4   {PROBLEM DESCRIPTION}
5   ```
6
7   The five-element formulation is as follows.
```

```
8   ```
9   {FIVE-Element}
10  ```
11
12  The code is as follows.
13  ```
14  {SOLVER CODE}
15  ```
16
17  Run the code and get the following running information.
18  ```
19  {OUTPUT INFORMATIONS}
20  {ERROR INFORMATIONS}
21  ```
22
23  Please judge whether the above five-element and code are correct, and
        give your analysis according to the template below.
24
25  ```
26  The five-element is [Fill in True/False here].
27
28  The code is [Fill in True/False here].
29
30  Analysis:
31  [Fill in your analysis here]
32  ```
```

Listing 4: Instruction template of self-correction.

## I    TEMPLATES FOR DATA AUGMENTATION

In this section, a general template for data augmentation is introduced. The same as previous section, the curly braces "{·}" should be fully filled. The "ONE OF THE RULES BELOW" represents one of the rules of generating new questions, selected randomly from the ones introduced below.

```
1   Please generate an optimization problem according to the following
        requirements and the given format.
2
3   {ONE OF THE RULES BELOW}
4
5   The original optimization problem is as follows:
6   ```
7   {ORIGINAL OPTIMIZATION PROBLEM DESCRIPTION}
8   ```
9
10  Please construct a new optimization problem according to the above
        requirements and the provided questions and in the following format:
11  ```
12  [Write your new problem here]
13  ```
```

Listing 5: General template for data augmentation.

```
1   1. The following is an optimization problem. Please construct a new
        optimization problem based on the context of this problem.
2   2. The following is an optimization problem. Please find similar
        problems in other fields and construct a new optimization problem
        with a new scenario.
3   3. There are two optimization problems. Please construct a new
        optimization problem based on the scenario of problem A and the
        optimization problem type of problem B.
4   4. The following is an optimization problem. Please modify the
        constraints of this problem and construct a new optimization problem.
```

```
5  5. The following is an optimization problem. Please modify the
      constraints and object of this problem and construct a new
      optimization problem.
6  6. The following is an optimization problem. Please modify the variables
      and parameters of this problem reasonably and construct a new
      optimization problem.
7  7. The following is an optimization problem. Please modify the
      description of some statements and construct a new optimization
      problem without changing the meaning of the original problem.
```

Listing 6: Rules for data augmentation.

## J EXAMPLES OF FIVE-ELEMENT FORMULATION FOR DIFFERENT OPTIMIZATION PROBLEMS

This section presents examples of five-element formulations for different optimization problems, including seven examples spanning linear programming, integer programming, mixed-integer programming, nonlinear programming, and combinatorial optimization. By illustrating the five-element formulation across these diverse problem types, we aim to show the broad applicability of this modeling approach for general optimization problems.

---

**Linear Programming**

**Problem Statement:**

A person plans to invest \$100,000 in stocks and bonds over the next year to maximize the return on their investment portfolio. The expected annual return rate for stocks is 8%, while the annual return rate for bonds is 4%. At the same time, the risk coefficient for stocks is 0.08, and the risk coefficient for bonds is 0.02. The investor wants at least 60% of the funds to be invested in bonds, and the total risk of the entire portfolio cannot exceed 0.05. The investor needs to determine how to allocate this \$100,000 under the above conditions to achieve the maximum possible expected return.

---

**Five-Element Formulation:**

**Sets**
Set of investment methods: $\mathcal{S} = \{s, b\}$

**Parameters**
Annual return rate for stocks: $R_s = 0.08$
Annual return rate for bonds: $R_b = 0.04$
Risk coefficient for stocks: $C_s = 0.08$
Risk coefficient for bonds: $C_b = 0.02$
Total investment amount: $I = 100000$
Minimum proportion of bonds: $\alpha = 0.60$
Maximum total risk allowed: $C_{max} = 0.05$

**Variables**
Amount of money to be invested in stocks: $x_s$
Amount of money to be invested in bonds: $x_b$

**Objective**
Maximize the total expected return:

$$\max_{x_s, x_b} \quad R_s x_s + R_b x_b$$

**Constraints**
Total investment constraint:

$$x_s + x_b = I$$

Minimum investment in bonds constraint:

$$x_b \geq \alpha I$$

Total portfolio risk constraint:

$$C_s \frac{x_s}{I} + C_b \frac{x_b}{I} \leq C_{max}$$

Non-negativity constraint:

$$x_s \geq 0, \quad x_b \geq 0$$

## Integer Programming

**Problem Statement:**

An accounting firm employs part time workers and full time workers. Full time workers work 8 hours per shift while part time workers work 4 hours per shift. In addition, full time workers are paid $300 per shift while part time workers are paid $100 per shift. Currently, the accounting firm has a project requiring 450 hours of labor. If the firm has a budget of $15000, how many of each type of worker should be scheduled to minimize the total number of workers.

- - - - - - - - - - - - - - - - - - - - - - - - - - - - - - - - - - - - - - - - - - - - - - - - - - - -

**Five-Element Formulation:**

**Sets**
Set of employee types: $\mathcal{S} = \{f, p\}$

**Parameters**
Hours per shift: $h_f = 8, h_p = 4$
Wages per shift: $w_f = 300, w_p = 100$
Total labor time required: $T = 450$
Budget: $B = 15000$

**Variables**
Employees shifts: $x_f, x_p$

**Objective**
Minimize employee number: $\min_{x_f, x_p} \ x_f + x_p$

**Constraints**
Labor time constraint:
$$h_f x_f + h_p x_p \geq T$$

Budget constraint:
$$w_f x_f + w_p x_p \leq B$$

Positive integer constraint:
$$x_f, x_p \in \mathbb{Z}^+$$

## Integer Programming 2

**Problem Statement:**

A company intends to construct residential buildings on a new property and needs to complete the design and construction drawings. Influenced by various factors such as the market, the time required for design and construction exhibits cyclical peaks and troughs throughout the 12 months of the year. Specifically, the time required to complete the design drawings for each month is as follows: 3, 3, 4, 5, 6, 7, 6, 5, 4, 4, 3, 3 months; the time required for construction is as follows: 3, 4, 4, 5, 5, 6, 7, 8, 9, 7, 5, 4 months. What is the shortest time required for the company to go from the start of the design phase to the completion of construction?

- - - - - - - - - - - - - - - - - - - - - - - - - - - - - - - - - - - - - - - - - - - - - - - - - - - -

**Five-Element Formulation:**

**Sets**
Month Set:
$$M = \{1, 2, 3, 4, 5, 6, 7, 8, 9, 10, 11, 12\}$$

**Parameters**
Design time (per month) is $t_d(x)$:
$$t_d = [3, 3, 4, 5, 6, 7, 6, 5, 4, 4, 3, 3]$$

Construction time (per month) is $t_c(y \bmod 12)$:
$$t_c = [3, 4, 4, 5, 5, 6, 7, 8, 9, 7, 5, 4]$$

**Variables**
The month design starts: $x$
The month construction starts: $y$

**Objective**
Minimize the total time span:
$$\min_{x,y} \ \ y + t_c((y-1) \bmod 12 + 1) - x$$

**Constraints**
Order constraint:
$$y \geq x + t_d(x)$$

Value constraint:
$$x \in \{1, 2, \ldots, 12\}$$

Value constraint:
$$y \in \mathbb{Z}^+$$

## Mixed Integer Programming

**Problem Statement:**

MarketFlow Inc. must decide how to allocate resources for efficiently supplying six retail stores. They have identified four potential distribution centers, each with different opening costs and capabilities. The challenge is to choose the right mix of centers and optimal transportation routes to meet retail demands at the lowest total cost, which includes both opening expenses and transportation costs. Specifically, the supply capacity of each distribution center, measured in units, is as follows: Center 1 has a capacity of 1,631 units, Center 2 has 1,954 units, Center 3 has 1,446 units, and Center 4 has 820 units. The demand for each retail store, expressed in units, is as follows: Store 1 has a demand of 910 units, Store 2 has 875 units, Store 3 has 589 units, Store 4 has 962 units, Store 5 has 966 units, and Store 6 has 643 units. The opening costs for each distribution center are as follows: Center 1 costs $151,000, Center 2 costs $192,000, Center 3 costs $114,000, and Center 4 costs $171,000.

The transportation cost per unit from each distribution center to retail stores is as follows:

- From Center 1: $5 to Store 1, $5 to Store 2, $2 to Store 3, $3 to Store 4, $3 to Store 5, $3 to Store 6
- From Center 2: $5 to Store 1, $4 to Store 2, $3 to Store 3, $5 to Store 4, $2 to Store 5, $4 to Store 6
- From Center 3: $2 to Store 1, $4 to Store 2, $5 to Store 3, $1 to Store 4, $4 to Store 5, $2 to Store 6
- From Center 4: $5 to Store 1, $4 to Store 2, $1 to Store 3, $1 to Store 4, $3 to Store 5, $3 to Store 6

MarketFlow Inc. aims to efficiently meet demand at its six retail stores while minimizing costs related to opening distribution centers and transporting goods. This involves strategically allocating resources by selecting which centers to open and how much to transport to each store, all within supply constraints. What is the optimal total cost for MarketFlow Inc. to open the necessary centers and transport goods, ensuring demands are met at the lowest overall expense?

- - - - - - - - - - - - - - - - - - - - - - - - - - - - - - - - - - - - - - - - - - - - - - -

**Five-Element Formulation:**

**Sets**
Set of potential distribution centers: $I = \{1, 2, 3, 4\}$
Set of retail stores: $J = \{1, 2, 3, 4, 5, 6\}$

**Parameters**
Demand of each store: $\boldsymbol{d} = (910, 875, 589, 962, 966, 643)^\top$
Supply capacity of each center: $\boldsymbol{s} = (1631, 1954, 1446, 820)^\top$
Opening cost of each center: $\boldsymbol{c} = (151000, 192000, 114000, 171000)^\top$
Transportation cost per unit $t_{ij}$ from center $i$ to store $j$:

$$T_{|I| \times |J|} = \begin{bmatrix} 5 & 5 & 2 & 3 & 3 & 3 \\ 5 & 4 & 3 & 5 & 2 & 4 \\ 2 & 4 & 5 & 1 & 4 & 2 \\ 5 & 4 & 1 & 1 & 3 & 3 \end{bmatrix}$$

**Variables**
Amount of goods transported from distribution center $i$ to retail store $j$: $x_{ij}$
Binary variable indicating whether distribution center $i$ is open: $y_i$

**Objective**

$$\min \sum_{i \in I} c_i y_i + \sum_{i \in I} \sum_{j \in J} t_{ij} x_{ij}$$

**Constraints**
Demand fulfillment constraint:

$$\sum_{i \in I} x_{ij} = d_j, \quad \forall j \in J$$

Supply capacity constraint:

$$\sum_{j \in J} x_{ij} \leq s_i y_i, \quad \forall i \in I$$

Non-negativity constraint:

$$x_{ij} \geq 0, \quad \forall i \in I, \forall j \in J$$
$$y_i \in \{0, 1\}, \quad \forall i \in I$$

## Nonlinear Programming

**Problem Statement:**

The Chinese University of Hong Kong, Shenzhen decides to build a circular fountain on the campus. The school wants the fountain to be round and as large as possible but it must be restricted in a polygonal construction field, which is given by the following points: (0, 1), (0, 6), (4, 10), (8, 10), (11, 7), (11, 4), (7, 0), and (1, 0), the unit is m. Give a linear optimization formulation and find the maximal area. Keep your answer in four significant digit number.

- - - - - - - - - - - - - - - - - - - - - - - - - - - - - - - - - - - - - - - - - - - - - - - - - -

**Five-Element Formulation:**

**Sets**

Set of polygon vertices: $P = \{(x_i, y_i) \mid i = 1, \ldots, 8\}$

**Parameters**

Polygon vertices:

$$P_1 = (0, 1) \qquad P_2 = (0, 6) \qquad P_3 = (4, 10) \qquad P_4 = (8, 10)$$
$$P_5 = (11, 7) \qquad P_6 = (11, 4) \qquad P_7 = (7, 0) \qquad P_8 = (1, 0)$$

**Variables**

Center of the circular fountain: $(x_c, y_c)$
Radius of the circular fountain: $r$

**Objective**

$$\max_{x_c, y_c, r} \pi r^2$$

**Constraints**

Center inside polygon constraint:
For each edge defined by vertices $(x_i, y_i)$ and $(x_{i+1}, y_{i+1})$, ensure:

$$\frac{(y_{i+1} - y_i)x_c - (x_{i+1} - x_i)y_c + x_{i+1}y_i - y_{i+1}x_i}{\sqrt{(y_{i+1} - y_i)^2 + (x_{i+1} - x_i)^2}} \geq 0 \quad \text{for } i = 1, \ldots, 8$$

Distance to edges constraint:
The distance from the center $(x_c, y_c)$ to each edge must be at least the radius $r$:

$$\frac{|(y_{i+1} - y_i)x_c - (x_{i+1} - x_i)y_c + x_{i+1}y_i - y_{i+1}x_i|}{\sqrt{(y_{i+1} - y_i)^2 + (x_{i+1} - x_i)^2}} \geq r \quad \text{for } i = 1, \ldots, 8$$

Non-negativity constraint:
$$r \geq 0$$

---

### Combinatorial Optimization 1 (0-1 Knapsack Problem)

**Problem Statement:**

There are 4 items with weights of 4, 3, 1, and 1, and their values are 300, 200, 150, and 200 respectively, with only one of each item available. If we select items such that the total weight does not exceed 5, what is the maximum value that can be obtained?

- - - - - - - - - - - - - - - - - - - - - - - - - - - - - - - - - - - - - - - - - - - - - - - - - -

**Five-Element Formulation:**

**Sets**
Set of items: $I = \{1, 2, 3, 4\}$

**Parameters**
Weight of items: $\boldsymbol{w} = (4, 3, 1, 1)^\top$
Value of items: $\boldsymbol{v} = (300, 200, 150, 200)^\top$
Maximum allowable weight: $W = 5$

**Variables**
Binary variable indicator: $\boldsymbol{x}$ signifies whether item $i$ is selected ($x_i = 1$) or not ($x_i = 0$), for $i \in I$

**Objective**
Maximize the total value:

$$\max \sum_{i \in I} v_i x_i$$

**Constraints**
Weight constraint:

$$\sum_{i \in I} w_i x_i \leq W$$

Binary constraint:

$$x_i \in \{0, 1\} \quad \text{for all } i \in I$$

---

### Combinatorial Optimization 2 (Traveling Salesman Problem)

**Problem Statement:**

In a network consisting of four cities, namely A, B, C, and D, the distances between the cities are as follows: the distance from city A to city B is 10 units, the distance from city A to city C is 15 units, the distance from city A to city D is 20 units, the distance from city B to city C is 35 units, the distance from city B to city D is 25 units, and the distance from city C to city D is 30 units. All distances are symmetrical, meaning the distance from city $i$ to city $j$ is equal to the distance from city $j$ to city $i$. A travel route is defined as: starting from a certain city, visiting all cities exactly once, and ultimately returning to the starting city. Please find a travel route that meets the requirements and minimizes the total travel distance.

- - - - - - - - - - - - - - - - - - - - - - - - - - - - - - - - - - - - - - - - - - - - - - - - - -

**Five-Element Formulation:**

**Sets**
Set of cities: $I = \{A, B, C, D\}$.

**Parameters**
Distance between city $i$ and city $j$:

$$d_{ij} = \begin{cases} 10 & \text{if } \{i, j\} = \{A, B\} \\ 15 & \text{if } \{i, j\} = \{A, C\} \\ 20 & \text{if } \{i, j\} = \{A, D\} \\ 35 & \text{if } \{i, j\} = \{B, C\} \\ 25 & \text{if } \{i, j\} = \{B, D\} \\ 30 & \text{if } \{i, j\} = \{C, D\} \\ 0 & \text{if } i = j \end{cases}$$

**Variables**
Binary route indicator: $x_{ij}$, $\forall i, j \in I$
Visit order of city $i$: $u_i$, $\forall j \in I$

**Objective**
Minimize the total travel distance:

$$\min \sum_{i \in I} \sum_{j \in I, i \neq j} d_{ij} x_{ij}$$

**Constraints**
Visit and leave constraints:

$$\sum_{j \in I, i \neq j} x_{ij} = 1 \quad \forall i \in I$$

$$\sum_{i \in I, i \neq j} x_{ij} = 1 \quad \forall j \in I$$

Subtour elimination constraint:

$$u_i - u_j + |I| \cdot x_{ij} \leq |I| - 1, \forall i, j \in I, i \neq j$$

## K    Detailed Process of Data Labeling and Augmentation

In this section, we introduce the process of data labeling and augmentation, which is divided into four stages: preliminary review, expert labeling, expert review, and data aggregation.

1. **Preliminary review**. Initially review the data and remove unfeasible problems (unfeasible not means difficult). With the help of GPT-4o, we divide optimization problems into two categories according to their difficulty. Problems that meet one of the following conditions will be classified as difficult problems: at least 3 out of 5 solutions using GPT-4o are inconsistent, the code generated by GPT-4o has errors, and experts have found complex constraints, reasoning, or large amounts of data.

2. **Expert labeling**. For simple questions, 2 experts independently annotate. And 3 experts independently label for complex questions. For each question, five-element and solver code need to be labeled, and the code must be run without errors. In this stage, experts may use GPT-4o to generate text that meets the expert's intentions to reduce typing time and generate more formatted code. In order to improve data quality, experts may modify questions appropriately to make them more suitable for the problem scenario, or delete inappropriate questions (such as unfeasible ones).

3. **Expert review**. For each question, a new expert checks the labels of other experts in the previous step, which is based on the correctness of the problem modeling and the consistency of the labels of different experts (the same question may have different but correct labels from different experts). Highly controversial questions or those with incorrect labels are included in an independent challenging dataset.

4. **Data aggregation**. Five experts discuss and analyze each of the data in the independent difficult dataset to determine whether the problem has a solution or can be adjusted to a feasible problem, and decide whether to abandon the problem based on this. If not, the experts will discuss and complete the labeling of these data. The correctly labeled questions that have passed the expert review will be summarized. And the other feasible questions that have been incorrectly labeled during the entire labeling process will be summarized.

The above steps are completed by 12 experts. The "preliminary review" and "expert labeling" are finished by 9 undergraduate students with bachelor's degrees or above (computer science or mathematics, all of whom have taken optimization courses), including 4 master's students in related fields (1 doctoral student). The "expert review" is finished by 1 university professor whose research is optimization in machine learning and 2 algorithm engineers working on operations research optimization. "Data aggregation" is finished by the experts except undergraduates. During the data labeling process, we ensure that each expert completed the labeling independently. In the first three stages, a question would not be assigned to the same expert twice.

The review pass rate for the seven experts assigned to simple question labeling exceeds 90%, while the pass rate for the four experts labeling complex question labeling is above 80% (2 experts labeled both simple and complex questions). Considering the above statistics, along with the fact that the labeling results will undergo review by senior experts in the third stage, the reliability and effectiveness of the expert labeling process are well supported.

## L    More Experiments for Generalization Performance

In order to demonstrate the generality of LLMOPT, we re-analyzed all experimental results and classified the types of optimization problems involved in each dataset, including Linear Programming (LP), Integer Programming (IP), Mixed Integer Programming (MIP), Nonlinear Programming (NP), Combinatorial Optimization (CO), Multi-objective Programming (MOP) and Others. The following table shows the SA performance of LLMOPT in solving various types of problems on different datasets. The data are the solving accuracy (SA), and the values in brackets represent "number of problems solved correctly/number of problems in the dataset". Detailed statistics are shown in Table 10. The results show that LLMOPT is universal in various types of optimization problems and can solve almost all kinds of optimization problems. However, due to the different distribution of problem formulating difficulties, the accuracy of LLMOPT on these datasets is also different, which will be one of the focuses of future work.

Table 10: SA of LLMOPT (based on Qwen-1.5-14B) organized by the type of optimization problems. The values in brackets represent "number of problems solved correctly/number of problems in the dataset".

| | NL4Opt | MamoEasy | MamoComplex | IndustryOR | NLP4LP | ComplexOR |
|---|---|---|---|---|---|---|
| **LP** | 90.5% (38/42) | - | 76.0% (19/25) | 60.0% (12/20) | 77.8% (7/9) | 60.0% (3/5) |
| **IP** | 95.8% (46/48) | 100.0% (36/36) | 100.0% (5/5) | 45.5% (5/11) | 0.0% (0/1) | - |
| **MIP** | 90.0% (9/10) | 95.3% (61/64) | 73.9% (17/23) | 47.7% (21/44) | 95.2% (20/21) | 80.0% (4/5) |
| **NP** | - | - | 100.0% (1/1) | - | 100.0% (3/3) | - |
| **CO** | - | - | 60.0% (21/35) | 33.3% (3/9) | 33.3% (1/3) | 100.0% (1/1) |
| **MOP** | - | - | - | 37.5% (3/8) | - | - |
| **Others** | - | - | 50.0% (5/10) | 25.0% (2/8) | - | - |
| **Total** | **93.0% (93/100)** | **97.0% (97/100)** | **68.0% (68/100)** | **46.0% (46/100)** | **83.8% (31/37)** | **72.7% (8/11)** |

Table 11: Performance of Different Inference Models and Correction Mechanisms

| Inference Model | Correction Mechanism | IndustryOR Dataset | NL4Opt Dataset |
|---|---|---|---|
| LLMOPT (Qwen-1.5) | Self-correction | **46.0%** | **93.0%** |
| LLMOPT (Qwen-1.5) | Correction by GPT-4o | 41.0% | 89.0% |
| LLMOPT (Qwen-1.5) | Best of 12 repeats | 42.0% | 89.0% |
| GPT-4o OpenAI (2023) | Correction by GPT-4o | 34.0% | 84.0% |
| GPT-4o OpenAI (2023) | Best of 12 repeats | 32.0% | 84.0% |
| ORLM Tang et al. (2024) | Best of 12 repeats | 39.0% | 88.0% |

## M  MORE ABLATION EXPERIMENTS FOR CORRECTION MECHANISM

To further explore the superiority of self-correction and LLMOPT, we deploy experiments on the NL4Opt and IndustryOR datasets (NL4Opt has relatively simple problems, while IndustryOR has relatively complex problems). We change two correction mechanisms, one is correction by GPT-4o with the same prompt of self-correction, and the other is to repeat the inference 12 times and manually judge the optimal solution (which means that only one optimal solution needs to be found in 12 repeated experiments). The reason we chose 12 is that self-correction is limited to a maximum of 12 repeated checks, so this is fair. We also conduct experiments on GPT-4o and ORLM Tang et al. (2024). We reproduce the open source model of ORLM Tang et al. (2024), but found that this model seems to have lost other abilities except writing coptpy code for optimization problems. We find that ORLM has a serious seesaw problem, which is performance as without generalization ability, and cannot answer other questions. Therefore, only the "Best of 12 repeats" correction mechanism is experimented. The results are shown in Table 11. The results show that, when LLMOPT (based on Qwen-1.5) is used as the inference model, the correction performance of GPT-4 is lower than the self-correction solving accuracy of LLMOPT. This indicates that the Qwen-1.5 model learned by LLMOPT shows stronger overall capabilities in both solving optimization problems and correction compared to other methods. Although manually selecting the best result from 12 repetitions shows performance improvement (considering once solving correct if one out of 12 repetitions is accurate), it still falls short of the effectiveness compared with the self-correction mechanism. This highlights that identifying and correcting errors is more critical than simply repeating executions, emphasizing the necessity of implementing a correction mechanism.

