# OpenReview forum: "LLMOPT: Learning to Define and Solve General Optimization Problems from Scratch"
_ICLR.cc/2025/Conference — ICLR 2025 Poster_

### Official Review · Reviewer_feJ7 · 2024-10-20

**Soundness:** 3
**Presentation:** 3
**Contribution:** 2
**Rating:** 5
**Confidence:** 3

**Summary:**

This paper proposes a learning-based LLM for solving Optimization Problems. The experimental results demonstrate the effectiveness of the proposed LLMOPT compared with prompt-based methods.

**Strengths:**

1. The introduction of this manuscript is clear.

2. Some improvement over baseline results.

**Weaknesses:**

1. It is questionable whether LLMs are capable of solving large-scale optimization problems. If their applicability is limited to small-scale problems, such as the Traveling Salesman Problem with just 10 nodes, the significance of this method is unclear. A problem of this size can even be solved by the Hopfield network proposed in the 1980s. In fact, both traditional heuristics and more recent neural combinatorial optimization methods are capable of handling problems with thousands of nodes [1, 2].

2. In Abstract, the authors mention, "to prevent hallucinations in LLMs, such as sacrificing solving accuracy to avoid execution errors." However, I do not believe this phenomenon should be described as hallucination. Hallucination typically refers to instances where the output of an LLM appears reasonable but is, in fact, fabricated. In this case, reducing solving accuracy to prevent execution errors does not align with the conventional meaning of hallucination. Please provide a more precise explanation or reconsider this terminology.

3. As shown in Table 3, the self-correction mechanism plays a decisive role in the performance of LLMOPT. However, this mechanism is not originally proposed in this manuscript. As the authors describe on Page 6, "Inspired by Chen et al. (2024), to enhance optimization generalization, we implement self-correction to automatically analyze the output results and identify errors arising during the execution of the solver code." Furthermore, the prior study [3] has also employed this mechanism to improve the performance of LLMs in solving vehicle routing problems.

4. In Section 4, the authors do not report the training time of LLMOPT, making it unfair to directly compare it with prompt-based methods.

5. Please provide the code generated by the LLM to solve the optimization problem. I am doubtful that current LLMs are capable of generating code sophisticated enough to solve medium- or large-scale problems.

[1] Fu Luo, et al. Neural combinatorial optimization with heavy decoder: Toward large scale generalization. In Proceedings of the Advances in Neural Information Processing Systems, 2023.

[2] Huigen Ye, et al. GNN&GBDT-guided fast optimizing framework for large-scale integer programming. In Proceedings of International Conference on Machine Learning, pp. 39864–39878. 2023.

[3] Zhehui Huang, et al. Can Large Language Models Solve Robot Routing?. arXiv, 2024.

**Questions:**

Please refer to the weakness section.

---

> ### Author Response · Authors · 2024-11-21
> **Reply to Reviewer feJ7 (1/8)**
>
> Thank you for your comprehensive review and insightful feedback. We have carefully considered your feedback and provided detailed responses below. If there are any other questions, please feel free to ask, and we will respond promptly.
>
> ### Response to your concerns in Question 1: Capabilities of LLM and implementation of LLMOPT.
>
> Thank you for your question! We carefully checked the two articles [1, 2] mentioned in your review, which focus on efficiently and accurately solving specific optimization problems. We address your concerns from two aspects:
>
> 1. **LLMOPT is not an LLM-based solver or solving method, so LLMOPT is different from solving methods like [1, 2].** In real-world tasks, data is not pre-processed and requires human experts to extract data and model optimization problems from the natural language descriptions. Taking the Traveling Salesman Problem as an example, node and path information is not directly provided; experts need to manually extract and identify the problem type, then transform it into a specific data structure (such as a set of nodes and edge weights) to apply solving methods. In LLMOPT, the role of the LLM is not to solve the problem but to replace experts in completing the modeling task and orchestrating the entire solving process. **Specifically, it extracts key information from the natural language description of the optimization problem, identifies the problem type, models the problem, and subsequently automatically invokes the specified solver or solving method**. LLMOPT eliminates the need for expert modeling, achieving full automation from the original problem description to the solution. Therefore, LLMOPT is fundamentally different from solving methods like [1, 2].
> 2. **LLMs understand the description of optimization problems and invoke solvers rather than directly solving optimization problems.** The ability of LLMs to solve large-scale optimization problems remains an open question, as it involves multiple fields such as mathematical reasoning, computation, and advanced algorithms. **Therefore, we focus on correctly modeling large-scale optimization problems and generating code for solving them, rather than directly using LLMs to solve the problems.** We believe the strength of LLMs lies in understanding natural language and generating code, which means LLMs excel at transforming natural language descriptions of original problems into mathematical models and code, rather than directly solving them. Thus, we designed LLMOPT, leveraging LLMs to model problems into a general _five-element_ formulation, select appropriate solvers, and generate solving code. The actual solving process is delegated to specialized, powerful solvers to ensure the accuracy and efficiency of the solutions.
>
> We explained the motivation of LLMOPT from two aspects, aiming to clarify the rationale and approach of LLMOPT. Regarding your question about whether LLMs have the capability to solve large-scale optimization problems, **under the LLMOPT framework, this depends on the modeling capability of the LLM and the solving capability of the specific solver used.** In the experiments of this paper, we utilized three open-source solvers—GLPK [3], IPOPT [4], and SCIP [5]—via Pyomo code, where SCIP is capable of handling problems with hundreds of thousands of variables and constraints. Theoretically, as long as an LLM can correctly model the problem based on its description and generate Pyomo solving code, these large-scale problems can be correctly solved. Therefore, the goal of LLMOPT is to enable the LLM to learn how to properly define problems and generate solving code to automatically invoke these powerful solvers and solution methods.

---

> ### Author Response · Authors · 2024-11-21
> **Reply to Reviewer feJ7 (2/8)**
>
> ### Response to Question 2: Explanation of hallucination.
>
> Thank you for your question! Your understanding of hallucinations is correct. Hallucinations typically refer to situations where an LLM generates outputs that appear reasonable but are actually fabricated, such as citing non-existent references, calling non-existent methods when writing code, or inferring false facts.
>
> In LLMOPT, the LLM aims to generate solver code that can correctly formulate and solve the optimization problem described in natural language. During this process, we have observed several types of hallucination issues. For example:
>
> 1. When solving a knapsack problem, the task describes a 0-1 knapsack problem. However, the LLM inexplicably assumes, "_We assume all items are infinite_" and writes code based on this incorrect assumption. This behavior of **erroneously attaching unrelated assumptions based on prior knowledge** is a typical example of hallucination.
> 2. When solving a Traveling Salesman Problem (TSP), the LLM incorrectly introduces an assumption during its reasoning process, stating, "_We can take A as the starting point and G as the endpoint, and model the problem accordingly_." While this approach simplifies the generation of the solver code, it **arbitrarily adds conditions that do not align with the original problem description**. As a result, the generated solver code fails to solve the original problem. This behavior is another example of hallucination.
> 3. Even in simple problems, LLMs can exhibit hallucinations. For example, when generating solver code, the LLM directly uses `>` or `<` to represent strict inequality constraints. Although Python supports these symbols, most solvers do not support strict inequality modeling and require such constraints to be converted into non-strict inequalities by adding a small positive value. This **inappropriate analogy and subjective inference** are typical examples of hallucination.
>
> **Because hallucinations do exist, we designed model alignment to address them.** Thank you for your constructive questions and concerns! We will revise the expressions of the article and provide more accurate explanations based on your suggestions.

---

> ### Author Response · Authors · 2024-11-21
> **Reply to Reviewer feJ7 (3/8)**
>
> ### Response to Question 3: About self-correction.
>
> Thank you for your questions and concerns! We carefully reviewed the results and performed a more detailed analysis of self-correction. **The self-correction mechanism does not play a decisive role. And although the correction mechanism is widely used, the ability of correction is brought by the learning pipeline of LLMOPT.** We analyze the role of the self-correction mechanism through the following experiments.
>
> ### Experiment 1: Fair comparison of self-correction.
>
> **In Table 3 of the paper, it is** _**unfair**_ **to directly compare the full LLMOPT with LLMOPT w/o self-correction.** Taking the IndustryOR dataset in the table as an example, the average solving times (AST in the table) of the full LLMOPT is 8.35, while the AST of w/o self-debug is 1.00. This means that w/o self-debug only performs 1 inference, which is unfair to compare with the full LLMOPT which performs 8.35 inferences on average.
>
> **A fair self-correction ablation experiment is as follows.** To ensure a fair comparison on Average Solving Times (AST), we use LLM to repeatedly solve the problem 12 times and manually selected the best solution among these repeated reasonings (which means that only one optimal solution needs to be found in 12 repeated experiments). The reason we chose 12 is that self-correction is limited to a maximum of 12 repeated checks, so this is fair. The experimental results on LLMOPT and GPT-4o are as follows:
>
> | **Inference Model** | **Correction Mechanism** | **NL4Opt** | **IndustryOR** | **Mamo_E** | **Mamo_C** |
> | :---: | :---: | :---: | :---: | :---: | :---: |
> | LLMOPT (Qwen-1.5) | Self-correction | **93.0%** | **46.0%** | **97.0%** | **68.0%** |
> | LLMOPT (Qwen-1.5) | Best of 12 repeats | 89.0% | 42.0% | 94.0% | 65.0% |
> | GPT-4o | Self-correction | 84.0% | 34.0% | 90.0% | 38.0% |
> | GPT-4o | Best of 12 repeats | 84.0% | 32.0% | 89.0% | 35.0% |
>
>
> From the results in the table, it can be observed that the self-correction mechanism has a clear advantage compared to the best of 12 repeats, indicating that the self-correction mechanism is effective for both LLMOPT and GPT-4o. However, **regardless of whether GPT-4o employs self-correction, its performance is consistently worse than that of LLMOPT, suggesting that self-correction is not the decisive factor influencing the results. The ability to correctly solve problems and perform corrections depends on the model’s inherent capabilities**. Regardless of the correction mechanism used, the superiority of LLMOPT over GPT-4o is evident, demonstrating that the learning pipeline of LLMOPT is an effective method for enhancing the correction capabilities of LLMs. In the following Experiment 2, we will design more detailed experiments to further elaborate on these findings.

---

> ### Author Response · Authors · 2024-11-21
> **Reply to Reviewer feJ7 (4/8)**
>
> ### Experiment 2: The success of correction comes from LLMOPT.
>
> Thank you for your interest in self-correction! In fact, **multi-instruction SFT and KTO alignment are the basis of self-correction. They not only improve the modeling and solving capabilities of LLMOPT, but also bring correction capabilities. If SFT and KTO are absent, high-quality correction cannot be achieved.**
>
> To further analyze this issue, we conducted additional experiments. We compared the following methods:
>
> 1. LLMOPT. Full LLMOPT with self-correction during testing. Same as in the paper.
> 2. LLMOPT w/o KTO. LLMOPT with multi-instruction SFT but no model alignment during learning, with self-correction during testing.
> 3. LLMOPT corrected by GPT-4o. Inference is performed using the full learned model, and correction is performed using GPT-4o and the same prompt.
> 4. LLMOPT w/o self-correction (best of 12 repeats). Inference is repeated 12 times using the full learned model, and the best solution among the 12 solutions is manually selected as the final solution (which means that only one optimal solution is found in 12 repeated experiments). The reason we choose 12 is that self-correction is limited to a maximum of 12 repeated checks, so this is fair.
> 5. ORLM (best of 12 repeats). In addition, we reproduce ORLM [3] and add a correction mechanism. Since the ORLM model has lost its generalization ability, ORLM is used to repeat the reasoning 12 times and manually select the best solution among the 12 solutions to simulate the correction process.
>
> The experimental results of Solving Accuracy (SA) are as follows:
>
> | | **NL4Opt** | **IndustryOR** | **Mamo_E** | **Mamo_C** |
> | --- | :---: | :---: | :---: | :---: |
> | **LLMOPT** | **93.0%** | **46.0%** | **97.0%** | **68.0%** |
> | LLMOPT w/o KTO | 90.0% | 43.0% | **97.0%** | 65.0% |
> | LLMOPT corrected by GPT-4o | 89.0% | 41.0% | 95.0% | 66.0% |
> | LLMOPT w/o self-correction (best of 12 repeats) | 89.0% | 42.0% | 94.0% | 65.0% |
> | ORLM (best of 12 repeats) | 88.0% | 39.0% | 87.0% | 46.0% |
>
>
> From the results, it can be seen that, despite using the same prompt, the performance of LLMOPT w/o KTO is worse than that of the original LLMOPT across all three tasks. This demonstrates that KTO not only improves the ability to formulate optimization problems but also enhances the self-correction capabilities of the LLM. Furthermore, the correction method utilizing GPT-4o performs worse than LLMOPT, indicating that the learning processes through multi-instruction SFT and KTO also improves the LLM’s correction ability. **Overall, the LLMOPT pipeline enhances the model’s comprehensive ability to handle optimization problems.** The experimental results for the reproduced ORLM (best of 12 repeats) show that LLMs fine-tuned using LLMOPT exhibit significantly stronger correction ability compared to those fine-tuned using ORLM.

---

> ### Author Response · Authors · 2024-11-21
> **Reply to Reviewer feJ7 (5/8)**
>
> ### Response to Question 4: Training and testing details of LLMOPT.
>
> Thank you for your concerns! **In our submission, we have already provided the necessary details, including the hardware used for training, as well as the detailed parameters of SFT and KTO, in Appendix B.** Below, we provide a detailed explanation of the training time and computational requirements.
>
> During the training phase, LLMOPT uses Qwen-1.5-14B as the base model, and the FLOPS calculation includes both training and inference. For those prompt-based methods, the FLOPS calculation only accounts for the inference stage. OptiMUS [7] and Chain-of-Expert [8] both utilize GPT-4 as their inference engine. Based on publicly available information [9], we can roughly estimate that GPT-4 comprises approximately 16*110B model parameters. Here is the detailed calculation of FLOPS.
>
> 1. **Single-step training FLOPS of LLMOPT.** The calculation for single-step training FLOPS in LLMOPT includes three components: forward FLOPS, backward FLOPS, and optimizer FLOPS. Forward FLOPS, comprising the embedding layer and transformer computations, are approximately $20.68 \times 10^{12}$. Backward FLOPS are twice the forward FLOPS, resulting in approximately $41.36 \times 10^{12}$. Optimizer FLOPS are calculated as $12 \times 14 \times 10^9 = 168 \times 10^9 = 0.168 \times 10^{12}$. Combining these, the total single-step training FLOPS is approximately $20.68 \times 10^{12} + 41.36 \times 10^{12} + 0.168 \times 10^{12} = 62.21 \times 10^{12}$. However, since fine-tuning uses **LoRA**, the actual single-step training FLOPS should be close to the forward FLOPS, approximately $20.68 \times 10^{12}$.
> 2. **SFT and KTO FLOPS of LLMOPT.** Both SFT and KTO use LoRA for training, resulting in similar FLOPS requirements. The batch sizes for SFT and KTO are 24 and 4, respectively, with 3,000 and 30,000 training steps, and **training durations of approximately 26 hours and 72 hours**. Consequently, the total training FLOPS for LLMOPT can be calculated as $5.76 \times 10^5 \times 20.68 \times 10^{12} + 9.6 \times 10^5 \times 20.68 \times 10^{12} = 3.17 \times 10^{19}$.
> 3. **Single inference FLOPS in LLMOPT.** The approximate FLOPS for a single inference in LLMOPT is $20.68 \times 10^{12}$.
> 4. **Single inference FLOPS in those prompt-based methods.** For prompt-based methods like OptiMUS [7] and Chain-of-Expert [8], we focus solely on inference costs, disregarding training costs. Assuming that a single expert is activated during the inference stage, the approximate FLOPS required for a single inference with GPT-4 is $3.88 \times 10^{15}$.
>
> In summary, during the training of LLMOPT, SFT and KTO require approximately 26 hours and 72 hours, respectively. When the number of calls reaches 9,437, the FLOPS of LLMOPT will be lower than other prompt-based methods using GPT4, which shows our cost advantage and huge application potential when calling on a large scale usage.
>
>
> ### Response to Question 5: Examples of code generated by LLMOPT.
>
> As mentioned in the response to Question 1, LLMOPT aims to enable LLMs to better model optimization problems and generate correct solver codes. **There is no doubt that LLM is capable of generating code to solve large-scale complex problems.** Here we provide the code generated by the LLM to solve the optimization problem.

---

> ### Author Response · Authors · 2024-11-21
> **Reply to Reviewer feJ7 (6/8)**
>
> ### Example 1:
>
> Based on LLMOPT, we developed an industrial copilot product (name omitted for anonymity), which has been widely applied to real-world scenarios such as financial lending, road planning, and travel optimization, in conjunction with our self-developed large-scale industrial solver (name omitted for anonymity).
>
> The following is an example of large-scale application in the context of car insurance leads distribution. The problem involves setting upper and lower limits on allocations to ensure balanced distribution of users among insurance companies, optimizing callback effectiveness, and safeguarding the interests of partners.
>
> **Code generated by LLMOPT:**
>
> ```python
> import pandas as pd
> import numpy as np
> import pyomo.environ as pyomo_env
> from Anonymous.Python.Package import PyomoDistributedModelBuilder
>
> # Custom model builder class extending PyomoDistributedModelBuilder
> class myModelBuilder(PyomoDistributedModelBuilder):
>     def __init__(self, input_conf, output_conf, model_conf, data_conf, solver_conf, shard_id, shard_count):
>         # Initialize the parent class
>         super().__init__(input_conf, output_conf, model_conf, data_conf, solver_conf, shard_id, shard_count)
>
>     def build_model(self):
>         # Build the optimization model by assembling datasets, variables, objectives, and constraints
>         model = self.model
>         model = self.read_dataset(model)
>         model = self._attach_variables(model)
>         model = self._attach_objective(model)
>         model = self._attach_local_constraints(model)
>         model = self._attach_global_constraints(model)
>         self.model = model
>         return model
>
>     def read_dataset(self, model):
>         # Load data: define sets and parameters
>         model.I = Set(initialize=self.parse_sets(input_conf))  # Set of leads
>         model.J = Set(initialize=self.parse_sets(input_conf))  # Set of communities
>         model.p = pyomo_env.Param(model.I, model.J, initialize=self.parse_params(input_conf))  # Weights
>         model.u = pyomo_env.Param(model.J, initialize=self.parse_params(input_conf))          # Upper bounds
>         model.l = pyomo_env.Param(model.J, initialize=self.parse_params(input_conf))          # Lower bounds
>         return model
>
>     def _attach_variables(self, model):
>         # Decision variable: `x[i, j]` represents assignment from lead `i` to community `j`
>         model.x = pyomo_env.Var(model.I, model.J, within=pyomo_env.NonNegativeReals, bounds=(0, 1))
>         return model
>
>     def _attach_objective(self, model):
>         # Maximize the weighted sum of assignments
>         obj_expr = sum(model.p[i, j] * model.x[i, j] for i in model.I for j in model.J)
>         model.obj = pyomo_env.Objective(expr=obj_expr, sense=pyomo_env.maximize)
>         return model
>
>     def _attach_local_constraints(self, model):
>         # Each lead must be assigned to exactly one community
>         def _attach_leads_sum_n(model, i):
>             return sum(model.x[i, j] for j in model.Com) == 1
>         model.x_sum = pyomo_env.Constraint(model.Leads, rule=_attach_leads_sum_n)
>         return model
>
>     def _attach_global_constraints(self, model):
>         # Ensure total assignments to each community meet upper and lower bounds
>         for j in model.Com:
>             total_com = sum(model.x[i, j] for i in model.Leads)
>             self.add_global_ineq_constraint(model, self._global_ineq_constr_num,
>                                             pyomo_env.Constraint(expr=total_com <= model.u[j]))
>             self._global_ineq_constr_num += 1
>
>             self.add_global_ineq_constraint(model, self._global_ineq_constr_num,
>                                             pyomo_env.Constraint(expr=total_com >= model.l[j]))
>             self._global_ineq_constr_num += 1
>         return model
> ```

---

> ### Author Response · Authors · 2024-11-21
> **Reply to Reviewer feJ7 (7/8)**
>
> ### Example 2:
>
> The problem is as follows.
>
> ```
> A machine can operate continuously for 4 years or can be sold at the end of each year and replaced with a new one. The purchase price of a new machine at the beginning of each year and the resale price of machines of different ages at the end of each year are listed in Table 4.2 (unit: 10,000 yuan). Additionally, the operation and maintenance costs for a new machine in its first year are 0.3 million yuan, while the annual operation and maintenance costs for machines in their second, third, and fourth years of usage are 0.8 million yuan, 1.5 million yuan, and 2.0 million yuan, respectively. Determine the optimal replacement strategy for the machine to minimize the total costs incurred for replacement, purchase, and operation/maintenance over 4 years.
> \begin{table}[h]
> \centering
>     \begin{tabular}{|c|c|c|} \hline
>         Year & Purchase Price at Beginning of Year (10,000 yuan) & Resale Price of Machine Used for j Years (10,000 yuan) \\ \hline
>         Year 1 & 2.5 & 2.0 \\ \hline
>         Year 2 & 2.6 & 1.6 \\ \hline
>         Year 3 & 2.8 & 1.3 \\ \hline
>         Year 4 & 3.1 & 1.1 \\ \hline
>     \end{tabular}
> \caption{Table 4.2: Machine Purchase Price and Resale Price }
> \end{table}
> ```
>
> **Code generated by LLMOPT:**
>
> ```python
> # Import Pyomo and solver
> from pyomo.environ import *
> from pyomo.opt import SolverFactory
>
> # Create a Pyomo model
> model = ConcreteModel()
>
> # Define sets of nodes
> model.N = RangeSet(1, 5)
>
> # Define arcs and their costs based on the given W matrix
> arcs = [
>     (1, 2), (1, 3), (1, 4), (1, 5),
>     (2, 3), (2, 4), (2, 5),
>     (3, 4), (3, 5),
>     (4, 5)
> ]
>
> # Define the costs for each arc (from the W matrix)
> w = {
>     (1, 2): 0.8, (1, 3): 2.0, (1, 4): 3.8, (1, 5): 6.0,
>     (2, 3): 0.9, (2, 4): 2.1, (2, 5): 3.9,
>     (3, 4): 1.1, (3, 5): 2.3,
>     (4, 5): 1.4
> }
>
> # Define the set of arcs
> model.A = Set(initialize=arcs, dimen=2)
>
> # Define variables x[i,j] = 1 if arc (i,j) is selected, 0 otherwise
> model.x = Var(model.A, within=Binary)
>
> # Define the objective function: minimize total cost
> def obj_rule(model):
>     return sum(w[i, j] * model.x[i, j] for (i, j) in model.A)
> model.obj = Objective(rule=obj_rule, sense=minimize)
>
> # Flow conservation constraints
> def flow_rule(model, k):
>     # For node 1 (source)
>     if k == 1:
>         return sum(model.x[1, j] for j in model.N if (1, j) in model.A) == 1
>     # For node 5 (sink)
>     elif k == 5:
>         return sum(model.x[i, 5] for i in model.N if (i, 5) in model.A) == 1
>     # For intermediate nodes
>     else:
>         return sum(model.x[i, k] for i in model.N if (i, k) in model.A) == sum(model.x[k, j] for j in model.N if (k, j) in model.A)
> model.flow = Constraint(model.N, rule=flow_rule)
>
> # Solve the model using a solver
> solver = SolverFactory('glpk')
> solver.solve(model)
>
> # Print the results
> print("Optimal Replacement Strategy:")
> for (i, j) in model.A:
>     if value(model.x[i, j]) > 0.5:
>         print("Replace machine at year {} and use until year {}".format(i, j-1))
>
> print("Optimal Total Cost: {:.2f} million yuan".format(value(model.obj)))
> ```

---

> ### Author Response · Authors · 2024-11-21
> **Reply to Reviewer feJ7 (8/8)**
>
> We hope that our response has addressed your concerns, but if we missed anything please let us know.
>
> **References**:
>
> [1] Fu Luo, et al. Neural combinatorial optimization with heavy decoder: Toward large scale generalization. In Proceedings of the Advances in Neural Information Processing Systems, 2023.
>
> [2] Huigen Ye, et al. GNN&GBDT-guided fast optimizing framework for large-scale integer programming. In Proceedings of International Conference on Machine Learning, pp. 39864–39878. 2023.
>
> [3] GLPK (GNU Linear Programming Kit) solver. [https://www.gnu.org/software/glpk/](https://www.gnu.org/software/glpk/)
>
> [4] IPOPT (Interior Point OPTimizer) solver. [https://coin-or.github.io/Ipopt/](https://coin-or.github.io/Ipopt/)
>
> [5] SCIP (Solving Constraint Integer Programs) solver. [https://www.scipopt.org/index.php](https://www.scipopt.org/index.php)
>
> [6] ORLM open-source model. [https://huggingface.co/CardinalOperations/ORLM-LLaMA-3-8B](https://huggingface.co/CardinalOperations/ORLM-LLaMA-3-8B)
>
> [7] Ali AhmadiTeshnizi, et al. OptiMUS: Scalable optimization modeling with (MI)LP solvers and large language models. ICML 2024.
>
> [8] Ziyang Xiao, et al. Chain-of-Experts: When LLMs meet complex operations research problems. ICLR 2024.
>
> [9] GPT4: All Details Leaked. [https://medium.com/@daniellefranca96/gpt4-all-details-leaked-48fa20f9a4a](https://medium.com/@daniellefranca96/gpt4-all-details-leaked-48fa20f9a4a)

---

> ### Comment · Area_Chair_neSr · 2024-11-25
> **ICLR Public Discussion Phase Ending Soon**
>
> Dear Reviewer,
>
> This is a kind reminder that the dicussion phase will be ending soon on November 26th. Please read the author responses and engage in a constructive discussion with the authors.
>
> Thank you for your time and cooperation.
>
> Best,
>
> Area Chair

---

> > ### Comment · Reviewer_feJ7 · 2024-11-27
> > **Thank you for the response**
> >
> > Thank you for your detailed response. However, I still have some concerns. As I mentioned in W1 and W5, can LLMOPT effectively address large-scale optimization problems? The example provided in your response (Part 7) does not appear to represent a large-scale optimization problem (please correct me if I am mistaken). Could you provide an example demonstrating how LLMOPT can solve large-scale optimization problems, such as a TSP instance with 1,000 nodes? If not, could you clarify the current limitations on the problem scale that LLMOPT can handle? I am confident that this will make a significant contribution to advancing this field.

---

> > > ### Author Response · Authors · 2024-11-29
> > > **Reply to Reviewer feJ7**
> > >
> > > Thank you for your response! We would like to clarify that **LLMOPT can handle large-scale problems**:
> > >
> > > 1. In our paper, the experiments cover nearly all datasets of optimization problems described in natural language, including linear programming problems with over 100 constraints and 500 variables, as well as traveling salesman problems with around 30 nodes. **This is not the limit of the problem scale that LLMOPT can handle.**
> > > 2. In our real industrial scenarios, as the previous reply, Example 1 in Reply (6/8) is an example of a large-scale linear programming problem in the context of car insurance leads distribution, which involves **2M+ leads (users) and 22 communities (companies)**, corresponding to the `model.I` and `model.J` variables in the code, read from the data files.
> > > 3. **As long as the solver can handle the corresponding problem scale, LLMOPT only needs to focus on formulating the problem and generating code**. As we previously clarified, the challenge of LLMOPT lies in problem defining, not solving. For problems with simple descriptions but large data volumes, LLMOPT can handle them well, as it focuses on correct formulating and matching suitable solvers.
> > > 4. **The challenge for future work lies in how to extract the correct problem formulation from more complex language descriptions (e.g., long texts)** and match a broader variety of solvers when dealing with complex problems, which is the direction we are currently exploring.
> > >
> > > Thanks again for your question! If you have any other questions, and please feel free to ask.

---

> > > ### Author Response · Authors · 2024-12-02
> > > **Gentle Reminder of the Rebuttal Deadline**
> > >
> > > Dear Reviewer feJ7,
> > >
> > > As the deadline approaches, we sincerely hope to address your concerns and discuss the rebuttal with you further. If you have any questions, please feel free to ask directly! Moreover, if you find our response satisfactory, could you please kindly consider the possibility of updating the rating. Thank you very much for your valuable suggestion.
> > >
> > > Best regards,
> > >
> > > Authors

---

> > > ### Author Response · Authors · 2024-12-03
> > >
> > > Dear Reviewer feJ7,
> > >
> > > We have carefully revisited your question and provide explanations from the following two perspectives in the hope of addressing your concerns.
> > >
> > > **Problem scale**: We have successfully deployed LLMOPT in large-scale real-world industrial scenarios. For example, in the first case mentioned above (reply 6/8), it involves a financial optimization problem with over 2,000,000 users and 22 insurance companies. LLMOPT can correctly model the problem and leverage our anonymous solver to find a solution. We are confident in LLMOPT’s ability to handle problems of this scale effectively.
> > >
> > > **Limitations**: The primary challenge for LLMOPT lies in extracting and modeling optimization problems from complex and diverse natural language texts, rather than solving the problems directly (as solving them is the task of the solver). For instance, some problems may not explicitly state that "the number of people must be a positive integer," yet such constraints can be inferred from common sense. Moreover, certain optimization problems require appropriate relaxation in modeling to be effectively solvable; otherwise, the solver might fail to find a solution. Currently, LLMOPT has achieved state-of-the-art performance on existing datasets. Additionally, we are actively collecting more complex problem descriptions and striving to enhance LLMOPT’s reasoning capabilities to enable more precise modeling and resolution of complex problems.
> > >
> > > As the deadline approaches, we sincerely hope to address your concerns further. We appreciate your effort and constructive comment once again!
> > >
> > > Best regards,
> > >
> > > Authors

---

> > > > ### Comment · Reviewer_feJ7 · 2024-12-03
> > > > **Official Comment by Reviewer feJ7**
> > > >
> > > > Thank you for your response. However, I believe my concern remains unresolved. You mentioned that
> > > >
> > > > > which involves 2M+ leads (users) and 22 communities (companies), corresponding to the model.I and model.J variables in the code, **read from the data files**.
> > > >
> > > >  but this seems inconsistent with your statement about
> > > > >The primary challenge for LLMOPT lies in **extracting and modeling optimization problems from complex and diverse natural language texts**
> > > >
> > > > Reading variables from data files appears unrelated to the task of modeling problem from natural language texts.
> > > >
> > > > Furthermore, as you pointed out, LLMOPT can solve the Traveling Salesman Problem with approximately 30 nodes, which is an extremely small scale. There is no evidence provided to demonstrate LLMOPT's performance on larger TSPs. Therefore, I would prefer to maintain my score.

---

> > > > > ### Author Response · Authors · 2024-12-03
> > > > >
> > > > > Thank you for your response! For LLMOPT, the difficulty of correctly formulating mathematical models is the same regardless of the scale of the optimization problem. The difference is that small-scale problems derive data from the problem description, whereas large-scale problems rely on data files. LLMOPT focuses solely on mathematical modeling and invoking solvers. Whether a large-scale problem (e.g., large-scale TSP) can be correctly solved is the responsibility of the professional solver and falls outside the scope of LLMOPT’s capabilities.
> > > > >
> > > > > We would like to emphasize that LLMOPT focuses on finding out a feasible learning-based way to formulate and solve optimization problems automatically, which achieves an average improvement of 11.08% in optimization generalization on a wide range of the existing common optimization benchmarks.

---

### Official Review · Reviewer_iJY5 · 2024-10-31

**Soundness:** 3
**Presentation:** 2
**Contribution:** 2
**Rating:** 5
**Confidence:** 3

**Summary:**

This paper proposes to finetune LLMs to improve MILP modeling from natural languages. Specifically, from natural language description of a MILP problem, this paper proposes to formulate the MILP problem as a five element formulation, and then generate solver code from the  formulation. It uses data augmentation to expand the data set, and ask domain experts to label the data set to finetune the LLMs. Through extensive empirical study, it shows performance improvement from a variety of competitive baseline methods.

**Strengths:**

- The authors did a good job benchmarking their method on a variety of benchmarks with many competitive baseline methods. The empirical evaluation seems to be comprehensive.
- To my knowledge, there has not been many works that aim to fine-tune the LLM for MILP modeling tasks, so the task is relatively novel.
- I appreciate the detailed discussion provided by the authors, which I think is valuable and can help guide the community thinking about future steps.

**Weaknesses:**

- The fine-tuning data collection requires expert manual labels, which limit the scalability and applicability of applying this method.
- Despite strength 2 mentioned above, the related work ORLM cited by the paper already took an initial step in fine-tuning LLMs for MILP modeling, and RLHF/DPO/KTO has been commonly used in LLM literature to finetune LLMs, so I’m a bit concerned whether the novelty of this work is sufficient, especially the work requires expert manual labors for constructing the fine-tuning data.
- Table 3: it seems like a majority of performance improvement is from self-correction instead of the five element components and KTO. If I understand correctly, the self-correction is prompting the LLM to correct any error it makes and has been used in previous papers such as [1], and it’s not related to the KTO fine-tuning pipeline. Given this, I’m a bit concerned about the contribution of the two main components (five elements and KTO) in this work.
- I find certain parts of the paper missing details and somewhat confusing (see my questions below).

**Questions:**

- Five-element formulation: to my knowledge, OPTIMUS [1] also identifies components such as variables, constraints, parameters etc in the optimization description before they translate the optimization problem into code. Can the authors comment on the difference between the two modeling approaches?
- Line 222: “experts review the generated problems, removing those with unclear descriptions or infeasible solutions to ensure data diversity and quality.” Can the authors comment on what are the criteria for the experts to determine the descriptions are “unclear”? How long is the expert labeling process? Can experts make mistakes? Can the authors comment on whether there is anyway to consider the expert mistakes to further improve the learning performance?
- Line 276 equation (3): I find the description of KTO confusing. For example, what is the reference model pi_ref used by the authors? Also, what is the optimal model (is it the same thing as the learning model)?
- Table 3: Can the authors comment on what is the setup w/o KTO? Is there still a training / fine-tuning component? what is the alternative loss?

Additional Feedback:

- line 53: The authors provide Wrong citation for ORLM on page 1.
- line 249: “are correct labeled by experts” → “are correctly labeled by experts”

[1] AhmadiTeshnizi, Ali, Wenzhi Gao, and Madeleine Udell. "OptiMUS: Scalable Optimization Modeling with (MI) LP Solvers and Large Language Models." ICML (2024).

---

> ### Author Response · Authors · 2024-11-21
> **Reply to Reviewer iJY5 (1/6)**
>
> Thanks for your questions and concerns raised in the review! The following are our responses to these issues. If you have any additional questions, please don’t hesitate to ask, and we will respond promptly.
>
> ## About the LLMOPT Novelty and Contributions
>
> ### Response to Question 1: Different between five-element and approach in OptiMUS.
>
> The _five-element_ in LLMOPT and the _SNOP_ representation in OptiMUS are completely different.
>
> 1. **The way of modeling**. **The five-element is extracted completely at once, while the SNOP representation in OptiMUS is extracted step by step**. How the SNOP data is extracted can be found in the website of OptiMUS [2], where the description, parameters and clauses are extracted step by step. In this step-by-step extraction approach, the correctness of the previous step directly affects the next step. Unreasonable subdivision of the extraction process will lead to incoherent and incorrect modeling.
> 2. **Formulation**. The five-element is a mathematical expression that describes and expresses the optimization problem in more detail and is more suitable for modeling languages such as Pyomo. **While the SNOP representation in OptiMUS is not a mathematical model but more like a data structure of the entity extraction task in NLP**. As shown in Fig.3(a) of the original paper of OptiMUS [1], the SNOP representation is composed of 7 types of information including problem_type, problem_info, input_format, output_format, output_info, objective, solver, which is not a mathematical formulation of optimization problem.
> 3. **Usage and correction**. As shown in the live demo [2] of OptiMUS, when using OptiMUS, the problem and specific data are input separately. While when using LLMOPT, you only need to input the problem described in natural language, and the five-element and the code will be generated automatically. **In OptiMUS, if the user does not manually modify the SNOP representation during the extraction process, it cannot be modified or corrected**. In LLMOPT, self-correction can automatically determine whether it is a five-element or code problem, and provide a detailed analysis.
>
> Moreover, **our method demonstrates a clear computational cost advantage over prompt-based methods, making it more suitable for large-scale applications**. For instance, our training cost (SFT combined with KTO) requires a total of  $ 3.17 \times 10^{19}$  FLOPs, with  $ 20.68 \times 10^{12}$  FLOPs needed for a single inference. In contrast, Optimus, which utilizes GPT-4, requires  $ 3.88 \times 10^{15}$  FLOPs per single inference. After 9,000+ API calls, our total computational cost becomes lower than that of Optimus, highlighting the cost-effectiveness and scalability of our approach(_see more details in the resposon to reviewer feJ7(5/8): Response to Question 4: Training and testing details of LLMOPT_).

---

> ### Author Response · Authors · 2024-11-21
> **Reply to Reviewer iJY5 (2/6)**
>
> ### Response to concern in Weakness 2: Compared with ORLM
>
> We appreciate your recognition of fine-tuning LLMs for optimization modeling tasks as a novel area of research. The difference between LLMOPT and ORLM:
>
> 1. **ORLM focuses on data augmentation methods, while LLMOPT focuses on how learning is conducted.** Although ORLM introduced four kinds of data augmentation methods, it does not focus on the learning process and without comprehensively evaluate model performance. In contrast, LLMOPT designs a detailed process for data, learning, and auto-testing. It not only declares the learning workflow at the methodological level (e.g., multi-instruction SFT and model alignment) but also conducts a thorough evaluation of model performance. **Therefore, LLMOPT is the first novel approach to explore both what to learn and how to learn.**
> 2. **ORLM focuses solely on generating solution code, whereas LLMOPT addresses both the formulating and solving of optimization problems.** Specifically, ORLM performs a straightforward task: inputting an optimization problem and directly inferring the corresponding solver Python code. In contrast, LLMOPT introduces a new learning task **Learning to Define** as a general formulation for optimization problems, enabling the generation of more accurate code. By using the five-element formulation as an intermediate step, LLMOPT can clearly define the problem and identify potentially overlooked hidden conditions,  enabling LLMOPT resulting in higher-quality code generation.
> 3. **LLMOPT conducted comprehensive seesaw tests (see the Section 5 and Appendix E in our paper), while ORLM has largely lost its ability to solve other basic problems.** We have reproduced and evaluated ORLM’s performance using the open-source model provided in [1]. The results show that what ORLM (based on LLaMA-3-8B) can do is only generating Coptpy solver code and the ORLM model cannot answer any other questions (e.g., _If all cats can climb trees, and Mike’s pet is a cat, then can Mike’s pet climb trees?_). This indicates that ORLM has significantly lost its capability on solving basic problems but optimization.
> 4. **The additional experiments show the superior generalization performance of LLMOPT compared to ORLM**. We find a new dataset from the _ICML 2024 Challenges on Automated Math Reasoning (Task 3)_ [2], which was not used in the training of either LLMOPT or ORLM. Since the test data for this dataset does not have open-source ground truth, we randomly sampled 200 data from its training dataset to serve as the test data. The solving accuracy results are as follows.
>
> |  | **GPT-4o** | **ORLM** | **LLMOPT** |
> | :---: | :---: | :---: | :---: |
> | The Competition Dataset [2] | 78.5% | 84.0% | **89.5%** |
>
> The results show that (a) **Compared to ORLM, LLMOPT shows better generalization performance even on a completely new dataset.** (b) Both LLMOPT and ORLM outperform GPT-4o, highlighting the potential of learning-based approaches in solving optimization problems.

---

> ### Author Response · Authors · 2024-11-21
> **Reply to Reviewer iJY5 (3/6)**
>
> ### Response to concern in Weakness 3: The contribution of self-correction.
>
> **The learning processes through multi-instruction SFT and KTO also improves the LLM’s correction ability**. If SFT and KTO are absent, high-quality correction cannot be achieved. Multi-instruction SFT and KTO alignment are the basis of self-correction. They not only improve the modeling and solving capabilities of LLMOPT, but also bring correction capabilities.
>
> To further analyze this issue, we conducted additional experiments. We compared the following methods:
>
> 1. LLMOPT. Full LLMOPT with self-correction during testing. Same as in the paper.
> 2. LLMOPT w/o KTO. LLMOPT with multi-instruction SFT but no model alignment during learning, with self-correction during testing.
> 3. LLMOPT corrected by GPT-4o. Inference is performed using the full learned model, and correction is performed using GPT-4o and the same prompt.
> 4. LLMOPT w/o self-correction (best of 12 repeats). Inference is repeated 12 times using the full learned model, and the best solution among the 12 solutions is manually selected as the final solution (which means that only one optimal solution is found in 12 repeated experiments). The reason we choose 12 is that self-correction is limited to a maximum of 12 repeated checks, so this is fair.
> 5. ORLM (best of 12 repeats). In addition, we reproduce ORLM [3] and add a correction mechanism. Since the ORLM model has lost its generalization ability, ORLM is used to repeat the reasoning 12 times and manually select the best solution among the 12 solutions to simulate the correction process.
>
> The experimental results of Solving Accuracy (SA) are as follows:
>
> |  | **NL4Opt** | **IndustryOR** | **Mamo_E** | **Mamo_C** |
> | :--- | :---: | :---: | :---: | :---: |
> | **LLMOPT** | **93.0%** | **46.0%** | **97.0%** | **68.0%** |
> | LLMOPT w/o KTO | 90.0% | 43.0% | **97.0%** | 65.0% |
> | LLMOPT corrected by GPT-4o | 89.0% | 41.0% | 95.0% | 66.0% |
> | LLMOPT w/o self-correction (best of 12 repeats) | 89.0% | 42.0% | 94.0% | 65.0% |
> | ORLM (best of 12 repeats) | 88.0% | 39.0% | 87.0% | 46.0% |
>
> From the results, it can be seen that, despite using the same prompt, the performance of LLMOPT w/o KTO is worse than that of the original LLMOPT across all three tasks. This demonstrates that KTO not only improves the ability to formulate optimization problems but also enhances the self-correction capabilities of the LLM. Furthermore, the correction method utilizing GPT-4o performs worse than LLMOPT, indicating that the learning processes through multi-instruction SFT and KTO also improves the LLM’s correction ability. Overall, the LLMOPT pipeline enhances the model’s comprehensive ability to handle optimization problems. **The experimental results for the reproduced ORLM (best of 12 repeats) show that LLMs fine-tuned using LLMOPT exhibit significantly stronger correction ability compared to those fine-tuned using ORLM**.

---

> ### Author Response · Authors · 2024-11-21
> **Reply to Reviewer iJY5 (4/6)**
>
> ## About expert labeling
>
> ### Response to concern in Weakness 1: Necessity of manual label and the exploration of scalability.
>
> We are glad that you paid attention to the data issue!
>
> **We believe that manual data labeling by experts is necessary**. As discussed in Section 5 of the paper, high-quality data plays a critical role in the performance of the model. However, there is a lack of open-source data in the field of optimization, and the available data is often disorganized and unreliable. For example, the following problem is an example from the NL4Opt dataset [1], which has been open-sourced by ORLM. This is a simple  integer programming problem (with only two variables and two linear constraints). The problem is as follows.
>
> > A chair produced by Elm Furniture yields a profit of 43, while every dresser yields a 52 profit. Each week, 17 gallons of stain and 11 lengths of oak wood are available. Each chair requires 1.4 gallons of stain and 2 lengths of oak wood, while each dresser requires 1.1 gallons of stain and 3 lengths of oak wood. Determine the maximum profit.
>
> The answer provided by ORLM is 236.5. A clear inconsistency arises here: given that the profits for both chairs and dressers are integers, it is illogical for the maximum profit to be a decimal. Upon verification, this incorrect result stems from the solution of producing 5.5 chairs and no dressers. The correct ground truth should be 224, achieved by producing 4 chairs and 1 dresser. Errors in labeling are not uncommon in open-source datasets, and various types of mistakes are universal across different datasets, such the lack of ground truth, the infeasibility of the problem. This further aggravates the challenges posed by the already limited availability of open-source data. **To address this issue, we dedicate approximately one month to creating a high-quality training dataset with the help of 12 experts**. This effort aims to ensure higher-quality model training and more reliable performance evaluation.
>
> At the same time, we have further studied the issue of data scalability from two perspectives. **First**, leveraging high-quality data labeled by human experts, we apply various automated data augmentation techniques to enrich our dataset.  **Second**, we are exploring reinforcement learning methods, such as Monte Carlo Tree Search (MCTS), to generate reasoning paths by utilizing GPT-4 to produce high-quality annotation data. For example, Pyomo does not support strict inequality constraints directly, so constraints like x > y often need to be transformed into non-strict inequalities like x >= y + 1e-6. By providing such examples, we aim to guide GPT-4o in generating _processes_, which will guide more accurate modeling and code. Once the model learns to transform strict inequalities, it should also be able to autonomously understand transformations for absolute value constraints. This ability will be a key factor influencing the scalability of automatic data annotation.

---

> ### Author Response · Authors · 2024-11-21
> **Reply to Reviewer iJY5 (5/6)**
>
> ### Response to Question 2: Details of expert judgement and labeling.
>
> Thank you for your question about the process of data augmentation and expert labeling. We will add more detailed explanations in the article.
>
> 1. **Details of expert judgement**. After augmented data, experts check whether the questions are clear during labeling five-element and code. Specifically, there are two types of questions considered _unclear_. One type involves questions with no feasible solution; experts determine this by running code to check for feasibility (e.g., checking for constraint conflicts). The other type involves obvious errors that contradict common sense during the modeling process, such as a car being faster than an airplane in transportation scenarios. This is typically judged based on the expert’s societal experience.
> 2. **Time and cost of expert labeling**. The labeling process is completed by 12 experts working collaboratively, taking approximately one month. The experts include 9 students with at least a bachelor’s degree (majoring in computer science or mathematics, all of whom had taken optimization courses), 1 university professor specializing in machine learning and optimization, and 2 algorithm engineers specializing in operations research and optimization. Among the 9 students, 4 are master’s students in related fields (including 1 PhD student).
> 3. **Mistake prevention during labeling**. We designed a four-stage expert labeling process to prevent incorrect annotations: _preliminary review_, _expert labeling_, _expert review_, and _data aggregation_. In the first stage, one expert, with the help of GPT-4, classifies the questions by difficulty level. Subsequently, each question is independently labeled by 2-3 different experts in the second stage (the number of experts depended on the difficulty). The labeling results are then manually reviewed and finalized during the third stage. For highly contested data, such cases are set aside and discussed by a panel of 5-7 experts to decide their fate. Finally, the data was organized into a training dataset. In terms of labeling validity, the review pass rate for the 7 experts responsible for labeling simple questions exceeded 90%, while the 4 experts handling complex questions had a pass rate exceeding 80% (2 of these experts worked on both simple and complex questions), with the lowest being 83.6%. **For simple questions, the agreement rate between two experts was 93.4%. For complex questions, the rate of two consistent labels out of three was 87.1%.** Considering these statistics, along with the third-stage review conducted by senior experts, the accuracy and reliability of the expert annotations were further ensured.
>
> Thank you for your questions! We will add more detailed data augmentation details in the paper and the complete expert labeling process in the appendix.

---

> ### Author Response · Authors · 2024-11-21
> **Reply to Reviewer iJY5 (6/6)**
>
> ## About KTO.
>
> ### Response to Question 3: Equation (3) for the KTO description.
>
> Thank you for your question! We apologize for the confusion caused by notation in KTO, and we will improve these description in the paper.
>
> In equation 3, we use the common notation in model alignment research. Similar notation is also used in studies such as [5, 6]. Specifically, $\pi^*$ represents the optimal model, that is, the model after KTO alignment. $\pi_{\text{ref}}$ is the reference model, that is, the model after Supervised Fine-Tuning (SFT). In alignment research, the SFT model (reference model) is usually used as the initial model because it can already demonstrate a certain degree of ability (through supervised training with a large amount of labeled data). However, in order to eliminate model hallucinations and improve the comprehensive ability of the model, the SFT model is often not enough and needs to be aligned. In this process, the optimal model is the alignment target and is a model that is more in line with human needs, while the reference model is an unaligned model and is the benchmark for comparison.
>
> We will add relevant instructions in the paper. Thank you for your question!
>
> ### Response to Question 4: Experiment setup of w/o KTO.
>
> We apologize for the confusing experimental setup! The setup of w/o KTO in Table 3 is an ablation version of LLMOPT. Specifically, during the learning phase, only multi-instruction supervised fine-tuning (SFT) is performed and KTO alignment is not performed. That is, the model after supervised fine-tuning (LLM_{SFT} in Figure 2(b)) is directly used as the model for deployment in the auto-testing process. All other settings remain consistent with those of LLMOPT.
>
>
> We hope that our response has addressed your concerns, but if we missed anything please let us know.
>
> **References**:
>
> [1] OptiMUS: Optimization Modeling Using mip Solvers and large language models. [https://arxiv.org/pdf/2310.06116](https://arxiv.org/pdf/2310.06116)
>
> [2] OptiMUS website. [https://optimus-solver.com/](https://optimus-solver.com/)
>
> [3] ORLM open-source model. [https://huggingface.co/CardinalOperations/ORLM-LLaMA-3-8B](https://huggingface.co/CardinalOperations/ORLM-LLaMA-3-8B)
>
> [4] NL4Opt dataset. [https://huggingface.co/datasets/CardinalOperations/NL4OPT/viewer](https://huggingface.co/datasets/CardinalOperations/NL4OPT/viewer)
>
> [5] Kawin Ethayarajh, et al. Model alignment as prospect theoretic optimization. ICML 2024.
>
> [6] Rafael Rafailov, et al. Direct Preference Optimization: Your Language Model is Secretly a Reward Model. NeurIPS 2023.

---

> ### Comment · Area_Chair_neSr · 2024-11-25
> **ICLR Public Discussion Phase Ending Soon**
>
> Dear Reviewer,
>
> This is a kind reminder that the dicussion phase will be ending soon on November 26th. Please read the author responses and engage in a constructive discussion with the authors.
>
> Thank you for your time and cooperation.
>
> Best,
>
> Area Chair

---

> ### Comment · Reviewer_iJY5 · 2024-11-27
>
> Dear Authors,
>
> Thank you for your effort in preparing the rebuttal. It addresses some of the questions I raised. However, I still have concerns regarding the learning contribution and novelty of this paper, as well as the significant human effort required, which limits the scalability of the proposed method. I will maintain my score but remain open to discussion with the AC and other reviewers in the next stage.
>
> Thank you.

---

> > ### Author Response · Authors · 2024-11-29
> > **Reply to Reviewer iJY5**
> >
> > Thank you for your response! We're happy to hear that some of your concerns have been resolved.
> >
> > **In order to deal with the optimization generalization issue effectively, LLMOPT is the first learning-based approach to propose learning-to-define and intact framework for leveraging LLMs to define and solve general optimization problems, achieving state-of-the-art performance across various tasks**. Unlike ORLM, which focuses solely on data, LLMOPT designs a comprehensive learning process. LLMOPT also introduces the universal five-element formulation to define various optimization problems, designes the self-correction mechanism. Moreover, LLMOPT achieves SOTA optimization generalization performance across a wide range of optimization tasks, outperforming both prompt-based methods like OptiMUS and learning-based approaches like ORLM.
> >
> > High-quality data is crucial for research in every field. We believe that someone always needs to take the first step, investing time and effort to accomplish this task initially. We will propose a comprehensive benchmark for leveraging LLMs to solve optimization problems, featuring a well-designed dataset and a complete evaluation process. We are currently utilizing a range of automated data augmentation techniques combined with MCTS to generate simulation data, aiming to enhance our dataset and address the issue of data scalability.
> >
> > Thank you again for your kind questions! We would greatly appreciate the opportunity to discuss further with you to address your concerns, and please feel free to ask.

---

> > ### Author Response · Authors · 2024-12-02
> > **Gentle Reminder of the Rebuttal Deadline**
> >
> > Dear Reviewer iJY5,
> >
> > As the deadline approaches, we sincerely hope to address your concerns and discuss the rebuttal with you further. If you have any questions, please feel free to ask directly! Moreover, if you find our response satisfactory, could you please kindly consider the possibility of updating the rating. Thank you very much for your valuable suggestion.
> >
> > Best regards,
> >
> > Authors

---

> > ### Author Response · Authors · 2024-12-03
> >
> > Dear Reviewer iJY5,
> >
> > We have carefully revisited your question and provide explanations from the following two perspectives in the hope of addressing your concerns.
> >
> > **Learning contribution**: LLMOPT is the first learning-based approach to introduce a comprehensive framework that tackles the optimization generalization issue by proposing a learning-to-define methodology. This approach leverages LLMs to both define general optimization problems and enhance code accuracy. By employing multi-instruction learning, LLMOPT enables LLMs to model a wide range of problems and generate corresponding solving code, achieving state-of-the-art performance across more general tasks. Unlike related work, which often only focus on data augmentation or prompt engineering without truly addressing the process of learning, LLMOPT provides an intact framework that fully integrates data, learning, and auto-testing, setting a new standard in this field.
> >
> > **Human effort**: We acknowledge that a significant amount of human effort is invested in the data collection and annotation process, but we firmly believe this effort is essential. Complex, diverse, and high-quality data must first be curated by humans rather than relying solely on LLMs, as someone has to take the first step. Our team of human experts is actively constructing more complex optimization problem datasets and exploring the use of automated data augmentation techniques combined with Monte Carlo Tree Search (MCTS) to generate simulation data, aiming to enhance our dataset and address the challenge of data scalability.
> >
> > As the deadline approaches, we sincerely hope to address your concerns further. We appreciate your effort and valuable feedback once again!
> >
> > Best regards,
> >
> > Authors

---

> > > ### Comment · Reviewer_iJY5 · 2024-12-03
> > >
> > > Thank you for the detailed response. I have thoroughly reviewed the authors' response and their replies to other reviewers. At this time, I do not have any additional questions for the authors. I appreciate the authors' effort in explaining the contribution of the paper, but I would like to keep the score for now as I'm not sure if the paper's contribution is sufficient for ICLR. As mentioned previously, I remain open to discussions with the AC and other reviewers in the next stage.

---

> > > > ### Author Response · Authors · 2024-12-04
> > > >
> > > > We sincerely appreciate your recognition of our efforts during the rebuttal process. We warmly invite your active participation in the subsequent discussions, as your insights will be invaluable in further improving the quality of our paper. Once again, thank you for your thoughtful and constructive comments.

---

### Official Review · Reviewer_JwJz · 2024-11-02

**Soundness:** 2
**Presentation:** 2
**Contribution:** 2
**Rating:** 6
**Confidence:** 4

**Summary:**

This paper proposes LLMOPT, which finetunes an LLM to formulate optimization problems (from problem description -> mathematical model/solver model). The finetuning is performed in two stages: SFT stage (where the LLM is finetuned using MLE on ground truth outputs), and KTO-based alignment (where the LLM is finetuned based on desirability labels on LLM generated outputs).

**Strengths:**

The paper addresses an interesting and timely problem. Optimization modeling is a huge topic, and introducing techniques to automate (partially) this task will have significant impact.

To my knowledge, this is also one of the first works that builds/finetunes LLMs specifically for writing optimization models.

The empirical analysis is comprehensive (spanning many benchmarks) and the resulting performance gains are impressive.

**Weaknesses:**

1. **Novelty**: The authors mentioned ORLM, which similarly trains an LLM to do optimization modeling, but did not provide a direct comparison. I also read ORLM (not in the most detail), but it appears to do some data augmentation to train an LLM for model formulation. It seems the main difference is the alignment step (using KTO) and the self-reflection step, can the authors explain the novelty of their method compared to ORLM?

2. **KTO alignment**: There are a few comments on this:
* **Writing/clarity**: The writing in S3.3.2 is quite hard to follow, I had to read the original paper to understand what this part is doing. Importantly, Equation (3) is not correct, in the original DPO paper, the optimal reward function has an additional log partition function term. I did not check if this affected the rest of the formulation.
* **KTO dataset**: Based on my understanding: (1) the SFT step does not use the KTO dataset (which contains GPT4 generated responses, and desirability labels), and (2) the alignment step does not use the original dataset (which contains ground truth formulations). Is my understanding correct? If so, what is the motivation for not using the KTO dataset during SFT and original dataset in KTO alignment (where the ground truth are all labeled as 'desirable')?
* **Purpose of KTO**: The authors state that `KTO loss function encourages the optimal model $\pi^*$ to produce completions that align more closely with expert-labeled data`. **Minor**: this can't be the optimal model, but the learned policy $\pi$? **Major**: This is exactly the same purpose as SFT, so we are back to the above point of why the KTO dataset is not used directly through SFT, it would likely be more stable and have lower variance compared to KTO updates.
* **Ablation study**: Can the authors more comprehensively ablate the importance of the KTO step? I carefully examined the results in Table 3 (ablation study)---which seems to indicate that `w/o KTO` on many benchmarks does not significantly improve SA. I would be interested to also see how LLMOPT performs when the KTO dataset is used directly during SFT.

3. **Questions about results**:
* **AST**: Can the authors help me understand the big difference in average solution time (AST) plotted in Fig 4? I had a look at some examples in `NLP4LP`, these are very straightforward LP problems. As such, I am very surprised that LLMOPT and GPT4 have almost 2x difference in solution time. Given the simplicity of the problems in this dataset, (1) this difference is unlikely to be explained by clever reformulations that improve solution time, (2) and unlikely to be noise.
* **Solution accuracy**: Can the authors elaborate on exactly how solution accuracy is calculated? Is this based on some test cases? My intuition is that it is extremely difficult to check the formulation directly, so how is SA computed and is this a robust evaluation method?
* **Performance by problem type**: In L422-L431, the authors claim their method is more general, which I understood as achieving better performance on more types of problems (e.g. LP, MILPs, QPs). To confirm this claim, I would like to see the performance by problem type (similar to the Figure 8 in App G).

**Questions:**

I also have some minor concerns:

* **Dataset construction**: In L219, the authors mentioned they had 1763 'seed' problems. Did they label formulations for all 1763 problems? If so, this is a pretty big contribution if the authors choose to open-source their dataset.
* **Augmented data**: My understanding is that additional examples were introduced by mutating the original seed problems. Were these also formulated and labeled by human experts? How many examples in total (original + mutated) were used in training?
* Can the authors describe the procedure of collecting the labelled formulations (if human experts were indeed used). This seems like a huge undertaking especially if they are required to write out the mathematical models and code.
* **Eq(4) and (5)**: The notation $\nu \sim \cdot$ was slightly confusing, since it is not a random variable, which is where this notation is mainly used.
* **Eq (5)**: Are $\lambda_D$ and $\lambda_U$ learnt parameters or hyperparameters?
* **Solver**: What solver did LLMOPT use? Is it `Pyomo` as shown in the method figure? Are the baselines evaluated with the same solvers?
* ps. this is just a suggestion, but the use of the word `alignment` is slightly misleading, the authors are not aligning the LLM to principles, but doing a different stage of finetuning with a different dataset.
* pps. another minor suggestion, but the authors should make more prominent that KTO is an existing method, I saw the citation, but it is slightly buried, and might convey the impression that KTO is an original contribution of this work.

I am going to start with a conservative rating, but open to revising if the authors address my concerns.

---

> ### Author Response · Authors · 2024-11-21
> **Reply to Reviewer JwJz (1/9)**
>
> We appreciate your valuable and thoughtful feedback. We have carefully considered your feedback and provided detailed responses below. If there are any other questions, please feel free to ask, and we will respond promptly.
>
> ## Response to concerns in Weakness 1: Compared with ORLM.
>
> The contributions of LLMOPT and ORLM are fundamentally different.
>
> 1. **ORLM focuses on data augmentation methods, while LLMOPT focuses on how learning is conducted.** Although ORLM introduced four kinds of data augmentation methods, it does not focus on the learning process and without comprehensively evaluate model performance. In contrast, LLMOPT designs a detailed process for data, learning, and auto-testing. It not only declares the learning workflow at the methodological level (e.g., multi-instruction SFT and model alignment) but also conducts a thorough evaluation of model performance. **Therefore, LLMOPT is the first novel approach to explore both what to learn and how to learn.**
> 2. **ORLM focuses solely on generating solution code, whereas LLMOPT addresses both the formulating and solving of optimization problems.** Specifically, ORLM performs a straightforward task: inputting an optimization problem and directly inferring the corresponding solver Python code. In contrast, LLMOPT introduces the **Learning to define** as a general formulation for optimization problems, enabling the generation of more accurate code. By using the five-element formulation as an intermediate step, LLMOPT can clearly define the problem and identify potentially overlooked hidden conditions, resulting in higher-quality code generation.
> 3. **LLMOPT conducted comprehensive seesaw tests (see the Section 5 and Appendix E in our paper), while ORLM has largely lost its ability to solve other basic problems.** We have reproduced and evaluated ORLM’s performance using the open-source model provided in [1]. The results show that what ORLM (based on LLaMA-3-8B) can do is only generating Coptpy solver code and the ORLM model cannot answer any other questions (e.g., _If all cats can climb trees, and Mike’s pet is a cat, then can Mike’s pet climb trees?_). This indicates that ORLM has significantly lost its capability on solving basic problems but optimization.
> 4. **The additional experiments show the superior generalization performance of LLMOPT compared to ORLM.** We find a new dataset from the _ICML 2024 Challenges on Automated Math Reasoning (Task 3)_ [2], which was not used in the training of either LLMOPT or ORLM. Since the test data for this dataset does not have open-source ground truth, we randomly sampled 200 data from its training dataset to serve as the test data. The solving accuracy results are as follows.
>
> |   | **GPT-4o** | **ORLM** | **LLMOPT** |
> | :---: | :---: | :---: | :---: |
> | The Competition Dataset [2] | 78.5% | 84.0% | **89.5%** |
>
> The results show that (a) **Compared to ORLM, LLMOPT shows better generalization performance even on a completely new dataset.** (b) Both LLMOPT and ORLM outperform GPT-4o, highlighting the potential of learning-based approaches in solving optimization problems.

---

> ### Author Response · Authors · 2024-11-21
> **Reply to Reviewer JwJz (2/9)**
>
> ## Response to concerns about KTO. (Weakness 2 and Questions about equations)
>
> Thank you for your careful review! We apologize that the introduction of KTO is confusing, and we will carefully revise the description of KTO in the paper. Here we answer your concerns about KTO in Weaknesses and Questions one by one.
>
> 1. **Purpose of model alignment and KTO.**
>
> Thank you for your thoughtful feedback. The word "alignment" may be ambiguous, especially for readers in different fields. We apologize for not providing more detailed background information. In fact, aligning generative models with human feedback has been successfully used to make generations more helpful, factual, and ethical, among other desiderata [13, 15].
>
> **It is important to clarify that the targets and methodologies of model alignment and supervised fine-tuning (SFT) are different.** Alignment primarily aims to address the issue of hallucination in our work. In this paper, our goal is to enable LLMs to formulate optimization problems and generate corresponding solving code. However, despite SFT training the model to learn how to write solving code, LLMs may still exhibit hallucination when faced with novel problems. This hallucination manifests as outputs that appear plausible but are, in fact, fabricated or inaccurate.
>
> A simple yet typical example of hallucination involves the handling of strict inequality constraints (e.g., `>` and `<`). Most solvers cannot directly process such constraints; however, an LLM trained solely with SFT might incorrectly define these conditions when generating code. This error arises from flawed reasoning, where the LLM falsely analogizes Python-supported `>` and `<` operators to optimization problem modeling. The correct approach, however, is to approximate strict inequalities by converting them into non-strict inequalities with a small positive margin. This is a classic case of hallucination: the generated code appears plausible but is fundamentally incorrect. In such scenarios, **model alignment can be performed after SFT to address this issue.** By introducing preference-based interventions (e.g., explicitly marking certain samples as incorrect with desirability = False in the KTO framework used in this paper), hallucination can be mitigated. This results in logically sound models and more executable, standardized solving code. Therefore, alignment is both necessary and critical for achieving robust performance.
>
> Thank you again for your question! We will clarify the concept of alignment more thoroughly, provide sufficient citations, and offer a more detailed explanation of the purpose of KTO in the paper.
>
> 2. **Datasets of SFT and KTO.**
>
> We apologize for not clearly expressing the SFT and KTO dataset splits! We have clarified the details of the dataset in the _response to questions about the data augmentation and labeling by experts_ in the next response section. The data labeled as True in the KTO dataset is consistent with the data in the SFT dataset, with a total of 9,828 entries. The number of the False data in the KTO dataset is 9,735. Thank you again for your valuable comments!

---

> ### Author Response · Authors · 2024-11-21
> **Reply to Reviewer JwJz (3/9)**
>
> 3. **Explanation of equation 3.**
>
> Thank you for your meticulous review! Your understanding of DPO [15] is accurate, and the Equation 3 in our paper is also correct. It aligns with the reward expression in Equation 8 of the KTO paper [12] and the official KTO code [13]. **The misunderstanding stems from the fact that the reward function in KTO is different from that in DPO [15]**. This difference is explicitly clarified in Section 3.2 of the KTO paper [13], where the partition function is deliberately subtracted to ensure the validity of the reference point.
>
> Let us analyze this issue further. (In this paper, the data representations $u$ and $v$ correspond to $x$ and $y$ in KTO [13] and DPO [15], respectively. For consistency, we will use the notation $u$ and $v$ in the explanation below.)
>
> **Optimal policy in model alignment.** In both KTO [13] and DPO [15], the optimal policy can be expressed as $\pi^*(v \mid u) = \frac{1}{Z(u)} \pi_{\text{ref}}(v \mid u) \exp \left( \frac{1}{\beta} r(u, v) \right)$ as shown in equation 4 in [15]. Here, the partition function $Z(u) = \sum_v \pi_{\text{ref}}(v \mid u) \exp \left( \frac{1}{\beta} r(u, v) \right)$ is a constant that depends only on the input $u$, so that the value of the optimal policy $\pi^*(v \mid u)$ is normalized.
>
> **Reward function in DPO.** According to the expression of the optimal strategy $\pi^*(v \mid u)$, we can deduce that the reward function of DPO is $r(u, v) = \beta \log \frac{\pi^*(v \mid u)}{\pi_{\text{ref}}(v \mid u)} + \beta \log Z(u)$, where the partition function $Z(u)$ is a constant that is independent of $v$. In DPO [15], it is necessary to compare the relative merits of two different responses $v_1$ and $v_2$ for the same input $u$, and use the difference in their rewards to express the preference $p(v_1 \succ v_2 \mid u) = \sigma \left( r(u, v_1) - r(u, v_2) \right)$. At this time, the partition function $Z(u)$ is eliminated, that is, the partition function has no effect on the preference of DPO.
>
> **Reward function in KTO.** Unlike DPO [15], the preference in KTO is no longer achieved by comparing $v_1$ and $v_2$, but refers to the human preference for $v$ in a global context, that is, the desirability of $v$. To achieve this preference, for question $u$, KTO uses $z_{\text{ref}} = \beta\text{KL}(\pi^*(v' \mid u) \mid \mid \pi_{\text{ref}}(v' \mid u))$ to represent the reference point to calculate the value function of answering $v$. The reward function KTO is just the original reward shifted by an input-specific term (i.e., the partition function term is ignored) [13] , resulting in the value function in KTO (as shown in equation 4 in the submitted paper) showing a form similar to the DPO preference function, achieving model alignment. KTO proves that this approach of omitting the partition function is equivalent to the original reward function (Lemma 1 in [13]).
>
> Therefore, KTO deliberately designs a special reward function based on the choice of the reference point (characterized by the absence of a partition function term) and demonstrates the validity of this reward formulation, successfully achieving model alignment [13].

---

> ### Author Response · Authors · 2024-11-21
> **Reply to Reviewer JwJz (4/9)**
>
> 4. **Explanation of hyperparameters (such as lambda_D and lambda_U in equation 5).**
>
> Thank you for your interest in the details of LLMOPT! **In the submitted paper, all the hyper-paremeters have been introduced in Appendix B.** In equation 5, lambda_D and lambda_U are hyper-parameters used to represent the desirable and undesirable weights. In LLMOPT, we set lambda_D=1.0 and lambda_U=1.0 according to the default setting of the KTO paper [12] and the official code [14].
>
> 5. **Explanation of notation in equation 4 and equation 5.**
>
> Thank you for raising such valuable questions! In equation 4 and equation 5, **the notation $v \sim \cdot$ is consistent with the expression in equation 8 of the KTO original paper [12].** We have carefully reviewed all formulas and this notation, and we agree with your view that “the notation $\sim$ typically denotes being drawn from a probability distribution.” This representation may cause confusion. Specifically, we will change $v \sim v_{\text{desirable}} | u$ to $d = \text{True} \mid u, v$, and $v \sim v_{\text{undesirable}} | u$ to $d = \text{False} \mid u, v$.
>
> 6. **Ablation analysis for KTO.**
>
> Thank you for your interest in the effect of KTO! We designed an experiment to perform ablation analysis on KTO. All the following experiments are performed in a fair setting with the five-element formulations and the self-correction mechanism. The baseline is GPT-4o. On the one hand, we conducted an SFT-only experiment on Qwen1.5-14B to analyze the performance improvement brought by SFT. On the other hand, we performed KTO alignment on the above SFT-based model to analyze the performance improvement brought by KTO over SFT. The experimental results are shown in the following table.
>
> |   | **NL4Opt** | **MamoEasy** | **MamoComplex** | **IndustryOR** | **NLP4LP** | **ComplexOR** | **Avg. of Improvement** |
> | :---: | :---: | :---: | :---: | :---: | :---: | :---: | :---: |
> | **GPT-4o** | 83.0% | 90.0% | 38.0% | 34.0% | 35.2% | 36.4% | / |
> | **LLMOPT (only SFT)** | 90.0% | 97.0% | 65.0% | 43.0% | 64.9% | 54.6% | / |
> | **SFT Improves over Baseline** | **+7.0%** | **+7.0%** | **+27.0%** | **+9.0%** | **+29.7%** | **+18.2%** | **+16.3%** |
> | **LLMOPT (SFT+KTO)** | 93.0% | 97.0% | 68.0% | 46.0% | 83.8% | 72.7% | / |
> | **KTO Improves over SFT** | **+3.0%** | **+0.0%** | **+3.0%** | **+3.0%** | **+18.9%** | **+18.1%** | **+7.7%** |
>
> The results show that (1) SFT achieves an average improvement of 16.3% over the baseline, while KTO achieves an average improvement of 7.7% over SFT. The ratio of improvement brought by KTO to that brought by SFT is approximately 1:2. The goal of model alignment is to reduce hallucinations in LLMs, thereby improving their overall performance when dealing with various optimization problems. As a result, KTO is less likely to achieve the rapid performance boost for specific tasks that SFT can deliver. We believe the 1:2 performance improvement ratio is reasonable. (2) For datasets not included in the training set, such as NLP4LP and the Complex dataset, the performance improvement brought by KTO is even more significant. Problems in these datasets are more novel for LLMs and are more likely to trigger hallucinations or other types of errors. **The impressive improvements observed with KTO highlight the effectiveness and necessity of KTO alignment**.

---

> ### Author Response · Authors · 2024-11-21
> **Reply to Reviewer JwJz (5/9)**
>
> ## Response to Questions about the data augmentation and labeling by experts.
>
> **(1) The process of expert labeling**, which is divided into four stages: preliminary review, expert labeling, expert review, and data aggregation.
>
> + _**Preliminary review**_**.** Initially review the data and remove unfeasible problems (unfeasible not means difficult). With the help of GPT-4o, we divide optimization problems into two categories according to their difficulty. Problems that meet one of the following conditions will be classified as difficult problems: at least 3 out of 5 solutions using GPT-4o are inconsistent, the code generated by GPT-4o has errors, and experts have found complex constraints, reasoning, or large amounts of data.
> + _**Expert labeling**_**.** For simple questions, 2 experts independently annotate. And 3 experts independently label for complex questions. For each question, five-element and solver code need to be labeled, and the code must be run without errors. In this stage, experts may use GPT-4o to generate text that meets the expert's intentions to reduce typing time and generate more formatted code. In order to improve data quality, experts may modify questions appropriately to make them more suitable for the problem scenario, or delete inappropriate questions (such as unfeasible ones).
> + _**Expert review**_**.** For each question, a new expert checks the labels of other experts in the previous step, which is based on the correctness of the problem modeling and the consistency of the labels of different experts (the same question may have different but correct labels from different experts). Highly controversial questions or those with any incorrect labels are included in a independent challenging dataset.
> + _**Data aggregation**_**.** Five experts discuss and analyze each of the data in the independent difficult dataset to determine whether the problem has a solution or can be adjusted to a feasible problem, and decide whether to abandon the problem based on this. If not, the experts will discuss and complete the labeling of these data. The correctly labeled questions that has passed the expert review will be summarized. And the other feasible questions that has been incorrectly labeled during the entire labeling process will be summarized.
>
> **(2) About data for SFT and KTO.** In the process described above, during the expert labeling process, GPT-4o is used to assist in generating more standardized text, and experts review these results to ensure they align with their intentions. Since the texts generated by GPT-4o are not always correct, incorrect labeling are found by the experts during the second and third stages. In the fourth stage, all the feasible problems are divided into two datasets according to correct labels and incorrect labels. Data with correct labels are used to construct the SFT training set as well as the positive samples in the KTO training set. Meanwhile, data with incorrect labels are used to construct the negative samples in the KTO training set. Consequently, the data included in the SFT training set are inherently part of the KTO training set as well. We will provide detailed explanations in the paper to eliminate any potential misunderstandings regarding the data labeling process. Thanks again for your suggestions!

---

> ### Author Response · Authors · 2024-11-21
> **Reply to Reviewer JwJz (6/9)**
>
> **(3) Results of labeling.** We have already introduced this information in Appendix A.2, and a more detailed description is provided below.
>
> First, we used 1,763 seed problems for data augmentation, resulting in approximately 3,000 raw optimization problems. Then, through the expert labeling process described in Response (1), we obtained 3,276 optimization problems correctly labeled with five-element formulations and solver codes. Among the incorrectly labeled data, 3,145 entries are incorrectly labeled for the five-element, while 3,295 entries are incorrectly labeled for the solver code.
>
> Then, we constructed the SFT dataset using the 3,276 correctly labeled data points. These data points are used to generate three types of data pairs to support multi-instruction SFT: _(question, five-element)_, _(question, code)_, and _(five-element, code)_. As a result,**the SFT dataset contains  3x3,276=9,828  training samples**.
>
> Next, we introduce the construction of the KTO dataset.** In the KTO dataset, samples with **_**desirability**_** labeled as True are identical to the samples in the SFT dataset, totaling 9,828 samples.** Samples labeled as False are constructed from incorrectly labeled data. Similar to the multi-instruction SFT approach, these samples are divided into three categories: data incorrectly labeled for the five-element representation are used to construct _(question, five-element, False)_ samples, while data incorrectly labeled for the solver code are used to construct _(question, code, False)_ and _(five-element, code, False) _samples. As a result, the samples labeled as False in the KTO dataset total 3,145 + 3,295 * 2 = 9,735. **Thus, the entire KTO dataset contains 9,828 (True) + 9,735 (False) = 19,563 samples.**
>
> Finally, general open-source data are added to both the SFT and KTO datasets to maintain the generalization capability of the LLM. Specifically, 20,000 samples are randomly selected from [3] and add to the SFT dataset, while 30,000 samples are randomly selected from [4] and add to the KTO dataset. These have been described in Appendix A.2 of the submitted paper. As a result, the final SFT dataset contains 29,828 samples, and the KTO dataset contains 49,563 samples.
>
> **(4) Reliability of expert labeling.**
>
> Here is a brief and anonymized introduction to the **qualifications of the experts** involved. The data labeling process is completed by a team of 12 experts working collaboratively. The preliminary review and expert labeling phases are carried out by 9 undergraduate or with above degree (majoring in Computer Science or Mathematics, all of whom have taken optimization courses), including 4 graduate students in related fields (1 of whom is a Ph.D. candidate). The expert review phase is conducted by 1 university professor interested in machine learning and optimization and 2 algorithm engineers researching operations optimization. The data aggregation phase is completed by all experts except the undergraduates. Throughout the data labeling process, **each expert worked independently, and no single problem was assigned to the same expert more than once in the first three phases**. These details will be provided in the paper.
>
> Thank you again for your concerns of the data processes! We will add the above content to the corresponding data description section and appendix in the paper.

---

> ### Author Response · Authors · 2024-11-21
> **Reply to Reviewer JwJz (7/9)**
>
> ## Response to concerns in Weakness 3: Explanation of the results.
>
> 1. **Response to AST experimental results.**
>
> We are glad that you pay attention to the specific results on the NLP4LP dataset. In fact, NLP4LP is a medium-difficulty dataset for LLM. Although the problems in this dataset are all linear programming, they often involve a rich variety of constraints and problem contexts. From the results in Table 2 of the submitted paper, we can see that the three methods of GPT-4 Directly, Reflexion, and Chain-of-Experts all achieved low solving accuracy (35.8%, 46.3%, and 53.1% respectively).
>
> As mentioned in Section 4.1 in the paper, **the a verage solving times (AST) refers to the average number of self-correction processes performed during the test, and the maximum number of self-correction re-solves is limited to 12.** To better explain the phenomenon you raised, we provide the specific numerical values of the results on the NLP4LP dataset in Figure 4(a), as shown in the following table.
>
> |   | **GPT-4-Turbo** | **GPT-4o** | **LLMOPT (Qwen-1.5-14B)** |
> | :---: | :---: | :---: | :---: |
> | **SA** | 37.8% | 35.2% | 83.8% |
> | **AST** | 10.08 | 10.40 | 7.00 |
>
> The self-correction mechanism is employed to verify whether the five-element or solver code contains errors and to iteratively refine the it in an attempt to produce a correct solution. To avoid inefficient loops of corrections, this mechanism imposes a maximum number of attempts, set at 12 as mentioned in the paper. However, for complex code, even advanced LLMs sometimes struggle to complete the correction process successfully, manifesting in repeated attempts that fail to yield a correct solution. In such cases, the weaker-performing LLMs reveal significant limitations. First, their solving accuracy (SA) is relatively low, resulting in a higher rate of incorrect solutions. Second, the frequency of unsuccessful correction attempts significantly increases the solving time. Although the optimal solution could not be found, these ineffective attempts lead to a sharp rise in the average solving time (AST). From the table, it can be observed that GPT-4-Turbo and GPT-4o exhibit much lower SA compared to LLMOPT, indicating the lack of ability to tackle NLP4LP problems. **Even with the self-correction mechanism, these models fail to effectively enhance solving accuracy. However, these LLMs will still try to correct repeatedly, resulting in an increase in AST.** Therefore, for LLM with poor performance, not only is the solving accuracy very low, but frequent invalid corrections will also significantly drag down the average solution time, ultimately resulting in a double disadvantage.
>
> 2. **The evaluation of solving accuracy (SA).**
>
> As described in Section 4.1 of the paper, solving accuracy (SA) indicates the percentage of LLMs that correctly solve the optimization problem, i.e., find the optimal solution. SA or similar evaluation methods have not been specifically mentioned in previous work [9-11]. In the experiments of this paper, _the judgment of correct solution_ is sufficient when the following three conditions are met at the same time:
>
> **(a) LLM generates code based on the problem and five-element, and executes it without error**.
>
> **(b) Executing the solver code output the optimal solution and its corresponding objective function value in the terminal**.
>
> **(c) The output objective function value is consistent with the ground truth provided by the dataset**.
>
> Conditions (a) and (b) can be automated using Python. When determining condition (c), we use GPT-4o to analyze the consistency between the output of the executed code and the ground truth, and then determine whether the solution is correct (this is obviously within the capabilities of GPT-4o). The prompt used is: _The optimal objective function value of an optimization problem is {ground_truth}. Please determine whether the following solution printed by the solver is correct. The output of the solver is: {solver_output}._
>
> We have checked the LLMOPT solutions on the NL4Opt dataset, and the experts' judgment on whether all problems _are_ _solved correctly_ is consistent with the judgment of the above process.

---

> ### Author Response · Authors · 2024-11-21
> **Reply to Reviewer JwJz (8/9)**
>
> 3. **Performance by problem type.**
>
> Thank you for your valuable question! In order to demonstrate the generality of LLMOPT, we re-analyzed all experimental results and classified the types of optimization problems involved in each dataset, including Linear Programming (LP), Integer Programming (IP), Mixed Integer Programming (MIP), Nonlinear Programming (NP), Combinatorial Optimization (CO), Multi-objective Programming (MOP) and Others. The following table shows **the SA performance of LLMOPT in solving various types of problems on different datasets**. The data is the _solving accuracy (SA)_, and the values in brackets represent _(number of problems solved correctly/number of problems in the dataset)_. Detailed statistics are as follows:
>
> |   | **NL4Opt** | **MamoEasy** | **MamoComplex** | **IndustryOR** | **NLP4LP** | **ComplexOR** |
> | :---: | :---: | :---: | :---: | :---: | :---: | :---: |
> | **LP** | 90.5% (38/42) | - | 76.0% (19/25) | 60.0% (12/20) | 77.8% (7/9) | 60.0% (3/5) |
> | **IP** | 95.8% (46/48) | 100.0% (36/36) | 100.0% (5/5) | 45.5% (5/11) | 0.0% (0/1) | - |
> | **MIP** | 90.0% (9/10) | 95.3% (61/64) | 73.9% (17/23) | 47.7% (21/44) | 95.2% (20/21) | 80.0% (4/5) |
> | **NP** | - | - | 100.0% (1/1) | - | 100.0% (3/3) | - |
> | **CO** | - | - | 60.0% (21/35) | 33.3% (3/9) | 33.3% (1/3) | 100.0% (1/1) |
> | **MOP** | - | - | - | 37.5% (3/8) | - | - |
> | **Others** | - | - | 50.0% (5/10) | 25.0% (2/8) | - | - |
> | **Total** | **93.0% (93/100)** | **97.0% (97/100)** | **68.0% (68/100)** | **46.0% (46/100)** | **83.8% (31/37)** | **72.7% (8/11)** |
>
> The results show that **LLMOPT is universal in various types of optimization problems and can solve almost all kinds of optimization problems.** However, due to the different distribution of problem formulating difficulties, the accuracy of LLMOPT on these datasets is also different, which will be one of the focuses of future work.
>
>
> ## Response to Questions about the solver.
>
> Thank you for your attention to solver fairness! We provide a detailed description of the solvers used by LLMOPT and the comparison methods here.
>
> 1. **Solvers used by LLMOPT.** We use Pyomo [5] code in LLMOPT, which is a Python-based, open-source optimization modeling language with a diverse set of optimization capabilities. However, Pyomo is just a modeling language. In the learning process of LLMOPT, the training data consists of calling three open-source solvers (GLPK [6], IPOPT [7], and SCIP [8]) with the help of Pyomo code. Which solver to use for inference is at the discretion of the LLM. The codes used for training are labeled by experts as mentioned above.
> 2. **An introduction of the solvers.** GLPK (GNU Linear Programming Kit, [6]) focuses on solving linear programming (LP) and mixed integer linear programming (MILP) problems, and is suitable for small to medium-sized problems. IPOPT (Interior Point OPTimizer, [7]) is designed for solving large-scale nonlinear optimization problems (NLP), supports sparse constraints and quadratic programming, and performs well in the field of continuous optimization. SCIP (Solving Constraint Integer Programs, [8]) is a powerful and flexible solver that supports mixed integer nonlinear programming (MINLP) and complex constrained optimization (CP) problems, and is suitable for handling large-scale complex problems. **These three optimizers cover most types and sizes of optimization problems, and are all open source.**
> 3. **Solver details of baselines.** In OptiMUS [9] and Chain-of-Experts [10], the solver used is determined by the 'Solver' keyword in the SNOP representation, which is automatically determined by the method. In ORLM [11], the open source model [1] has lost its generalization performance and can only use the coptpy solver.
>
> Thanks again for your attention to the solver details! We will add these details to the paper.
>
> We hope that our response has addressed your concerns, but if we missed anything please let us know.

---

> ### Author Response · Authors · 2024-11-21
> **Reply to Reviewer JwJz (9/9)**
>
> **References**:
>
> [1] ORLM open-source model. [https://huggingface.co/CardinalOperations/ORLM-LLaMA-3-8B](https://huggingface.co/CardinalOperations/ORLM-LLaMA-3-8B)
>
> [2] ICML 2024 Challenges on Automated Math Reasoning (Task 3). [https://www.codabench.org/competitions/2438/](https://www.codabench.org/competitions/2438/)
>
> [3] Additional training data for SFT. [https://instructions.apps.allenai.org/](https://instructions.apps.allenai.org/)
>
> [4] Additional training data for KTO. [https://huggingface.co/datasets/argilla/ultrafeedback-binarized-preferences-cleaned-kto](https://huggingface.co/datasets/argilla/ultrafeedback-binarized-preferences-cleaned-kto)
>
> [5] Pyomo Modeling Language. [https://www.pyomo.org/](https://www.pyomo.org/)
>
> [6] GLPK (GNU Linear Programming Kit) solver. [https://www.gnu.org/software/glpk/](https://www.gnu.org/software/glpk/)
>
> [7] IPOPT (Interior Point OPTimizer) solver. [https://coin-or.github.io/Ipopt/](https://coin-or.github.io/Ipopt/)
>
> [8] SCIP (Solving Constraint Integer Programs) solver. [https://www.scipopt.org/index.php](https://www.scipopt.org/index.php)
>
> [9] Ali AhmadiTeshnizi, et al. OptiMUS: Scalable optimization modeling with (MI)LP solvers and large language models. ICML 2024.
>
> [10] Ziyang Xiao, et al. Chain-of-Experts: When LLMs meet complex operations research problems. ICLR 2024.
>
> [11] Zhengyang Tang, et al. ORLM: Training large language models for optimization modeling. CoRR, abs/2405.17743, 2024.
>
> [12] Kawin Ethayarajh, et al. Model alignment as prospect theoretic optimization. ICML 2024.
>
> [13] Definition of loss function in the KTO code. [https://github.com/huggingface/trl/blob/main/trl/trainer/kto_trainer.py#L1109-L1179](https://github.com/huggingface/trl/blob/main/trl/trainer/kto_trainer.py#L1109-L1179)
>
> [14] Settings of hyperparameters in the official KTO code. [https://github.com/huggingface/trl/blob/main/trl/trainer/kto_config.py#L86-L89](https://github.com/huggingface/trl/blob/main/trl/trainer/kto_config.py#L86-L89)
>
> [15] Rafael Rafailov, et al. Direct Preference Optimization: Your Language Model is Secretly a Reward Model. NeurIPS 2023.

---

> ### Comment · Area_Chair_neSr · 2024-11-25
> **ICLR Public Discussion Phase Ending Soon**
>
> Dear Reviewer,
>
> This is a kind reminder that the dicussion phase will be ending soon on November 26th. Please read the author responses and engage in a constructive discussion with the authors.
>
> Thank you for your time and cooperation.
>
> Best,
>
> Area Chair

---

> ### Comment · Reviewer_JwJz · 2024-11-26
> **Response to Rebuttal**
>
> I thank the authors for the hard work and gathering additional results for the rebuttal.
>
> ### KTO vs SFT
>
> I have another suggestion. Based on your clarification in `2. Datasets of SFT and KTO`, my understanding is that the SFT dataset contains only the preferred outputs, whereas the KTO dataset contains the preferred (labelled as 'desirable') and less preferred responses.
>
> In a way, SFT is finetuning only preferred responses. In the DPO paper, the authors ablated the two settings (i.e. what they termed `Preferred-FT` in S6 is equivalent to SFT stage here, if I understood correctly), and found DPO (based on preference dataset) is superior to `Preferred-FT` (SFT stage). I appreciate the ablation results, but they don't completely show that both stages are necessary. It would be useful to compare against an ablation where **only KTO is performed**.
>
> Do let me know if this is impractical given the rebuttal window.
>
> ### AST Clarification
>
> Thanks for clarifying this, I might have misunderstood AST as the runtime of the solver itself, which should not be that different for different formulations/code of the same LP problem.
>
> ### Evaluating formulation correctness
>
> I am slightly puzzled about why an LLM is used to evaluate correctness. Assuming that you have access to the 'ground-truth' objective value, is it not possible to compare directly against the optimal objective returned by the solver/program?
>
>
> ### Choice of solver
>
> Thanks for clarifying the use of solvers for LLMOPT and the considered baselines. Could the authors clarify whether the choice of solvers (and the framework-specific API) affects performance results? It seems that the authors have compared against OPTIMUS and ORLM using their original solvers and not the Pyomo wrapper.
>
>
> ### About dataset labelling
>
> I appreciate the detailed description of the data labelling process. Can the authors confirm (1) whether they plan on releasing the labelled dataset, (2) provide additional details on how the labelers were compensated and how working conditions were kept fair.
>
> As I mentioned before, the dataset contains a lot of problems, even if each problem can be labelled on the order of a few minutes (which is infeasible for more challenging problems), this is a huge undertaking.

---

> > ### Author Response · Authors · 2024-11-29
> > **Reply to Reviewer JwJz (1/2)**
> >
> > Thank you for your reply and constructive comments!
> >
> > ## Response: About KTO vs SFT
> > Thank you for your suggestion! We are glad that you recognized the results of the previous ablation experiments.
> >
> > In the DPO paper, the DPO method is conducted based on the SFT model because it requires a reference model to align newly generated content with existing quality benchmarks. Pre-trained models without SFT often lack sufficient domain-specific knowledge, making them unsuitable as reference models. That is the reason why we did not perform the experiment setting of only KTO.
> >
> > To address your concerns, we add the experiment you suggested, where only KTO is performed. Since the training and testing of the model requires time (approximately 2 days), we may share you the results later. Thank you for your patience!
> >
> > ## Response: AST Clarification
> > Your understanding is correct! The runtime of the solver should not be different. However, the AST, as the average number of correction processes performed, demonstrates the performance of the method and is therefore different. We are glad to clarify the definition of AST for you.
> >
> > ## Response: Evaluating formulation correctness
> > Since solver outputs often include print logs unrelated to the exact answer, a straightforward value-to-value comparison is unsuitable and can result in statistical inaccuracies.
> >
> > Typically, the solver output looks like this:
> >
> > ```
> > Model unknown
> >   Variables:
> >     DecisionVariableX : Size=2, Index=Elements
> >         Key : Lower : Value  : Upper : Fixed : Stale : Domain
> >           1 :     0 : 6000.0 :  None : False : False : NonNegativeReals
> >           2 :     0 : 1400.0 :  None : False : False : NonNegativeReals
> >   Objectives:
> >     TotalProfit : Size=1, Index=None, Active=True
> >         Key  : Active : Value
> >         None :   True : 192000.0
> >   Constraints:
> >     WeightedSumConstraint : Size=1
> >         Key  : Lower : Body : Upper
> >         None :  None : 40.0 :  40.0
> >     UpperBoundConstraints : Size=2
> >         Key : Lower : Body   : Upper
> >           1 :  None : 6000.0 : 6000.0
> >           2 :  None : 1400.0 : 4000.0
> > The optimal solution is:
> > Allocate 6,000 to production 1.
> > Allocate 1,400 to production 2.
> > The corresponding objective function value (total profit) is 192,000.
> > ```
> >
> > To evaluate accuracy, the optimal value found by the solver should first be extracted from the above string, i.e., "192,000". Then, the accuracy is determined by comparing this value with the ground truth.
> >
> > Firstly, extracting the objective value from solver logs using string matching is challenging. Secondly, matching values like “192000.0” with “192,000,” “2.666666667” with “2.67,” and other unexpected variations is equally difficult. A common approach is to use LLMs (such as GPT-4) for extraction and comparison through carefully designed prompts [1].

---

> > ### Author Response · Authors · 2024-11-29
> > **Reply to Reviewer JwJz (2/2)**
> >
> > ## Response: Choice of solver
> > Thanks for your question! Here is the differences between the solvers used by LLMOPT, ORLM, and OptiMUS.
> >
> > 1. LLMOPT can **automatically choose** the most suitable solver from **three open-source solvers** (covering nearly all optimization types) based on the problem type.
> > 2. **The close-source coptpy solver is the only choice of ORLM** [2, 3]. The coptpy solver is developed by Sunshu Technology.
> > 3. OptiMUS generates code by calling GPT-4, so it can use various solvers. However, OptiMUS has to **specify the solver manually**, and **only the close-source Gurobi** solver is used in [4].
> >
> > **On solving the optimization problem correctly, the key issue lies in correctly modeling the problem, generating the appropriate code**. Different solvers have minimal impact on the solution for the same type of problem. However, using the wrong solver for a problem can have significant consequences. For example, the GLPK solver does not support nonlinear optimization problems. If applied to such problems, it will lead to incorrect results. Therefore, we believe that LLMOPT’s ability to select the appropriate solver based on the problem type is a significant advantage. LLMOPT selects the most suitable solver from three open-source options. As mentioned in the previous response, these solvers are designed for different types of problems. Assigning a single specific solver to all problems would undoubtedly result in a decline in performance. Thank you for your question!
> >
> > ## Response: About dataset labelling
> > Thanks for your concerns about the dataset labeling! As mentioned above, these data used in this paper are labeling by 12 experts over a period of 1 month, which is turly a huge undertaking. In fact, we are continuing to collect and label new data from real industrial and financial scenarios. We will propose a comprehensive benchmark for leveraging LLMs to solve optimization problems, featuring a well-designed dataset and a complete evaluation process.
> >
> > We also place great importance on protecting the rights of experts. To support this project, we establish a collaboration fund between the anonymous enterprise and the anonymous university to ensure experts are fairly compensated. We also ensure a reasonable workload for them and provide dedicated working conditions. For instance, to reduce the typing burden, we provide each expert with access to GPT-4o APIs to assist in completing annotations efficiently. We firmly believe that high-quality data can only be produced when the rights of annotators are safeguarded.
> >
> > **References**:
> >
> > [1] Pan Lu, et al. MathVista: Evaluating Mathematical Reasoning of Foundation Models in Visual Contexts. ICLR 2024.
> >
> > [2] Zhengyang Tang, et al. ORLM: Training large language models for optimization modeling. CoRR, abs/2405.17743, 2024.
> >
> > [3] ORLM open-source model. [https://huggingface.co/CardinalOperations/ORLM-LLaMA-3-8B](https://huggingface.co/CardinalOperations/ORLM-LLaMA-3-8B)
> >
> > [4] Ali AhmadiTeshnizi, et al. OptiMUS: Scalable optimization modeling with (MI)LP solvers and large language models. ICML 2024.

---

> > ### Author Response · Authors · 2024-12-02
> > **Reply to Reviewer JwJz**
> >
> > Thank you for your patience. We conduct additional experiments for LLMOPT (only KTO), and the results are as follows.
> >
> > |  | **NL4Opt** | **MamoEasy** | **MamoComplex** | **IndustryOR** | **NLP4LP** | **ComplexOR** |
> > | :---: | :---: | :---: | :---: | :---: | :---: | :---: |
> > | **GPT-4o** | 83.0% | 90.0% | 38.0% | 34.0% | 35.2% | 36.4% |
> > | **LLMOPT (only SFT)** | 90.0% | 97.0% | 65.0% | 43.0% | 64.9% | 54.6% |
> > | **LLMOPT (only KTO)** | 90.0% | 95.0% | 57.0% | 42.0% | 56.8% | 45.5% |
> > | **LLMOPT (SFT+KTO)** | 93.0% | 97.0% | 68.0% | 46.0% | 83.8% | 72.7% |
> >
> >
> > The results indicate that **neither only SFT nor only KTO is sufficient to achieve optimal performance**. The only SFT configuration lacks model alignment, leading to hallucination issues. In the only KTO configuration, the pre-trained model is used as the reference model. However, without fine-tuning the reference model using domain-specific knowledge, the maximum performance improvement cannot be achieved (as the KL divergence between the two models is incorporated in the KTO loss).

---

> > ### Author Response · Authors · 2024-12-02
> > **Gentle Reminder of the Rebuttal Deadline**
> >
> > Dear Reviewer JwJz,
> >
> > As the deadline approaches, we sincerely hope to address your concerns and discuss the rebuttal with you further. If you have any questions, please feel free to ask directly! Moreover, if you find our response satisfactory, could you please kindly consider the possibility of updating the rating. Thank you very much for your valuable suggestion.
> >
> > Best regards,
> >
> > Authors

---

> > > ### Comment · Reviewer_JwJz · 2024-12-03
> > > **Thank you**
> > >
> > > I thank the authors for a detailed and committed rebuttal.
> > >
> > > The latest round of responses and ablation study have addressed all my main concerns. I encourage the authors to incorporate all the latest results in future versions of this paper. As such, I am happy to recommend acceptance of this work, and have changed the rating accordingly.
> > >
> > > Best,
> > >
> > > The Reviewer

---

> ### Author Response · Authors · 2024-12-03
>
> Dear Reviewer JwJz,
>
> We are delighted to hear that your concerns have been addressed and that you have raised the score. We are glad to participate in the discussion with you, and we will incorporate all the latest results in future versions of this paper.
>
> Thank you for recommending our paper for acceptance. We sincerely appreciate your valuable feedback and suggestion once again!
>
> Best regards,
>
> Authors

---

### Official Review · Reviewer_izRU · 2024-11-04

**Soundness:** 3
**Presentation:** 3
**Contribution:** 2
**Rating:** 6
**Confidence:** 4

**Summary:**

The authors of this paper introduced **LLMOPT**, a novel learning-based framework designed to enhance large language models' (LLMs) capability to define and solve general optimization problems described in natural language. The key innovation is the **five-element formulization**, which structures optimization problems into sets, parameters, variables, objectives, and constraints, enabling more accurate problem representation and solver code generation. LLMOPT also incorporates **multi-instruction supervised fine-tuning (SFT)** and **model alignment** using the Kahneman-Tversky Optimization (KTO) method to improve both the accuracy and generalization of solutions.

Another significant contribution is the **self-correction mechanism** in the auto-testing process, which automates error analysis and solution refinement during execution, ensuring robust performance without manual intervention in test-time. The framework was evaluated on six real-world datasets, covering diverse optimization types and fields, achieving an **average accuracy improvement of 11.08%** over state-of-the-art methods. This demonstrates LLMOPT's effectiveness in boosting both the accuracy and generalizability of LLMs in solving complex optimization problems.

**Strengths:**

- The paper introduces a *novel five-element formulization* to define optimization problems, which significantly enhances the ability of large language models (LLMs) to accurately interpret and transform natural language descriptions into solvable optimization problems. This structured approach captures the essential elements of optimization scenarios, ensuring clearer problem representation and facilitating better code generation. The inclusion of elements such as sets, parameters, variables, objectives, and constraints helps the model produce more precise solutions and reduces the risk of omitting implicit problem aspects.

- Another notable feature is the *self-correction in test-time*, an automated process called *Auto-testing* integrated within LLMOPT to identify and rectify errors in generated solver code during execution. This mechanism analyzes output logs and determines if adjustments are necessary, enhancing the robustness and accuracy of problem-solving without manual intervention. By automatically looping back to problem reformulation or code generation when needed, LLMOPT can iteratively improve the accuracy of its solutions and adapt effectively to complex challenges.

- The paper boasts *strong evaluation results*, demonstrated by extensive testing across six real-world datasets encompassing approximately 20 different fields and various optimization problem types, such as linear and nonlinear programming and combinatorial optimization. LLMOPT shows superior performance, achieving a notable 11.08% average improvement in solving accuracy compared to existing state-of-the-art methods. This result underscores the framework’s generalization capabilities and effectiveness in diverse optimization scenarios, validated by comprehensive comparisons and ablation studies.

**Weaknesses:**

1. **Complexity of Data Labeling**: The proposed five-element formulation and the use of expert labeling for optimization problem formulations and solver code rely significantly on manual validation. While the authors mention the use of GPT-4 to assist in data generation, human experts are still required to verify the correctness of the outputs. This introduces potential scalability issues as extensive human oversight could limit the practicality of the approach, especially in large-scale deployments. Additionally, the authors do not provide a qualitative or quantitative analysis of the labeling quality or reliability performed by the human experts. Evidence demonstrating the expertise and qualifications of these experts should be presented to support the validity of the labeling process.

2. **Insufficient Theoretical Justification for the Five-element Formulation**: The authors claim that the five-element formulation is a universal method for defining optimization problems, but they do not provide sufficient references or theoretical analysis to support this claim of 'universal'. Similarly, in the experimental results, a detailed and decomposed explanation of why the five-element formulation generalizes well and outperforms other models across different types of problems is lacking. Providing theoretical or empirical grounding beyond performance metrics would strengthen the argument for the effectiveness and universality of this approach.

3. **More comparative Analysis of the Self-correction Mechanism**: While the authors compare the performance of LLMOPT with and without the self-correction mechanism, it would be beneficial to include an analysis of the performance gains when self-correction is combined with baseline methods (e.g., GPT-4 directly). Additionally, a longitudinal comparison between the current self-correction design and other correction methods, such as manual debugging, could further elaborate on the advantages and limitations of the self-correction approach.

**Questions:**

**Questions and Suggestions for the Authors:**

1. **Clarification on Expert Involvement in Data Labeling**:
   - **Question**: Could the authors provide more details on the qualifications, expertise, or at least agreement rates of the human experts involved in the data labeling process?
   - **Suggestion**: Including a quantitative or qualitative analysis of expert reliability and consistency would strengthen the validity of the data labeling claims. Evidence such as statistics of certifications or relevant experience, and agreement rates among experts would be valuable to better assess the credibility of the labeled data.

2. **Theoretical Justification for the Five-element Formulation**:
   - **Question**: What is the theoretical basis for claiming that the five-element formulation is a universal method for defining optimization problems?
   - **Suggestion**: Providing additional references or an in-depth theoretical analysis that supports the universality and applicability of the five-element formulation across various optimization scenarios would enhance the argument. Empirical comparisons to alternative problem formulations could also be beneficial.

3. **Comparative Performance Analysis of Self-correction Mechanism**:
   - **Question**: How does the self-correction mechanism in LLMOPT compare with similar correction methods, such as manual debugging or integration with other LLMs or methods like GPT-4 in a correction loop?
   - **Suggestion**: Presenting a detailed comparison of LLMOPT’s self-correction mechanism with existing correction strategies or showing performance metrics of LLMOPT combined with baseline models could provide insights into its relative effectiveness. This would help demonstrate whether the self-correction offers unique advantages or is comparable to simpler, established methods.

4. **Scalability Concerns**:
   - **Question**: How do the authors envision scaling the current framework for large-scale practical deployments given the heavy reliance on expert validation?
   - **Suggestion**: Suggestions for automating or streamlining the expert review process, possibly by incorporating semi-supervised or active learning techniques, could address concerns about scalability and long-term sustainability of the labeling process.

5. **Detailed Explanation of Generalization Capabilities**:
   - **Question**: Can the authors provide more specific examples or case studies where the five-element formulation showed clear advantages in generalization compared to alternative models?
   - **Suggestion**: Adding empirical evidence or a breakdown of performance across different types of optimization problems, particularly ones not included in the training set, would strengthen the claim of its broad applicability and effectiveness.

6. **Addressing Potential Limitations in Self-correction**:
   - **Question**: How does the self-correction mechanism handle potential limitations or biases in repeated corrections, such as overfitting to a specific type of error?
   - **Suggestion**: Discussing safeguards or mechanisms in place to ensure diverse error handling and avoiding biased correction loops would provide more confidence in the robustness of the self-correction feature.

**Details Of Ethics Concerns:**

**Ethical Concern**:
The paper should include a statement regarding the treatment and compensation of human expert labelling annotators, as per conference requirements. Transparency in the ethical treatment, fair wages, and working conditions of these contributors is important. The authors are encouraged to provide details about how expert annotators were compensated and ensure that their work adhered to ethical standards. This declaration would reinforce the ethical integrity of the research and ensure compliance with conference guidelines on responsible and fair treatment of human contributors involved in the research process.

---

> ### Author Response · Authors · 2024-11-21
> **Reply to Reviewer izRU (1/5)**
>
> Thank you for your valuable questions raised in the review! We would like to discuss them with you, and if there are any other questions, please feel free to ask, and we will respond promptly.
>
> ## About data labeling.
>
> ### Response to Question 1: Expert involvement in data labeling.
>
> Thank you for your concerns about data labeling! First, we introduce **the process of data labeling**, which is divided into four stages: preliminary review, expert labeling, expert review, and data aggregation.
>
> + _**Preliminary review**_**.** Initially review the data and remove unfeasible problems (unfeasible not means difficult). With the help of GPT-4o, we divide optimization problems into two categories according to their difficulty. Problems that meet one of the following conditions will be classified as difficult problems: at least 3 out of 5 solutions using GPT-4o are inconsistent, the code generated by GPT-4o has errors, and experts have found complex constraints, reasoning, or large amounts of data.
> + _**Expert labeling**_**.** For simple questions, 2 experts independently annotate. And 3 experts independently label for complex questions. For each question, five-element and solver code need to be labeled, and the code must be run without errors. In this stage, experts may use GPT-4o to generate text that meets the expert's intentions to reduce typing time and generate more formatted code. In order to improve data quality, experts may modify questions appropriately to make them more suitable for the problem scenario, or delete inappropriate questions (such as unfeasible ones).
> + _**Expert review**_**.** For each question, a new expert checks the labels of other experts in the previous step, which is based on the correctness of the problem modeling and the consistency of the labels of different experts (the same question may have different but correct labels from different experts). Highly controversial questions or those with any incorrect labels are included in a independent challenging dataset.
> + _**Data aggregation**_**.** Five experts discuss and analyze each of the data in the independent difficult dataset to determine whether the problem has a solution or can be adjusted to a feasible problem, and decide whether to abandon the problem based on this. If not, the experts will discuss and complete the labeling of these data. The correctly labeled questions that has passed the expert review will be summarized. And the other feasible questions that has been incorrectly labeled during the entire labeling process will be summarized.
>
> Then, we anonymously introduce the experts' qualifications. The above steps are completed by 12 experts. The "preliminary review" and "expert labeling" are finished by 9 undergraduate students with bachelor's degrees or above (computer science or mathematics, all of whom have taken optimization courses), including 4 master's students in related fields (1 doctoral student). The "expert review" is finished by 1 university professor whose research is optimization in machine learning and 2 algorithm engineers working on operations research optimization. "Data aggregation" is finished by the experts except undergraduates. During the data labeling process, we ensure that **each expert completed the labeling independently**. In the first three stages, **a question would not be assigned to the same expert twice.**
>
> Finally, we explain **the effectiveness of the expert labels**. The review pass rate for the seven experts assigned to simple question labeling exceeds 90%, while the pass rate for the four experts labeling complex question labeling is above 80% (2 experts labeled both simple and complex questions). Considering the above statistics, along with the fact that **the labeling results will undergo review by senior experts in the third stage**, the reliability and effectiveness of the expert labeling process are well-supported.
>
> Thanks for your suggestion! We will add these details about expert labeling to the paper.

---

> ### Author Response · Authors · 2024-11-21
> **Reply to Reviewer izRU (2/5)**
>
> ### Response to Question 4: Scalability.
>
> We are glad that you are paying attention to the data problem! We share you our understanding of high-quality data and our attempts to make data scalable.
>
> **Why do we spent a lot of manpower and resources to manually label data**? As discussed in Section 5 of the paper, high-quality data plays a critical role in the performance of the model. However, there is a lack of open-source data in the field of optimization, and the available data is often disorganized and unreliable. For example, the following problem is an example from the NL4Opt dataset [1], which has been open-sourced by ORLM. This is  a simple _integer programming problem_ (with only two variables and two linear constraints). The problem is as follows.
>
> > A chair produced by Elm Furniture yields a profit of 43, while every dresser yields a 52 profit. Each week, 17 gallons of stain and 11 lengths of oak wood are available. Each chair requires 1.4 gallons of stain and 2 lengths of oak wood, while each dresser requires 1.1 gallons of stain and 3 lengths of oak wood. Determine the maximum profit.
>
> The answer provided by ORLM is 236.5. A clear inconsistency arises here: given that the profits for both chairs and dressers are integers, it is illogical for the maximum profit to be a decimal. Upon verification, this incorrect result stems from the solution of producing 5.5 chairs and no dressers. The correct ground truth should be 224, achieved by producing 4 chairs and 1 dresser. Errors in labeling are not uncommon in open-source datasets, and various types of mistakes are universal across different datasets, such the lack of ground truth, the infeasibility of the problem. This further aggravates the challenges posed by the already limited availability of open-source data. **To address this issue, we dedicate approximately one month to creating a high-quality training dataset with the help of 12 experts**. This effort aims to ensure higher-quality model training and more reliable performance evaluation.
>
> At the same time, we have further studied the issue of data scalability from two perspectives. **First**, leveraging high-quality data labeled by human experts, we applied various automated data augmentation techniques to enrich our dataset. **Second**, we are now exploring reinforcement learning methods, such as Monte Carlo Tree Search (MCTS), to generate reasoning paths by utilizing GPT-4 to produce high-quality annotation data. For example, Pyomo does not support strict inequality constraints directly, so constraints like x > y often need to be transformed into non-strict inequalities like x >= y + 1e-6. By providing such examples, we aim to guide GPT-4o in generating _processes_, which will guide more accurate modeling and code. Once the model learns to transform strict inequalities, it should also be able to autonomously understand transformations for absolute value constraints. This ability will be a key factor influencing the scalability of automatic data annotation.

---

> ### Author Response · Authors · 2024-11-21
> **Reply to Reviewer izRU (3/5)**
>
> ## About generalization and the five-element formulation.
>
> ### Response to Question 5: Detailed explanation of generalization capabilities
>
> Thank you for your suggestion to the five-element! Through two new experiments, we explain the generalization performance of LLMOPT from multiple perspectives.
>
> **Experiment 1: Comparative experiment on the generalization performance of five-element**.
>
> Based on 200 test data of different scenarios in the NL4Opt and IndustryOR datasets, experiments are deployed based on GPT-4o to compare the accuracy of solutions under different formulations. The instructions used for the open formulation is: _Please model the following optimization problem, and then write the corresponding pyomo code based on the modeling to solve the problem._
>
> In order to demonstrate the generalization performance of the five-element, we classify the data according to the problem scenario (the number of data is in brackets). The solving accuracy results under different formulations are shown in the following table:
>
> |  | **w/o Formulation** | **Open Formulation** | **CoT [2]** | **Five-element Formulation** |
> | :---: | :---: | :---: | :---: | :---: |
> | Manufacturing (62) | 53.2% | 58.1% | 53.2% | **59.7%** |
> | Health (23) | 65.2% | **78.3%** | 69.6% | **78.3%** |
> | Retail (23) | 56.5% | **65.2%** | 60.9% | **65.2%** |
> | Transportation (33) | 72.7% | 75.8% | **81.8%** | **81.8%** |
> | Agriculture (16) | 56.3% | 56.3% | 56.3% | **68.8%** |
> | Others (43) | 51.2% | 55.8% | 54.8% | **62.8%** |
> | All (200) | 58.0% | 63.5% | 61.0% | **67.5%** |
>
> The results show that: (a) **the five-element formulation has obvious advantages over other formulations in all scenarios.** (b) the solving accuracy of any formulation is higher than that of directly generating solving code. Using formulation as an intermediate process is conducive to generating correct code.
>
> **Experiment 2: Generalization performance on new dataset.**
>
> We find a new dataset on the ICML 2024 Challenges on Automated Math Reasoning (Task 3) [3], whose data is not used for the training of LLMOPT. Since the test dataset does not open source the ground truth, we randomly selected 200 pieces from the training data as test data. The solving accuracy results are shown in the following table:
>
> |  | **GPT-4o** | **GPT-4o + 5-elem** | **LLMOPT w/o 5-elem** | **LLMOPT** |
> | :---: | :---: | :---: | :---: | :---: |
> | The Competition Dataset | 78.5% | 81.5% | 87.0% | **89.5%** |
>
> From the above results, we can see that (a) **LLMOPT achieved the best performance on the new dataset,** which is 8.0% higher than GPT-4o; (b) on the new dataset, whether LLMOPT or GPT-4o, the five-element as an intermediate process brought significant performance improvement.
>
> It is worth to note that, in the experimental results shown in Table 2 of the paper, the data in NLP4LP and ComplexOR did not participate in the learning process and data augmentation. However, **the results still show that LLMOPT improves by 11.8% and 6.0% on these two datasets respectively, achieving SOTA performance** and demonstrating the generalization performance of LLMOPT.

---

> ### Author Response · Authors · 2024-11-21
> **Reply to Reviewer izRU (4/5)**
>
> ### Response to Question 2: In-depth analysis of five-element formulation.
>
> In order to generate the correct solver code, we believe that LLM should first learn to define the optimization problem, so a formulation is needed as an intermediate process for generating code. When choosing a suitable formulation, on the one hand, we considered that **the formulation should correspond one-to-one with the definition of the optimization problem**, that is, the problem that can be written as formula (1) in Section 3.2.1 of the paper can be written with this formulation. On the other hand, since the mathematical expression is too abstract, **this formulation should include a more detailed description of the problem than formula (1)** **and should be easier to convert into Pyomo code**. Considering the above two aspects, we propose the five-element: a formulation that aligns one-to-one with the mathematical definition of the optimization problem while also incorporating the necessary problem description.
>
> Intuitively, **any optimization problem that can be expressed in the form of formula (1) can be fully described by the five-element formulation, as the five-element is designed to correspond one-to-one with formula (1).** Specifically, variables, objectives, and constraints represent the core components of the optimization problem. To facilitate better code generation, sets and parameters are included as optional  descriptions, completing the five-element formulation.
>
> In order to illustrate that the five-element formulation are applicable to various optimization problems, we have given 7 examples of using the five-element to define different optimization problems in Appendix J, including linear programming, integer programming, mixed-integer programming, nonlinear programming, and combinatorial optimization (0-1 knapsack problem, traveling salesman problem), to illustrate the applicability of the five-element to various problems.
>
> ## About self-correction.
>
> ### Response to Question 3: Comparisons performance analysis to self-correction mechanism.
>
> To further explore the superiority of self-correction and LLMOPT, we deploy experiments on the NL4Opt and IndustryOR datasets (NL4Opt has relatively simple problems, while IndustryOR has relatively complex problems). We change two correction mechanisms, one is correction by GPT-4o with the same prompt of self-correction, and the other is to repeat the inference 12 times and manually judge the optimal solution (which means that only one optimal solution needs to be found in 12 repeated experiments). The reason we chose 12 is that self-correction is limited to a maximum of 12 repeated checks, so this is fair. We also conducted experiments on GPT-4o and ORLM [4]. (We reproduced the open source model of ORLM [4], but found that this model seems to have lost other abilities except writing coptpy code for optimization problems. We find that ORLM has a serious seesaw problem,  which is performance as without generalization ability, and cannot answer other questions. Therefore, only the "Best of 12 repeats" correction mechanism is experimented.) The results are as follows:
>
> | **Inference Model** | **Correction Mechanism** | **IndustryOR Dataset** | **NL4Opt Dataset** |
> | :---: | :---: | :---: | :---: |
> | LLMOPT (Qwen-1.5) | Self-correction | **46.0%** | **93.0%** |
> | LLMOPT (Qwen-1.5) | Correction by GPT-4o | 41.0% | 89.0% |
> | LLMOPT (Qwen-1.5) | Best of 12 repeats | 42.0% | 89.0% |
> | GPT-4o | Correction by GPT-4o | 34.0% | 84.0% |
> | GPT-4o | Best of 12 repeats | 32.0% | 84.0% |
> | ORLM [4] | Best of 12 repeats | 39.0% | 88.0% |
>
> The results show that: (a) When LLMOPT (Qwen-1.5) is used as the inference model, the correction performance of GPT-4 is lower than the self-correction solving accuracy of LLMOPT. This indicates that the Qwen-1.5 model learned by LLMOPT shows stronger overall capabilities in both solving optimization problems and correction compared to other methods. (b) Although manually selecting the best result from 12 repetitions shows performance improvement (considering once solving correct if one out of 12 repetitions is accurate), it still falls short of the effectiveness compared with the self-correction mechanism. **This highlights that identifying and correcting errors is more critical than simply repeating executions, emphasizing the necessity of implementing a correction mechanism.**

---

> ### Author Response · Authors · 2024-11-21
> **Reply to Reviewer izRU (5/5)**
>
> ### Response to Question 6: Further explanation for self-correction mechanism.
>
> Thank you for your interest in the details of the self-correction mechanism. When querying an LLM about code-related issues, the LLM may exhibit a tendency to fall into a loop of flawed reasoning during multi-turn conversations. Therefore, in our self-correction implementation, we do not input all historical information into the LLM. As shown in the instruction template in Listing 4 of Appendix H, each correction is handled as an independent call to the LLM, focusing solely on the correction of the current five-element and code without including history data or relying on multi-turn dialogue. **This approach helps to avoid the reasoning loops caused by multi-turn interactions and enhances the robustness of the correction process.**
>
> The robustness of the self-correction mechanism is one of the key topics we focus on, and we are actively exploring new approaches. One promising direction is **leveraging reinforcement learning to guide the LLM through a step-by-step self-correction process** [5]. Unlike the self-correction mechanism in LLMOPT, this approach breaks the correction process into multiple steps, using reinforcement learning to search for the correct correction logic chain. [5] has shown that this approach enables more precise problem identification and effective solutions. We are currently working on constructing relevant datasets and conducting experiments, with preliminary results indicating that this method could be more effective than the original self-correction. Thank you again for your interest!
>
> We hope that our response has addressed your concerns, but if we missed anything please let us know.
>
> **References**:
>
> [1] NL4Opt dataset. [https://huggingface.co/datasets/CardinalOperations/NL4OPT/viewer](https://huggingface.co/datasets/CardinalOperations/NL4OPT/viewer)
>
> [2] Wei, Jason, et al. Chain-of-thought prompting elicits reasoning in large language models. Advances in neural information processing systems 35 (2022): 24824-24837.
>
> [3] ICML 2024 Challenges on Automated Math Reasoning (Task 3). [https://www.codabench.org/competitions/2438/](https://www.codabench.org/competitions/2438/)
>
> [4] ORLM open-source model. [https://huggingface.co/CardinalOperations/ORLM-LLaMA-3-8B](https://huggingface.co/CardinalOperations/ORLM-LLaMA-3-8B)
>
> [5] Aviral Kumar, et al. Training language models to self-correct via reinforcement learning. 2024. [https://arxiv.org/pdf/2409.12917](https://arxiv.org/pdf/2409.12917)

---

> ### Comment · Area_Chair_neSr · 2024-11-25
> **ICLR Public Discussion Phase Ending Soon**
>
> Dear Reviewer,
>
> This is a kind reminder that the dicussion phase will be ending soon on November 26th. Please read the author responses and engage in a constructive discussion with the authors.
>
> Thank you for your time and cooperation.
>
> Best,
>
> Area Chair

---

> ### Author Response · Authors · 2024-12-02
> **Gentle Reminder of the Rebuttal Deadline**
>
> Dear Reviewer izRU,
>
> As the deadline approaches, we sincerely hope to address your concerns and discuss the rebuttal with you further. If you have any questions, please feel free to ask directly! Moreover, if you find our response satisfactory, could you please kindly consider the possibility of updating the rating. Thank you very much for your valuable suggestion.
>
> Best regards,
>
> Authors

---

> ### Author Response · Authors · 2024-12-03
>
> Dear Reviewer izRU,
>
> As the deadline approaches, we sincerely hope that your concerns have been addressed. If you have any questions, please feel free to ask directly!
>
> We have provided detailed responses regarding the data labeling process and the necessity of involving human experts. To address your concerns, we have also conducted ablation experiments on the five-element formulation and self-correction mechanism. The results show that the five-element outperforms other formulation approaches, achieving better performance and superior generalization on new datasets. Additionally, the experiments highlight the advantages of self-correction and LLMOPT can improve the capability of handling corrections. The specific details have been thoroughly discussed in our previous responses.
>
> We sincerely appreciate your effort and constructive comment once again!
>
> Best regards,
>
> Authors

---

### Author Response · Authors · 2024-11-21
**General Response to Reviewers and Revision Submitted**

We would like to express our heartfelt gratitude to all the reviewers for their valuable feedback and constructive suggestions. We are encouraged by their recognition of our proposed methods as novel (`Reviewer JwJz`, `Reviewer iJY5`), meaningful (`Reviewer JwJz`), and incorporating a well-designed five-element formulation (`Reviewer izRU`). Additionally, we appreciate the reviewers' acknowledgment of our comprehensively designed experiments and ablation studies (`Reviewer izRU`, `Reviewer JwJz`, `Reviewer iJY5`, `Reviewer feJ7`) and for noting that our paper is well-written (`Reviewer feJ7`) and includes detailed discussion (`Reviewer iJY5`).

**We have also made revisions to the manuscript. Below, we provide a summary of the key revisions of the paper (highlighted in blue text in the PDF)**.

1. We provide a detailed description of the process of data labeling and augmentation in Appendix K. We explain the role of experts in the process and the reliability of data labeling. (`Reviewer izRU`, `Reviewer JwJz`)
2. We add more detailed ablation experiments on the self-correction mechanism and analyze the experimental results, which are presented in Appendix M. The results show the overall performance improvement brought by LLMOPT, including enhancements in the formulation, solving, and correction capabilities of LLMs. (`Reviewer izRU`, `Reviewer iJY5`, `Reviewer feJ7`)
3. We provide a more detailed analysis of the results in Appendix L, including a statistical breakdown of solving accuracy by problem types, highlighting the improvement in generalization performance achieved by LLMOPT. (`Reviewer izRU`)
4. We have carefully reviewed Section 3.3.2 of the paper, supplemented the explanation of the model notations in equation 3 (`Reviewer iJY5`), added the purpose of KTO, andrevised the description of the conditions in equation 4 and equation 5 (`Reviewer JwJz`).
5. We carefully reviewed our paper and revised the Tyops. (`Reviewer iJY5`)

**Detailed responses to each reviewer’s comments are provided separately. If you have any questions or concerns, please do not hesitate to reach out. We are committed to providing detailed and professional responses to address any concerns you may have as promptly as possible**.

---

### Author Response · Authors · 2024-11-25
**Looking forward to your reply**

Dear reviewers:

We would like to draw your attention to our paper. We believe that the majority of the concerns raised by the reviewers regarding our paper revolve around the further clarification of LLMOPT. During the rebuttal period, we diligently addressed these concerns by providing point-to-point responses, which included **the incorporation of new experiments to clarify potential misunderstandings** (such as ablation analysis for KTO, more ablation analysis for self-correction, generalization analysis of five-element, and results by problem type) and enhancing the clarity of data and the learning process (such as **the process of data labeling, the details of KTO, examples of code generated by LLMOPT**, and **LLMOPT compared with ORLM and OptiMUS**).

While resolving each reviewer's concerns has required a significant amount of time and effort on our part, we have noticed that there has been no response from all reviewers.

The deadline for author-reviewer discussion is approaching. We sincerely hope that our efforts and improvements can be taken into consideration. We kindly request your assistance in reminding the reviewers of our responses and enhancements.

We sincerely appreciate your valuable time!

Thanks and regards,

Authors

---

### Meta-Review · Area_Chair_neSr · 2024-12-19

**Metareview:**

This paper introduces LLMOPT, which fine-tunes LLMs to enhance their ability to model optimization problems from natural language descriptions. Specifically, the fine-tuning process consists of two stages: (1) the supervised fine-tuning (SFT) stage, where the LLM is trained using MLE on ground truth outputs, and (2) the KTO-based alignment stage, where the LLM is fine-tuned with desirability labels applied to its generated outputs. LLMOPT employs a novel five-element formulation as a universal framework for defining diverse types of optimization problems, subsequently generating solver code based on this formulation. Experiments on six real-world datasets demonstrate LLMOPT's effectiveness in improving the accuracy and generalizability of LLMs in solving complex optimization problems.

Most reviewers agree that this paper makes meaningful contributions. During the rebuttal phase, the authors addressed the majority of the reviewers' concerns, and I suggest that the authors revise the manuscript accordingly in the final version.

In response to the ethical concerns about data labeling, the authors provided detailed explanations on the annotation process and the background of the annotators. Therefore, I believe further ethical review is unnecessary. Nevertheless, I strongly recommend that the authors provide the additional details (including aspects such as the methodology, the reliability, and the annotator compensation) in the final version.

Overall, I recommend accepting this paper.

**Additional Comments On Reviewer Discussion:**

Reviewers izRU, JwJz, iJY5, and feJ7 rated this paper as 6: borderline accept (keep the score), 3: reject (raised to 6), 5: borderline reject (keep the score), and 5: borderline reject (keep the score), respectively.

The reviewers raised the following concerns.
- Complexity of Data Labeling (raised by Reviewers izRU and iJY5).
- Insufficient Experiments (raised by Reviewers izRU and JwJz)
- Unclear Experiment Details (raised by Reviewers izRU, JwJz, and feJ7)
- Scalability (raised by Reviewers feJ7)
- Ethical Concerns Regarding Data Labeling (raised by Reviewer izRU)

In response, the authors addressed the concerns about the complexity of data labeling, insufficient experiments, and unclear experiment details by adding additional experiments and more clarifications in the rebuttal phase. Reviewer feJ7 preserves the concerns on scalability. However, scalability remains a common challenge in this domain, and for large-scale problems, we typically rely on structured data acquisition methods rather than extracting variable information directly from natural language descriptions. Despite the scalability issue, this paper still offers meaningful contributions. I strongly encourage the authors to incorporate the discussions to further address the concern in the final version.

Regarding the ethical concerns raised by izRU about data labeling, the authors have provided a detailed explanation on the annotation process and the background of the annotators. Therefore, I believe further ethical review is unnecessary. Nevertheless, I strongly recommend that the authors provide the additional details (including aspects such as the methodology, the reliability, and the annotator compensation) in the final version.

Overall, I recommend accepting this paper.

---

### Decision · Program_Chairs · 2025-01-22

Accept (Poster)

---

> ### Public Comment · ~Serdar_Kadioglu1 · 2025-08-30
> **Thank you & Related Work**
>
> Dear Authors,
>
> Thank you for your valuable contribution to the growing field of translating natural language into model specifications.
>
> Your proposed 5-element framework (covering sets, parameters, variables, constraints, and objectives) closely aligns with the approach taken in Ner4Opt published earlier, which extracts precisely the same components from natural language using named entity recognition. This problem of extracting these five core elements is referred to as the Ner4Opt Problem, along with several approaches for its resolution (citation below). Hope this reference proves helpful in your future work.
>
> https://dl.acm.org/doi/10.1007/s10601-024-09376-5
>
> Ner4Opt: Named Entity Recognition for Optimization Modelling from Natural Language
> Serdar Kadioglu, Parag Pravin Dakle, Karthik Uppuluri, Regina Politi, Preethi Raghavan, SaiKrishna Rallabandi, Ravisutha Srinivasamurthy
>
> Additionally, I’d like to highlight the Text2Zinc dataset and its associated leaderboard, which may serve as a valuable benchmark for evaluating your future approaches:
>
> https://arxiv.org/abs/2503.10642
>
> https://huggingface.co/datasets/skadio/text2zinc
>
> https://huggingface.co/spaces/skadio/text2zinc-leaderboard
>
> Text2Zinc: A Cross-Domain Dataset for Modeling Optimization and Satisfaction Problems in MiniZinc
> Akash Singirikonda, Serdar Kadioglu, Karthik Uppuluri Dataset: https://huggingface.co/datasets/skadio/text2zinc
>
> Thank you once again for your contributions and for making your work and code publicly accessible. It’s exciting to see the community advancing in this direction.
>
> Best regards,
> Serdar Kadioglu